# The Vps13-like protein BLTP2 regulates phosphatidylethanolamine levels to maintain plasma membrane fluidity and breast cancer aggressiveness

Subhrajit Banerjee[1], Stephan Daetwyler [2], Xiaofei Bai[3,4], Morgane Michaud[5], Juliette Jouhet[5], Derk Binns[1], Shruthi Madhugiri[2], Emma Johnson[1], Chao-Wen Wang[6], Reto Fiolka[2], Alexandre Toulmay [1]✉ & William A. Prinz [1]✉

Lipid transport proteins (LTPs) facilitate non-vesicular lipid exchange between cellular compartments and have critical roles in lipid homeostasis. A recently identified family of bridge-like LTPs (BLTPs) is thought to form lipid-transporting conduits between organelles. One of these, BLTP2, is conserved across species but its function is not known. Here we show that BLTP2 regulates plasma membrane (PM) fluidity by increasing phosphatidylethanolamine (PE) levels in the PM. BLTP2 localizes to endoplasmic reticulum (ER)–PM contact sites, and transports PE in vivo, suggesting it drives PE movement from ER to PM. We find that BLTP2 works in parallel with another pathway that regulates intracellular PE distribution and PM fluidity. BLTP2 expression correlates with breast cancer aggressiveness. We found that BLTP2 facilitates growth of a triple negative breast cancer cell line and sustains its aggressiveness in an in vivo model of metastasis, suggesting maintenance of PM fluidity by BLTP2 may be critical for tumorigenesis in humans.

Lipid transport proteins (LTPs) bind lipids and transfer them between membranes. There are many families of LTPs. Most are thought to transport lipids by binding a lipid monomer and shuttling it between membranes[1]. However, some LTPs are thought to transport lipids by a different mechanism: they interact with two membranes simultaneously and form a hydrophobic conduit that allows lipids to flow between membranes. The endoplasmic reticulum (ER)–mitochondria encounter structure, which bridges the ER and mitochondria in *Saccharomyces cerevisiae* (hereafter yeast), has recently been proposed to form a lipid-transporting tunnel between these organelles[2]. A similar lipid transport method is thought to be used by a family of LTPs

called bridge-like LTPs (BLTPs)[3]. BLTPs have rod-like structures with a hydrophobic channel formed by β-sheets that extend the length of the proteins. Proteins in this family have proposed lengths of 10–30 nm, long enough to interact with two organelles that are closely apposed, regions called membrane contact sites[4]. There are ten BLTPs in humans. The functions of most are not well understood, but they are thought to facilitate high-volume lipid transport.

The protein BLTP2 is conserved from yeast to human. Research in plants and *Drosophila* revealed that mutations in BLTP2 homologues reduce growth and cause defects in vesicular trafficking and phosphoinositide metabolism[3]. A recent study suggests that human BLTP2

[1]Department of Cell Biology, University of Texas Southwestern Medical Center, Dallas, TX, USA. [2]Lyda Hill Department of Bioinformatics, University of Texas Southwestern Medical Center, Dallas, TX, USA. [3]Department of Biology, University of Florida, Gainesville, FL, USA. [4]Genetics Institute, University of Florida, Gainesville, FL, USA. [5]Université Grenoble Alpes, CNRS, CEA, INRAE, IRIG, LPCV, Grenoble, France. [6]Department of Life Sciences, National Cheng Kung University, Tainan, Taiwan. ✉e-mail: Alexandre.Toulmay@UTSouthwestern.edu; william.prinz@utsouthwestern.edu

operates at organelle contact sites and inhibits ciliogenesis in retinal pigment epithelial-1 cells[5]. The finding that mice lacking BLTP2 are embryonic lethal suggests BLTP2 has an essential role in mammalian development[6,7]. BLTP2 has also been implicated in breast cancers. It is overexpressed in some breast cancer and associated with poor prognosis[8–10]. Profiling of the breast tumours reveals that the BLTP2 gene is frequently amplified[11]. BLTP2 has also been suggested to modulate the aggressiveness of both basal-like and non-basal-like breast cancer cells[12].

In this study, we investigated the function of BLTP2 in human and yeast cells. We find that BLTP2 and one of its homologues in yeast operate at contact sites between the ER and plasma membrane (PM). We show BLTP2 transports phosphatidylethanolamine (PE) from the ER to the PM, which helps maintain the fluidity and function of the PM, suggesting BLTP2 is part of a pathway that maintains PM fluidity. We also find that BLTP2 sustains proliferation of the triple negative breast cancer (TNBC) cell line MDA-MB-231 in a zebrafish xenograft model, indicating it is pro-tumoural.

## Results

### BLTP2 supports growth of invasive breast cancer cell lines

To investigate the importance of BLTP2 in cancer, we examined data from the Cancer Dependency Map (DepMap) and found that *BLTP2* is highly expressed and required for optimal grown of many invasive breast cancer cell lines (Fig. 1a), suggesting a critical role in breast cancer progression[13,14]. We found that CRISPR–Cas9-mediated knock-out of *BLTP2* (*BLTP2*-KO) in the cervical cancer cell line HeLa also reduced its rate of proliferation (Fig. 1b). Together, these results suggest that *BLTP2* is important for cell growth in human cells and may be particularly important for rapid proliferation of some cancer cells.

### Yeast lacking Fmp27 have defective PE homeostasis

To get more insight into BLTP2 function, we turned to yeast, which has two BLTP2 homologues: Fmp27 and Hob2 (ref. 15). Human BLTP2 and Fmp27 have similar predicted structures (Fig. 1c). Since the functions of Fmp27 and Hob2 are unknown, we sought to identify phenotypes for cells lacking either protein (*fmp27Δ* and *hob2Δ*, respectively) and found that *fmp27Δ* cells grew poorly at low temperatures (Fig. 1d). Consistent with a role for Fmp27 in cold adaptation, levels of endogenous Fmp27 increased fourfold when cells are grown at low temperature (18 °C; Fig. 1e). We ruled out that growth at low temperature alters the localization of Fmp27 as it is primarily found at ER–PM contact sites at both 30 °C (ref. 16) and 18 °C (Extended Data Fig. 1a).

While cells lacking Fmp27 grew poorly at low temperatures in synthetic complete (SC) media (Fig. 1d), growth was normal on yeast peptone dextrose (YPD; Extended Data Fig. 1b). This suggested that SC lacks the nutrients present in YPD that support the growth of *fmp27Δ* cells at low temperatures. Two nutrients present in YPD but absent from SC are ethanolamine (EtN) and choline (Cho). Exogenous EtN and Cho support the biosynthesis of PE and phosphatidylcholine (PC) by the Kennedy pathway (Fig. 1f). Yeast does not synthesize Cho and the only source of endogenous EtN is EtN-phosphate made by degrading dihydrosphingosine phosphate, catalysed by Dpl1 (ref. 17). When exogenous Cho and EtN are not present, PC and PE are synthesized from cytidine diphosphate diacylglycerol (CDP-DAG; Fig. 1f). We found that the growth defect of cells lacking Fmp27 in SC at 18 °C is corrected by supplementation of the medium with EtN but not Cho (Fig. 1g), suggesting *fmp27Δ* cells require EtN to grow at low temperatures, while growth at 30 °C was not affected by exogenous EtN (Extended Data Fig. 1c). Since SC is unbuffered and EtN is alkaline, we used buffered SC to rule out that the effect of EtN on growth is related to pH (Extended Data Fig. 1d).

As yeast only uses EtN to produce PE, *fmp27Δ* cells may grow poorly at low temperatures because they have insufficient PE. Consistent with this, these cells have reduced PE levels when grown at 18 °C (Fig. 1h). To obtain further evidence, we used strains lacking Dpl1 because Dpl1 produces low levels of EtN-phosphate that can be used to produce PE[17,18] (Fig. 1f). Cells lacking both Dpl1 and Fmp27 proliferated more poorly at 23 °C than cells lacking only Fmp27, and growth was restored by exogenous EtN but not Cho (Fig. 1i), suggesting the growth defect of *fmp27Δ* cells is caused by altered PE homeostasis. Moreover, growth of cells lacking Fmp27 at 23 °C was sensitive to the loss of the PE-biosynthetic enzymes Psd1 and Psd2 (Extended Data Fig. 1e,f), suggesting cells require Fmp27 to overcome cold stress in conditions when PE is limiting.

We ruled out that cells lacking Fmp27 have defects in producing PE via the CDP-DAG pathway (Fig. 1f). In this pathway, cells produce phosphatidylserine (PS) that is decarboxylated to form PE, which is methylated to generate PC. To estimate the flux through the CDP-DAG pathway in vivo, we labelled cells with L-[3-³H]serine at 18 °C for 60 min and determined the amount of radiolabel in PS, PE and PC. Labelling at this temperature is linear (Extended Data Fig. 1g). We found no significant difference (Fig. 1j and raw data in Extended Data Fig. 1h). We also ruled out that cells lacking Fmp27 have defects in producing PS, which would be expected to decrease rates of PE production via the CDP-DAG pathway. If this were correct, the growth defect of cells lacking Fmp27 would be corrected by overproducing PS synthase (Cho1), but this was not observed (Extended Data Fig. 2a).

Since cells lacking Fmp27 do not seem to have defects producing PE by the CDP-DAG pathway or using PE to produce PC, we wondered whether they have defects in other metabolic processes requiring PE. One is the production of glycosylphosphatidylinositol (GPI) anchors, which require three molecules of PE per GPI[19]. Cells with defects in GPI-anchor biosynthesis are hypersensitive to calcofluor white[20]. However, we found that cells lacking Fmp27 are not hypersensitive to calcofluor white (Extended Data Fig. 2b), suggesting they do not have a defect in GPI-anchor biosynthesis. This conclusion is further supported by our analysis of whether *fmp27Δ* cells are hypersensitive to tunicamycin, which induces ER stress. We previously observed that yeast cells lacking the Vps13 family protein Csf1 are hypersensitive to tunicamycin (which causes ER stress), but cells lacking Fmp27 are not (Extended Data Fig. 2c). We conclude that cells lacking Fmp27 have defects in PE homeostasis, but do not have defects in PE-requiring metabolism.

### Yeast lacking Fmp27 fail to maintain PM fluidity

We next considered whether the biophysical properties of PE are important for sustaining the growth of cells lacking Fmp27 at low temperatures. PE has a high negative monolayer spontaneous curvature, and some PE species form hexagonal phase structures in aqueous solution, properties thought to be necessary to sustain membrane dynamics and fluidity[21]. To test this, we grew cells with propanolamine (PpN; Fig. 1k). PpN is used to produce the phospholipid phosphatidylpropanolamine, which has the biophysical properties of PE but cannot be converted to PC[18]. Since exogenous PpN improves the growth of *fmp27Δ* cells at 23 °C (Fig. 1i), the biophysical properties of PE are critical for the growth of these cells at low temperature.

These properties are important for the role of PE in maintaining membrane fluidity, which is determined, in part, by the ratio of PE to bilayer-forming lipids such as PC (the most abundant lipid in most cellular membranes[22]). Cells also maintain membrane fluidity by modulating the saturation of acyl chains in membrane lipids, a process known as homeoviscous adaptation[23]. As a role for another BLTP protein in homeoviscous adaptation has been proposed[24,25], we wondered whether Fmp27 might regulate membrane fluidity at low temperatures by modulating the ratio of PE to PC. We first determined the PE:PC ratio for wild-type (WT) and *fmp27Δ* cells grown at 30 °C and 18 °C. When WT cells are grown in a medium without EtN, the PE:PC ratio increases significantly at 18 °C (Fig. 2a), which is probably a mechanism of homeoviscous adaptation. In contrast, the PE:PC ratio of cells lacking Fmp27 does not change when they are grown at 18 °C. The ratio is also

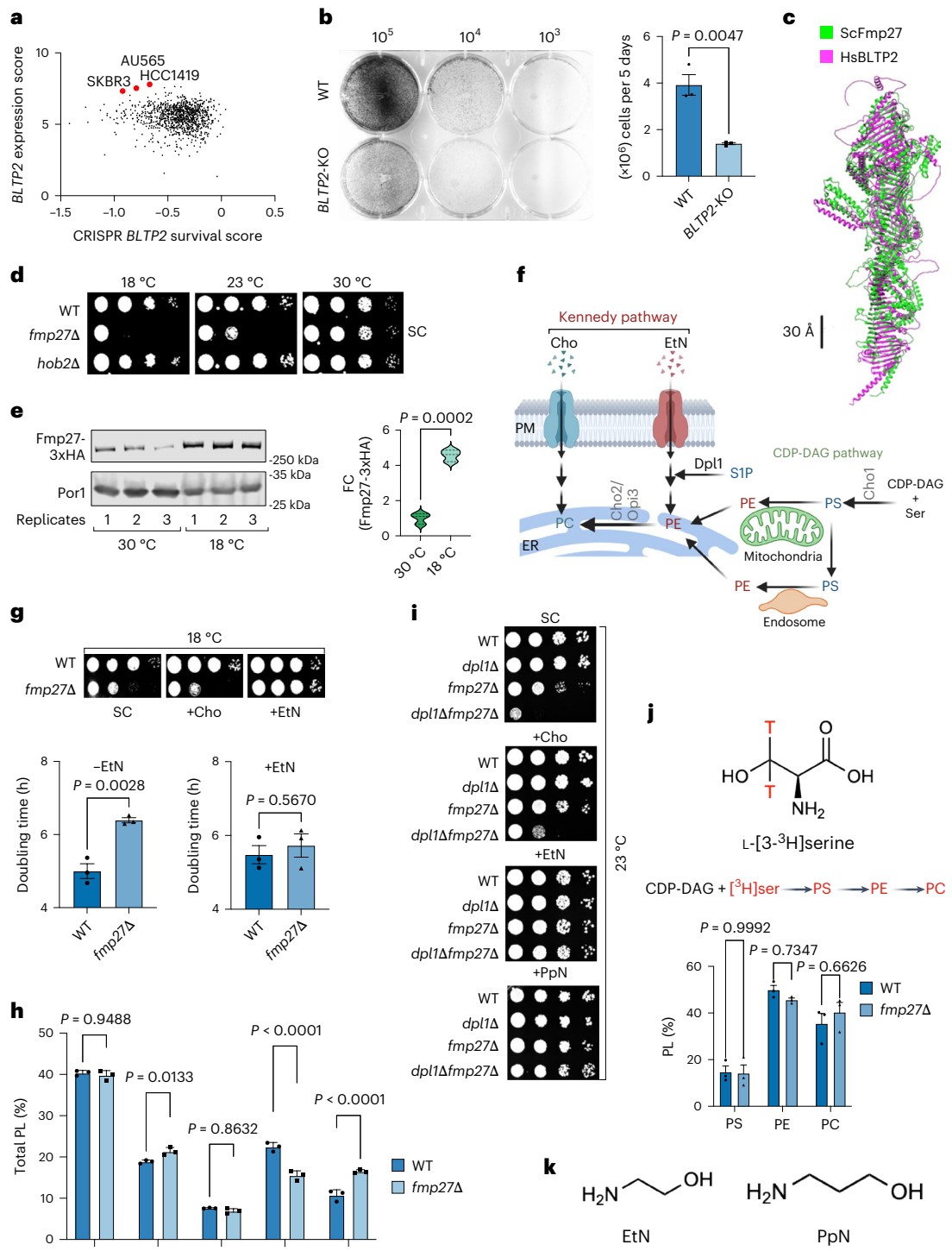

**Fig. 1 | PE rescues the cold sensitivity of yeast lacking the BLTP2 homologue Fmp27. a**, A plot of *BLTP2* mRNA expression score and survival score for ~1,000 cancer cell lines with a deletion of *BLTP2*. Data from DepMap. **b**, Crystal violet-stained colony forming units of HeLa cells plated at the cell number indicated on the top (left). Growth rate of WT and *BLTP2*-KO cells (*n* = 3) in DMEM with cFBS (right); mean cell number ± s.e.m.; *P* values from unpaired two-tailed *t*-tests. **c**, The AlphaFold predicted secondary structure of yeast Fmp27 (green) and human BLTP2 (magenta) compared and aligned using the Matchmaker plugin of UCSF Chimera. Scale bar, 30 Å. **d**, Serial dilutions of yeast strains spotted on SC without EtN supplementation and grown for 3 days. **e**, Western blot (left) of genomically expressed Fmp27-3xHA and Por1 (control). Cells were grown in SC and Fmp27-3xHA and Por1 were immunoprecipitated. Quantification of Fmp27–GFP normalized to Por1 (right). The results are shown as a box–violin plot (the solid bar shows the median and the dotted bar the quartiles); *P* values

from unpaired two-tailed *t*-tests, *n* = 3. **f**, A schematic of Kennedy and CDP-DAG pathways. Created with BioRender.com. **g**, Serial dilutions yeast strains were spotted on SC without or with 4 mM Cho or 4 mM EtN and grown at 18 °C for 3 days (top). Doubling time of strains grown at 18 °C, with or without EtN supplementation, in a 96-well plate; histograms show mean ± s.e.m. (*n* = 3); *P* values from unpaired two-tailed *t*-tests (bottom). **h**, The relative amount of the five major PLs in cells grown at 18 °C and labelled to steady-state with [³H] palmitate; mean ± s.e.m. (*n* = 3), *P* values from two-way ANOVA. **i**, Serial dilution of yeast strains spotted on SC medium without or with Cho, EtN or PpN and incubated at 23 °C for 3 days. **j**, The structure of L-[3-³H]serine; T indicates the position of tritium (top) and a scheme of PS, PE and PC production by exogenous [³H]serine (top). The bar graph shows the relative abundance of radiolabelled PLs after growth for 1 h at 18 °C in SC (bottom); mean ± s.e.m. (*n* = 3); *P* values from two-way ANOVA. **k**, The chemical structures of EtN and PpN.

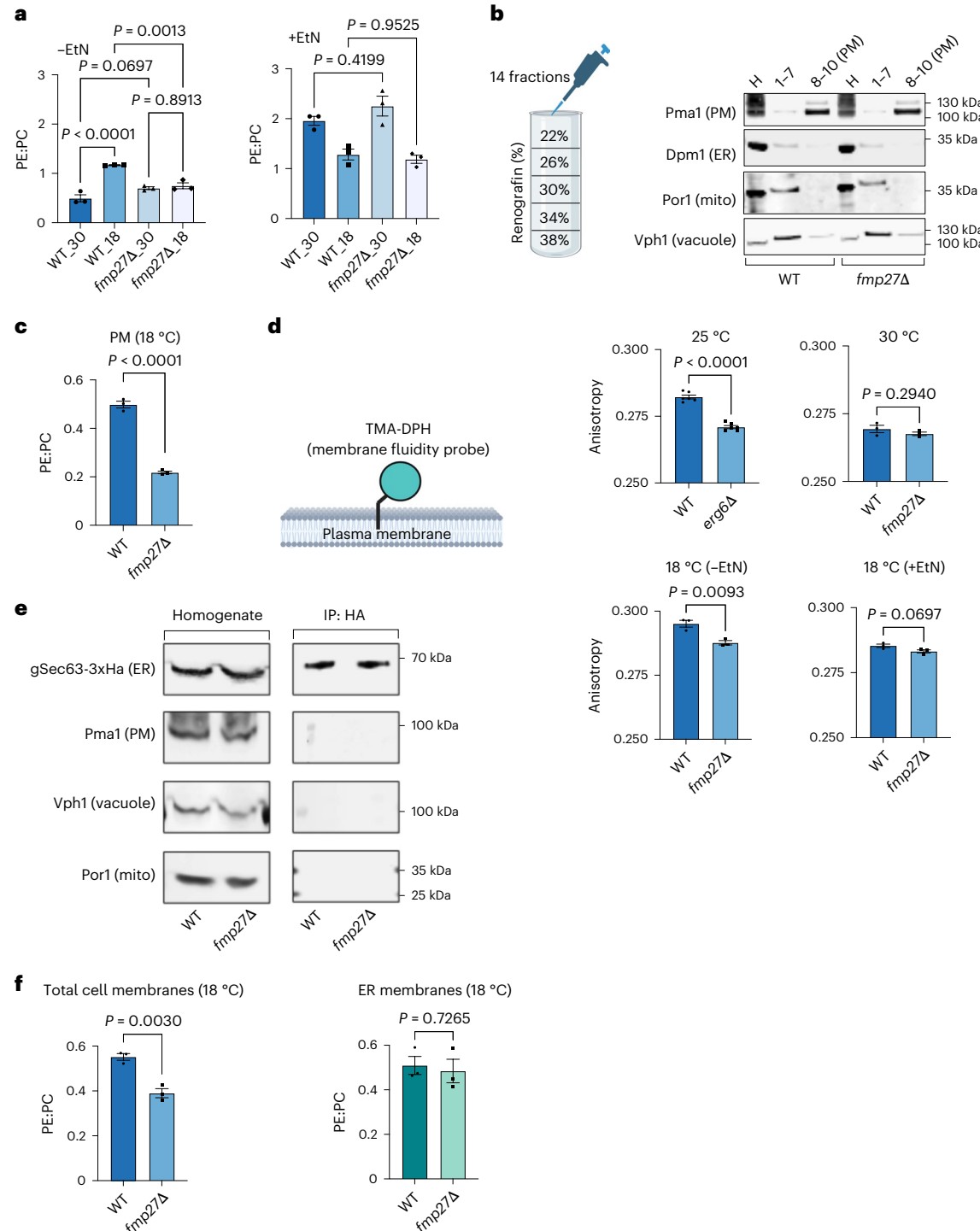

**Fig. 2 | Cells lacking Fmp27 have reduced PE in the PM and increased PM fluidity at low temperatures. a**, The PE:PC ratios from yeast cells grown at the indicated temperatures (18 °C or 30 °C) in SC medium with or without 4 mM EtN; mean ± s.e.m. (*n* = 3); *P* values from one-way ANOVA. PLs measured using ELSD. **b**, A scheme showing how renografin-76 density gradients were used to obtain PM-enriched fractions of cell lysates (left). Western blot of the indicated proteins from homogenate (H), pooled fractions 1–7 and pooled fractions fractions 8–10, which are enriched in PM. Representative blots from three experiments. Samples were loaded separately for the blots, one set for Pma1 and Dpm1, and another for Por1 and Vph1. **c**, The PE:PC ratios from PM-enriched fraction from strains grown at 18 °C in SC medium; mean ± s.e.m. (*n* = 3) of PE:PC; *P* values from unpaired two-tailed *t*-tests. PLs measured using ELSD. **d**, A cartoon of the fluidity probe

TMA-DPH (circle) in the outer leaflet of the PM (top). Created with BioRender. com. TMA-DPH anisotropy from yeast strains grown at 25 °C (*n* = 6) and 30 °C (*n* = 3) in SC medium and with (*n* = 3) or without (*n* = 3) EtN in SC medium at 18 °C (bottom); mean ± s.e.m.; *P* values from unpaired two-tailed *t*-tests. **e**, Western blots of immunopurified ER from yeast strains expressing Sec63-3xHA and grown at 18 °C in SC medium. Blots of whole cell homogenates (left) and after immunopurification of membranes using an anti-HA antibody (IP:HA; right). Representative blots from three experiments. **f**, The PE:PC ratios of total yeast cell membranes and ER membranes as in **e**; mean ± s.e.m. (*n* = 3; *P* values from unpaired two-tailed *t*-tests. Cells were labelled to steady state with [³H]palmitate to allow measurement of PLs.

significantly lower than WT at 18 °C (Fig. 2a, top). This suggests that *fmp27Δ* cells fail to alter PE:PC ratio at low temperatures when cells are grown without EtN. When EtN was added to the medium, there was no significant difference in the ratio of PE:PC in WT and *fmp27Δ* cells (Fig. 2a, bottom).

Since Fmp27 localizes to ER–PM contact sites, we determined whether it is particularly important for maintaining the ratio of PE:PC in the PM. Using density gradient centrifugation to isolate PM-enriched fractions[26] from cells grown at 18 °C (Fig. 2b), we found that the PE:PC ratio of the PM was ~2.5-fold higher in WT cells than in cells lacking Fmp27 (Fig. 2c). This suggests Fmp27 regulates PE levels in the PM to regulate fluidity. To directly assess PM fluidity, we used 1-(4-trimethylammoniumphenyl)-6-phenyl-1,3,5-hexatriene *p*-toluenesulfonate (TMA-DPH) to assess fluidity in the outer leaflet of the PM[27]. While PE is primarily localized to the inner leaflet of the PM[28], there is evidence that the composition of the inner leaflet of the PM can affect the fluidity of the outer leaflet[29,30]. Previous studies have shown that yeast cells lacking the sterol synthesis protein Erg6 (*erg6Δ*) have increased PM fluidity[27,31], a finding we confirmed (Fig. 2d, top left). We next found that the PM fluidity of *fmp27Δ* cells is indistinguishable from WT when cells are grown at 30 °C (Fig. 2d, top right). In contrast, when grown at 18 °C the PM fluidity of *fmp27Δ* cells was significantly higher than that of WT (Fig. 2d, bottom left). This difference is eliminated by adding EtN to the growth medium (Fig. 2d, bottom right). Together, these findings indicate that cells lacking Fmp27 fail to alter the ratio of PE:PC when grown at a low temperature and have increased PM fluidity. We speculated that Fmp27 transports PE from the ER to the PM, which maintains the PE:PC ratio of the PM at low temperatures.

Fmp27 does not seem to regulate the PE:PC ratio of the ER. We immunopurified ER-derived membranes[32] (Fig. 2e) and found no significant difference (Fig. 2f) between WT and *fmp27Δ* cells grown at 18 °C. Note that the PE:PC ratio for whole cell extracts in these experiments differs somewhat from that in Fig. 2a, probably because different methods were used to quantify phospholipids. Together, these results suggest that Fmp27 specifically regulates the PE:PC ratio of the PM but not the ER.

We wondered whether Fmp27 might also control membrane fluidity by altering the saturation of phospholipid acyl chains, perhaps by regulating acyl chain remodelling. To address this, we determined the relative abundance of glycerophospholipids species in WT and *fmp27Δ* cells grown at 30 °C and 18 °C, both in a media without (Fig. 3a, top) and with EtN (Fig. 3b, bottom). Volcano plots of the fold change (FC) of phospholipid species in cells lacking Fmp27 compared with WT show that all changes are either statistically insignificant or less than twofold. Therefore, no major differences were found. Calculation of an unsaturation index of glycerophospholipids revealed that while the index increases significantly when cells are shifted from 30 °C to 18 °C, there was no difference between WT and *fmp27Δ* cells (Fig. 3c). Interestingly, the unsaturation index was barely affected by growth temperature when EtN was in the medium (Fig. 3d), suggesting changes in acyl chain composition do not play a significant role in cold adaptation when PE is abundant. These findings suggest that Fmp27 does not regulate phospholipid acyl chain remodelling or production. This conclusion was also supported by a determination of whether the growth of *fmp27Δ* cells was affected by inositol, an important regulator of glycerolipid metabolism in yeast[33] (Extended Data Fig. 3a). We found that the relative growth rates of *fmp27Δ* and WT cells at 18 °C were the same regardless of the presence of inositol in growth media (Extended Data Fig. 3b,c). Together, these findings show that Fmp27 regulates PM fluidity at low temperature by maintaining PE:PC in the PM and not by altering the acyl chain composition of PE or other phospholipids.

## Fmp27 facilitates PE transport in cells
As our findings suggest that Fmp27 transports PE from the ER to the PM at ER–PM contact sites, we sought additional evidence that Fmp27 is a

PE transporter. If this is correct, enriching Fmp27 at ER–mitochondria contacts might increase the amount of PE in mitochondria (Fig. 3e). To move Fmp27 to ER–mitochondria contact sites, we used strains expressing Fmp27 endogenously tagged at the C-terminus with GFP and an anti-GFP nanobody (NB) fused to the first 30 amino acids of Tom70, a mitochondrial outer membrane protein (Fig. 3e). We confirmed that endogenously expressed Fmp27–GFP is functional (Extended Data Fig. 3d) and that Tom70(1–30)–anti-GFP NB causes Fmp27–GFP to become enriched at mitochondria (Extended Data Fig. 3e,f). Since the N-terminus of Fmp27 is anchored in the ER[15], NB probably enriches Fmp27–GFP at ER–mitochondria contacts. Fmp27–GFP and NB were expressed in cells lacking Psd1 (*psd1Δ*), an enzyme that produces PE in mitochondria. When grown in media without EtN, *psd1Δ* cells have low PE in mitochondria, which causes them to fragment[34], a finding we confirmed (Fig. 3f,g). We found that expressing Fmp27–GFP and NB significantly reduced mitochondrial fragmentation of *psd1Δ* cells (Fig. 3g), suggesting Fmp27–GFP at ER–mitochondria contact sites increases PE in mitochondria. To directly assess PE levels in mitochondria, we isolated mitochondrial membrane associated with the mitochondrial outer membrane protein OM45 using immunopurification (Fig. 3h) from cells labelled to steady state with [³H]palmitate. Like previous studies on *psd1Δ* cells[35], *psd1Δ* cells expressing Fmp27–GFP but not NB have a ~2-fold reduction in total PE and a 5-fold reduction in mitochondrial PE (Fig. 3i). Expression of both Fmp27–GFP and NB caused a significant increase in mitochondrial PE levels and no change in whole-cell PE levels (Fig. 3i). Together, these findings suggest that Fmp27 transports PE from the ER to other organelles at contact sites. What drives transport remains to be determined.

While Fmp27 probably moves PE to the PM in all growth conditions, PE must reach the PM by other mechanisms as well; the primary one may be vesicular trafficking. These mechanisms may require high levels of cellular PE to function optimally at low temperature since the need for Fmp27 to support growth at low temperatures is bypassed when the Kennedy pathway is active, increasing cellular PE levels.

## BLTP2 in *C. elegans* plays a role in PE homeostasis
We next determined whether BLTP2 has a similar function in the nematode *Caenorhabditis elegans*, which has one isoform of Fmp27 we named *bltp-2*. Worms with a deletion of *bltp-2* did not have growth or morphology defects (Extended Data Fig. 4a,b). While a previous study showed that *Drosophila* lacking BLTP2 (called Hob) had reduced pupal size[36], this was not found in the *C. elegans bltp-2* mutant (Extended Data Fig. 4a–c). However, the body scale of *bltp-2* worms was significantly affected by RNAi-mediated knockdown of *pcyt-1*, which produces phosphate cytidyltransferase 1, the rate-limiting enzyme in the Kennedy pathway[37] (Extended Data Fig. 4c–e). This treatment also reduced brood size (Extended Data Fig. 4f) and embryonic viability (Extended Data Fig. 4g). Together, these findings suggest that in *C. elegans*, as in yeast, BLTP2 is important for cell growth and development when PE production by the Kennedy pathway is limited.

## BLTP2 regulates PM fluidity in HeLa cells
We next investigated BLTP2 function in HeLa cells. The growth rate of HeLa BLTP2-KO cells in media with dialysed FBS (dFBS), which does not contain EtN or Cho[38], was improved by addition EtN but not Cho (Fig. 4a and Extended Data Fig. 5a). Notably, the addition of PpN also strongly improved growth, suggesting that the biophysical properties of PE are important for the growth of BLTP2-KO cells when they do not have sufficient exogenous EtN to support robust PE production by the Kennedy pathway. Interestingly, PpN significantly improves not only the growth of BLTP2-KO cells but also WT HeLa cells (Extended Data Fig. 5a), suggesting PE-like lipids benefit cells even in unstressed conditions. We measured PE and PC levels in cells grown in media with complete FBS (cFBS) (Fig. 4b) or dFBS (Fig. 4c) and found PE levels were significantly reduced in cells grown with dFBS. This was also

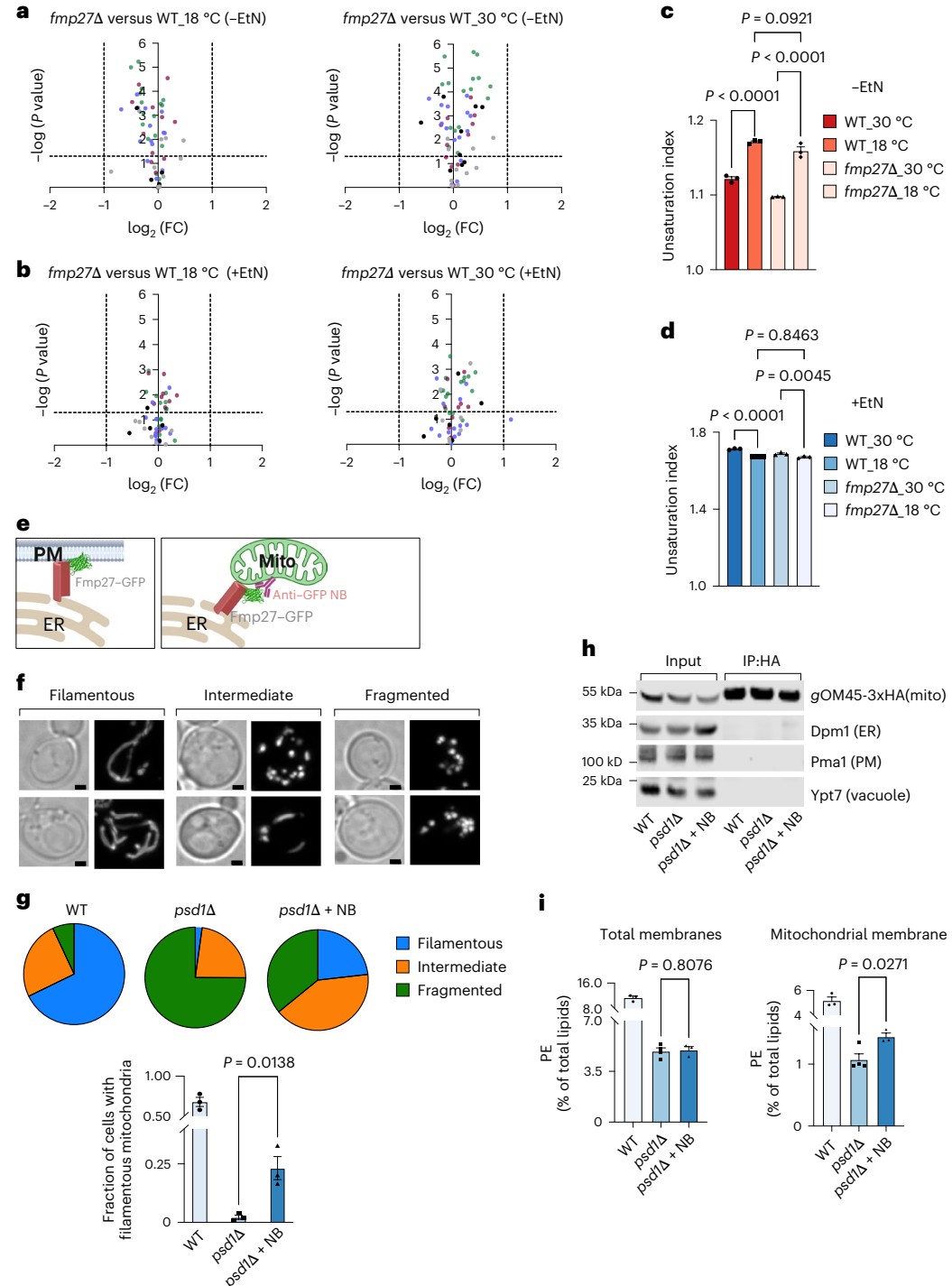

**Fig. 3 | Acyl chain remodelling is not altered in cells lacking Fmp27 and Fmp27 transports PE in vivo. a,b**, Volcano plots showing the $\log_2$ FC of PL species against statistical significance ($-\log P$ value) in yeast strains grown in SC medium without (**a**) or with (**b**) EtN. The dotted lines indicate the thresholds set at equivalents of a twofold change in the $x$ axis and a $P$ value of 0.05 in the $y$ axis. PL species in these PL types are highlighted; PC, green; PE, maroon; PI, blue; PS, black; and PA, grey. $P$ values are from unpaired two-tailed $t$-tests. **c,d**, Unsaturation indices of PLs from yeast grown in SC without (**c**) or with (**d**) EtN; mean ± s.e.m. ($n = 3$) values; $P$ values from one-way ANOVA. **e**, A cartoon showing the natural (left) and redirected (right) subcellular localization of Fmp27–GFP. **f**, Representative images from three experiments of types of mitochondrial shapes from yeast cells expressing the mitochondrial marker mitoDsRed: filamentous (mitochondria >2 μm long, left), intermediate (mitochondria filaments between 1 and 2 μm long, middle) or fragmented (no mitochondria filaments, right). Brightfield

images (left) and fluorescent images (right) are shown for each type; scale bars, 1 μm. **g**, Pie graphs of fraction of yeast cells with mitochondria of each shape type (top; as in **f**), and bar graphs for fraction of cells with filamentous mitochondria, when grown in SC medium at 30 °C (bottom); mean ± s.e.m. of 3 independent experiments and calculated from at least 100 cells per experiment. $P$ value from an unpaired two-tailed $t$-test. **h**, Western blots of immunopurified mitochondrial membranes from indicated yeast strains expressing OM45-3xHA grown in SC medium at 30 °C. Representative blots from three experiments of whole cell lysates (Input) and after immunopurification of mitochondrial membranes using an anti-HA antibody (IP:HA; right). **i**, PE as a percentage of total radiolabelled lipids from total cell membranes (left) and mitochondrial membranes (right) as in **h**; mean ± s.e.m. ($n = 4$); $P$ values from unpaired $t$-tests. Cells were labelled to steady state with [³H]palmitate to allow measurement of PLs.

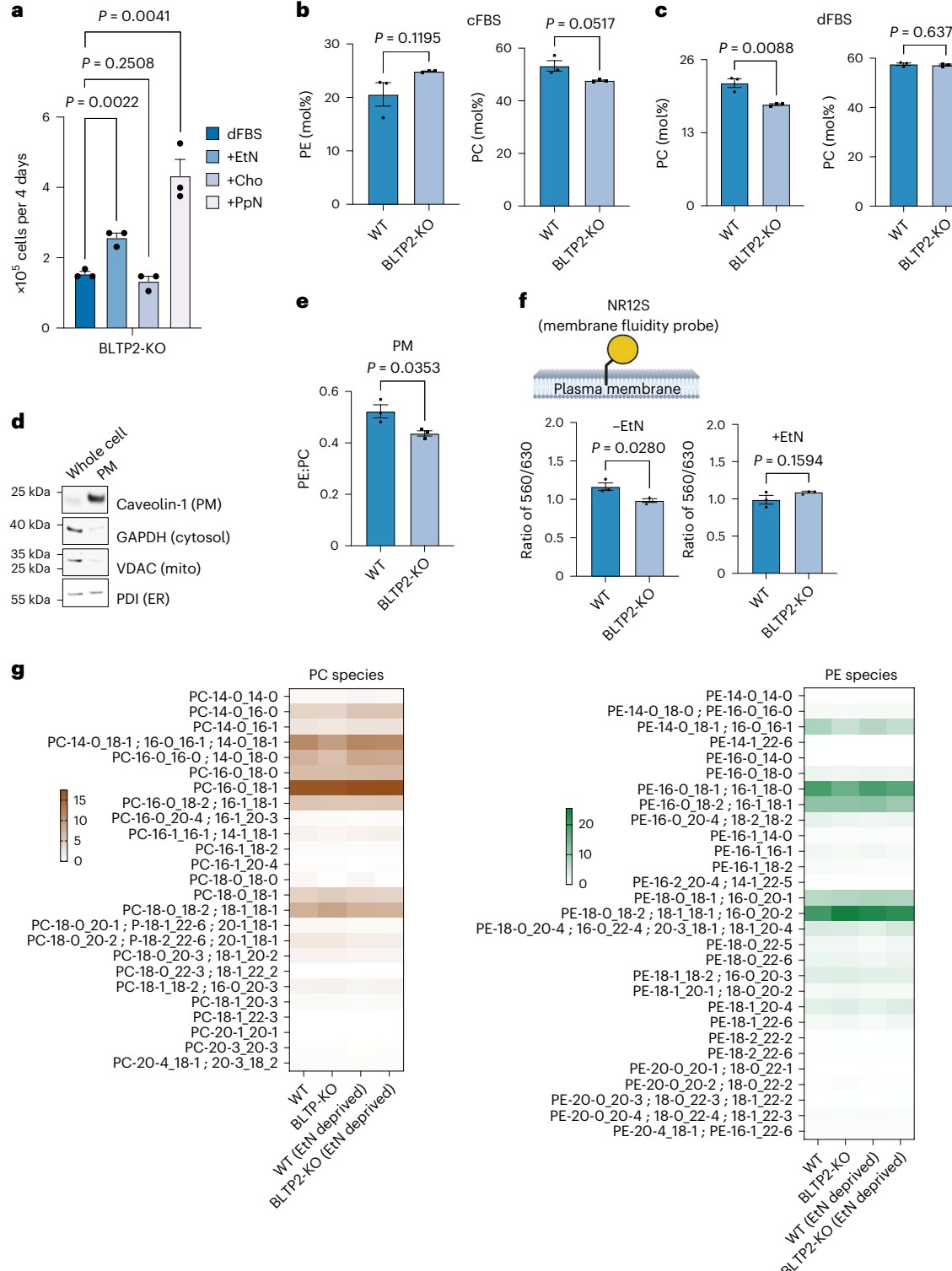

**Fig. 4 | BLTP2-KO HeLa cells have a decreased PM PE:PC ratio and increased PM fluidity. a**, An automated cell count assay for BLTP2-KO HeLa cells grown with DMEM with dFBS and without or with 10 μM EtN, Cho or PpN; mean ± s.e.m. (*n* = 3) values; *P* values are from unpaired two-tailed *t*-tests. **b**,**c**, The mol% of PLs from HeLa cells grown in DMEM with cFBS (**b**) or dFBS (**c**) determined using LC–MS/MS; mean ± s.e.m. (*n* = 3); *P* values from unpaired two-tailed *t*-tests. **d**, Western blots of whole cell lysates and PM-enriched fractions (PM) of HeLa cells grown in DMEM with dFBS. Blots are representatives from three experiments. **e**, PE:PC ratios of PM-enriched fraction from HeLa cells in d; mean ± s.e.m. (*n* = 3); *P* values from unpaired two-tailed *t*-tests. PL levels were determined by ELSD. **f**, A cartoon of the fluidity probe NR12S (circle) in the outer leaflet of the PM (top). Created with BioRender.com. The ratio of NR12S fluorescence at 560 nm and 630 nm of HeLa strains grown in DMEM with dFBS without (*n* = 3) or with (*n* = 3) 100 μM EtN supplementation (bottom); mean ± s.e.m.; *P* values from unpaired two-tailed *t*-tests. **g**, The relative abundance of PE and PC species from HeLa cells grown in DMEM with cFBS or dFBS and shown as a heat map (0, lowest to 15, highest abundance relative to total of a lipid class; mean of 3 independent replicates). The colour gradient keys are in mol% of PC and PE.

found when we measured the ratio of PE to PC in the PM. We used an established method to obtain a PM-enriched fraction from attached cells[39] (Extended Data Fig. 5b), confirmed the fraction was PM enriched (Fig. 4d) and found the PE:PC ratio was significantly reduced in PM from BLTP2-KO cells (Fig. 4e).

These findings prompted us to determine whether PM fluidity is reduced in BLTP2-KO cells as it is in yeast cells lacking Fmp27. We used the order-sensitive dye NR12S to measure fluidity in the outer leaflet of the PM[40]. We first confirmed we could measure membrane fluidity with this dye by showing that treating cells with methyl beta-cyclodextrin, which extracts cholesterol from cells, increases PM fluidity as previously shown[40] (Extended Data Fig. 5c). We found that BLTP2-KO cells have increased PM fluidity when EtN is not in the medium (Fig. 4f). We speculated that human BLTP2, like its yeast homologue, controls PM fluidity primarily by increasing the ratio of PE:PC in the PM, but not by affecting phospholipid acyl chain saturation. Consistent with this, we found only minor changes in the relative abundance of the major phospholipid species in BLTP2-KO cells (Fig. 4g and Extended Data Figs. 5f and 6a,b). Since cholesterol is a major determinant of membrane fluidity[41], we tested whether BLTP2-KO cells have altered levels of cholesterol in the PM but found no difference (Extended Data Fig. 5h).

Together, our findings suggest that BLTP2 in humans, like its yeast homologue, regulates PM fluidity by increasing the amount of PE in the PM when PE synthesis by the Kennedy pathway is compromised by a lack of exogenous EtN. We speculate that BLTP2 transports PE from the ER to the PM. Consistent with this, we found that overexpressed BLTP2 localizes to ER–PM contact sites (Extended Data Fig. 6c) like its homologues in yeast[16] and *Drosophila*[15].

### BLTP2 and TLCD1 independently regulate PM fluidity

We wondered whether BLTP2 in human cells operates together with a previously described pathway that regulates PM fluidity and PE metabolism. This pathway requires the PM protein Tram-Lag-CLN8 (TLC)-domain containing protein 1 (TLCD1)[42,43]. How TLCD1 functions is not well understood. We identified that TLCD1 is one of the top hits in a list of genes that are co-dependent with BLTP2 (refs. 13,14) (Cancer Dependency Map Portal (RRID:SCR_017655); Extended Data Fig. 7a). To confirm this, we treated HeLa cells in which TLCD1 has been knocked out (TLCD1-KO) with short hairpin (sh)RNAs targeting BLTP2 (shBLTP2) or a control, untargeted shRNA (shUT). We found that elimination of TLCD1 and BLTP2 has an additive effect on cell proliferation (Fig. 5a and Extended Data Fig. 7b) and maintaining PM fluidity (Fig. 5b). These findings confirm that BLTP2 plays a significant role in maintaining PM fluidity in human cells and indicate that some cell types have two independent pathways of regulating PM fluidity, both of which have PE-dependent mechanisms.

### BLTP2 knockdown inhibits TNBC cell migration in xenografts

We assessed the role of BLTP2 in the survival of 46 invasive breast cancer cell lines from data available in DepMap (Extended Data Fig. 8a). We compared the median survival score of BLTP2 deletion with the deletion mutants of ERBB2/HER2, a known driver of breast cancer growth[44], LAMTOR1, a gene critical for mTORC1-mediated growth signalling downstream of ERBB2/HER2, and VPS13C, encoding a homologue of BLTP2. We observed that the median survival score of BLTP2 deletion mutants is lower than deletion mutants of the other genes, suggesting that BLTP2 supports the growth of many invasive breast cancer cell lines. To further understand the function of BLTP2 in human breast cancer cells, we investigated its reported role in the aggressiveness of the TNBC cell line MDA-MB-231 (ref. 12). BLTP2 expression is threefold higher in these cells than in the non-cancerous breast epithelial cell line MCF10A (Fig. 5c). Acute depletion of BLTP2 (shBLTP2; Extended Data Fig. 8a) significantly reduced the proliferation rate of MDA-MB-231 cells but not control MCF10A cells (Fig. 5d,e). In addition, the depletion of BLTP2 significantly increased MDA-MB-231 cell apoptosis (Fig. 5f and Extended

Data Fig. 9), indicating BLTP2 is a pro-survival protein in TNBC cells. We next tested whether BLTP2 controls the invasiveness of MDA-MB-231 cells in a previously established zebrafish xenograft model[45,46]. MDA-MB-231 cells stably expressing the actin filament reporter *Ftractin-EGFP* and grown in a medium with FBS were treated with shBLTP2 or shUT and xenografted into larval zebrafish (Extended Data Fig. 8b). This allowed a quantitative assessment of how these two cell types survive and disseminate in situ. Quantifying the survival of micro-metastasized MDA-MB-231 cells in the zebrafish tail, away from the injection site, revealed their survival was significantly reduced after knockdown of BTLP2, particularly at 24 and 48 h post injection (hpi) (Fig. 5g). BLTP2 depletion also significantly affects dissemination; quantitative dissemination maps reveal *shBTLP2* MDA-MB-231 cells have a significantly reduced ability to spread throughout the zebrafish larva (Fig. 5h). Cell migration to the intersegmental vessels or the brain tissue was strongly impaired in BLTP2-depleted MDA-MB-231 cells (Fig. 5h). However, *shBTLP2* cells still spread to the caudal haematopoietic tissue and the gills, perhaps because of passive migration driven by blood flow. Interestingly, migration of BLTP2-depleted MDA-MB-231 cells into upper intersegmental veins from the caudal haematopoietic tissue is significantly impaired (Extended Data Fig. 8c). These findings demonstrate that BLTP2 depletion not only hinders TNBC cell proliferation, but also significantly impacts its migration in the zebrafish xenograft model.

Further evidence was provided by assessing actin ruffling at cell protrusions, a hallmark of cancer cell migration and invasiveness[47]. Imaging the xenografted MDA-MB-231 cells on a high-resolution axially swept light-sheet microscope revealed *shUT*-treated cells formed abundant actin ruffles, but ruffling was reduced in *shBTLP2* cells (Extended Data Fig. 8d)[48]. We propose that depletion of BLTP2 affects PM fluidity, which alters actin polymerization at the cell surface and reduces migration and invasiveness.

## Discussion

Our findings show that BLTP2 is part of a conserved pathway that regulates PM fluidity by increasing the amount of PE in the PM. BLTP2 probably directly transports PE from the ER to the PM. What drives the movement of PE to the PM and whether BLTP2 is specific for PE remain to be determined. However, it should be noted that what determines transport directionality and substrate specificity is not well understood for any member of the BLTP family[3]. It is likely that both the driving force for transport and the selection of lipids to be transported is provided by still unknown proteins the ER and PM that work together with BLTP2. Another important question is whether BLTP2 has acyl chain specificity. While Fmp27 knock out has little effect on the acyl chain composition of phospholipids in yeast, knock out of BLTP2 in HeLa cause some modest changes. It is possible that in some cell types or in some growth conditions, BLTP2 could affect the acyl chain composition of PE.

This study suggests BLTP2 and TLCD1 regulate PM fluidity independently. How changes in PM fluidity are sensed and what regulates the BLTP2- and TLCD1-dependent pathways remain to be determined. Since TLCD1 has been implicated in PE homeostasis[42,43], it may be that the primary of function of both BLTP2 and TLCD1 is regulating PE distribution or remodelling. More work is necessary to understand how both proteins affect PE homeostasis.

Why BLTP2 is particularly important for the proliferation of several breast cancer cell lines and the aggressiveness of MDA-MB-231 is an open question. Our results suggest BLTP2 is pro-tumoural and regulates invasiveness, presumably by controlling the PM fluidity of some cancer cells. This cell line could have altered PE or EtN metabolism, rendering it particularly dependent on BLTP2 to maintain PM fluidity. Alternatively, BLTP2 could affect some other aspect of phospholipid homeostasis in MDA-MB-231 and some other breast cancer cell lines. Whether BLTP2 controls tumorigenesis in a PE and PM fluidity-dependent manner and whether BLTP2 is a targetable protein in aggressive breast cancer treatment will be determined in future studies.

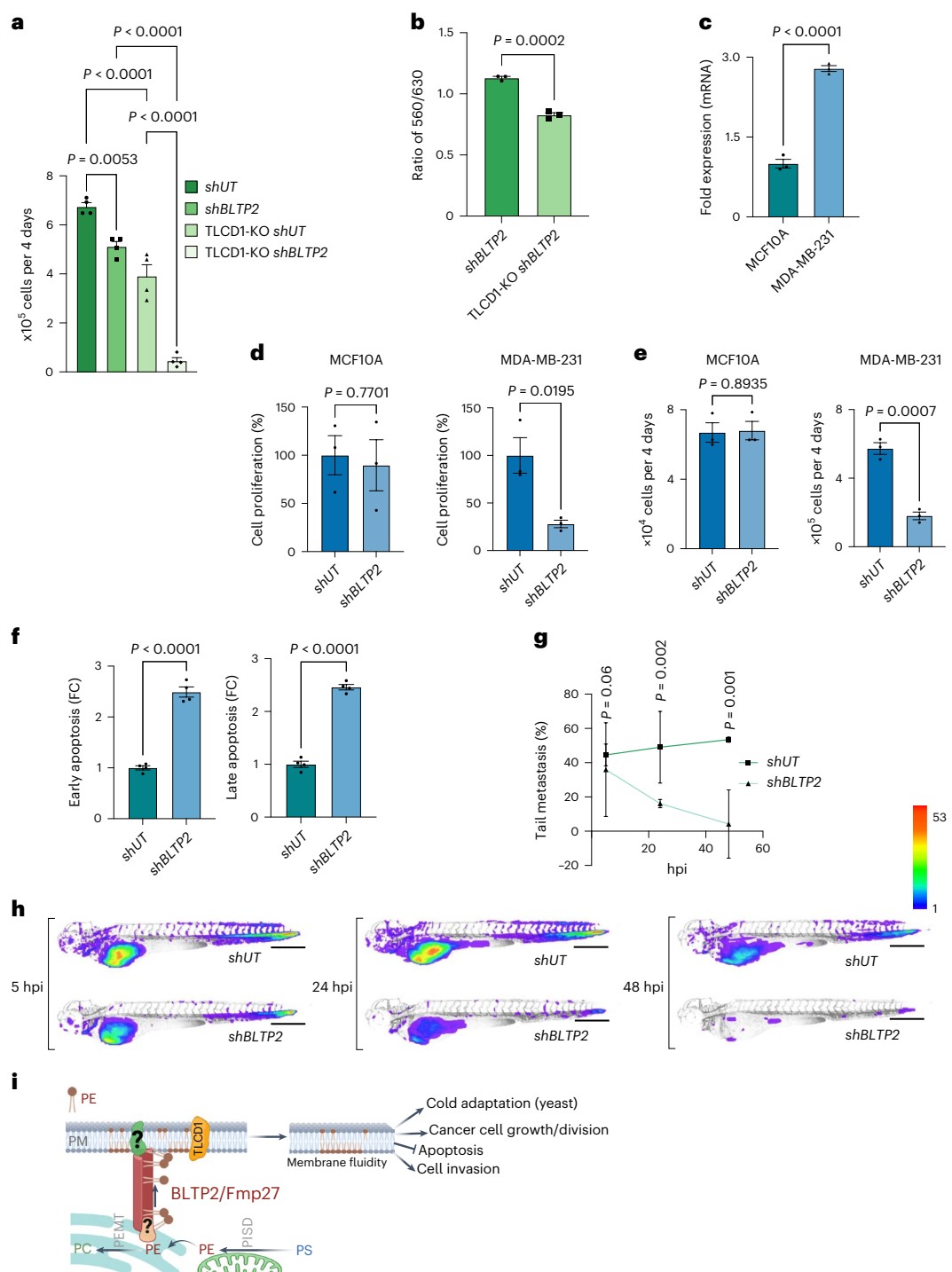

**Fig. 5 | BLTP2 regulates the survival and metastasis of MDA-MB-231 cells and synergizes with TLCD1 to control cell growth and PM fluidity. a**, Automated cell counting of HeLa strains grown with DMEM containing cFBS; mean ± s.e.m. ($n = 4$); $P$ values from one-way ANOVA. **b**, The ratio of NR12S fluorescence at 560 nm and 630 nm of HeLa strains grown in DMEM medium with cFBS and then shifted to DMEM with dFBS for 4 h; mean ± s.e.m. ($n = 3$); $P$ values from unpaired two-tailed $t$-tests. **c**, qRT–PCR showing the relative abundance of BLTP2 mRNA in MDA-MB-231 cells compared with MCF10A (non-cancerous) cells; mean ± s.e.m. ($n = 3$); $P$ values calculated from unpaired two-tailed $t$-tests are indicated. **d,e**, WST-1 assay (**d**) and automated cell count assay (**e**) comparing proliferation of BLTP2-silenced (*shBLTP2*) and control (*shUT*) MCF10A cells (left) and MDA-MB-231 cells (right); mean ± s.e.m. ($n = 3$); $P$ values from unpaired two-tailed $t$-tests. **f**, Fold difference in the percentage of cells undergoing early (left) and late (right) stages of apoptosis. Apoptotic cells were determined by annexin-V and propidium iodide dual staining and flow cytometry; mean ± s.e.m.

($n = 4$); $P$ values calculated from unpaired two-tailed $t$-tests. **g**, The per cent of xenografted larval zebrafish with tail-metastasized MDA-MB-231 expressing *F-tractin* and treated with *shUT* (control) or *shBLTP2* (BLTP2 depleted). Fish were imaged at 5, 24 and 48 hpi of MDA-MB-231 cells in the yolk. Points indicate the mean percentage of tail-metastasized fish from two independent experiments with shBLTP2 cells ($n_1 = 74$ and $n_2 = 63$ larvae) and shUT cells ($n_1 = 75$ and $n_2 = 80$ larvae); error bars indicate a 95% confidence interval; $P$ values for untailed two-tailed $t$-tests for each timepoint. **h**, Quantitative dissemination maps of zebrafish xenografts of MDA-MB-231 expressing *F-tractin* and treated with *shUT* (control) or *shBLTP2* (BLTP2 depleted) using a colour-coded scale bar of 1–53 for the number of cells at each location. Scale bars, 500 μm. **i**, Model for the cellular function of BLTP2/Fmp27. The proteins move PE to the PM to regulate PM fluidity and, in humans, work in parallel with the TLCD1 PM fluidity-regulating pathway; PISD, phosphatidylserine decarboxylase; PEMT, phosphatidylethanolamine N-methyltransferase. Created with BioRender.com.

## Online content

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

## Methods

### Yeast strains and culture

Yeast strains used in the study are listed in Supplementary Table 1. The oligonucleotides used in constructing and verifying yeast strains are listed in Supplementary Table 2. Yeast was grown in either SC medium or YPD medium prepared as described[16]. Where indicated, 4 mM EtN, Cho and PpN were added to the SC medium. Inositol-depleted medium was made using Yeast Nitrogen Base without inositol (US Biological Life Sciences) and supplemented with 0.1 M inositol where indicated.

For serial dilution-based growth assays, yeast cultures were grown to a mid-logarithmic growth phase and spotted on plates as described[16].

The strain expressing NB was constructed by using an integrative cassette plasmid *Ycplac111-LYS2-RTN1pr-TOM70(1-30)-nanobodyant iGFP-CYC1ter-CaURA-LYS2*. This cassette was integrated into the *LYS2* gene locus of *psd1Δ* FMP27–GFP.

### Mammalian cell culture

HEK293T (Takara, 632180), HeLa (American Type Culture Collection (ATCC), CRM-CCL-2) and MDA-MB-231; *F-Tractin-EGFP* cells were grown on DMEM (Gibco, 11965-092) supplemented with 10% FBS, penicillin and streptomycin at 37 °C with 5% $CO_2$. MDA-MB-231(CRM-HTB-26) cells were grown in Leibovit's L-15 medium supplemented with 10% FBS, penicillin and streptomycin at 37 °C without $CO_2$. MCF10A (ATCC, CRL-10317) cells were grown in mammary epithelial cell growth medium (Cell Applications, 815-500) at 37 °C with 5% $CO_2$. U2OS cells were grown in McCoy's 5A Medium (Gibco, 16600-082) with 10% FBS, penicillin and streptomycin at 37 °C with 5% $CO_2$. For experiments without EtN supplementation, DMEM with 25 mM HEPES (Gibco, 12430-054) supplemented with 10% dFBS (Gibco, A33820-01) was used and 100 μM of EtN, Cho and PpN was added to the medium. All cell lines were verified to be free of mycoplasma using the kit from the ATCC.

### Imaging of yeast and mammalian cells

Yeast cells were imaged live at the mid-logarithmic growth phase on a GE Deltavision Elite microscope. Images were processed using FiJi and brightness and contrast were set equivalently between yeast grown at different temperatures. Images for separate channels were then saved as TIF files, which were cropped using Adobe Photoshop.

To calculate the median fluorescence intensity with interquartile range of Fmp27–GFP within 200 nm of mitochondria (shown in Extended Data Fig. 3f), image analysis was carried out in Imaris (CF package, v.10.2.0, Oxford Instruments). In brief, individual channels were slightly smoothed with a Gaussian filter (Sigma: 0.065), followed by deconvolution. Next, within the 488 nm channel, 'spot detection' was used to determine the number of cells, as well as the location of green puncta. 'Surface detection' was applied to the 561 nm channel to determine mitochondrial morphology. Next, cells were classified as 'positive' or 'negative' based on the maximum intensity in the red channel. Finally, using the integrated MATLAB tools, green puncta were binned in either 'close' or 'far' from mitochondria, using a 200 nm threshold. The resulting quantification was exported from Imaris and plotted with GraphPad Prism.

To determine mitochondrial shape (to generate the results shown in Fig. 3g), maximum intensity projections were generated from stacks of images with 0.3 μm *z* steps. Cells containing at least one mitochondrion between 1 and 2 μm long were classified as intermediate, cells containing at least one mitochondrion more than 2 μm long were classified as filamentous and the rest were considered fragmented. Cells in each category were counted using the Cell Counter plugin of Fiji. The green signal from FMP27–GFP is mostly at the cell periphery and was used to visualize the cell boundary. More than 100 cells per strain in triplicate from three independent experiments were counted.

U2OS cells were imaged on the GE Deltavision system described above. The cells were transfected with shRNA targeting endogenous BLTP2 (Sigma, TRCN0000128410), which targets only the 3'UTR of

the BLTP2 mRNA. These cells were then transfected with pcDNA3.1(+) BLTP2–EGFP. They were also co-transfected with a plasmid containing mCherry-MAPPER, which was a kind gift from Dr. Jen Liou at UT Southwestern Medical Center[49]. The cells were imaged 48 h after transfection in four-well Labtech chambers. Images were processed and merged using Fiji and cropped using Adobe Photoshop 2024.

### Structure prediction and alignment

We first used Deepmind AlphaFold to predict the structure of human BLTP2 (Protein KIAA0100; Uniprot Q14667) and yeast Fmp27 (Protein FMP27; Uniprot Q06179). Then, we used UCSF Chimera to perform a structural alignment of HsBLTP2 and ScFmp27 using the Matchmaker plugin.

### BLTP2 score analysis

Publicly available data from the DepMap portal of the Broad Institute were used. We plotted the effect of BLTP2-CRISPR deletion (KIAA0100 Gene Effect (Chronos) CRISPR (DepMap Public 23Q2+Score, Chronos)) against the expression level of BLTP2 (KIAA0100 $\log_2$(transcripts per million + 1) Expression Public 23Q2) available from 1,020 cancer cell lines. We highlighted three metastatic breast cancer cell lines with red dots.

### Radioactive labelling of yeast PLs and quantitative measurements

To label phospholipids (PLs) to a steady state, 50 ml mid-logarithmic growth phase cultures were grown for five doublings in SC medium at 18 °C with 250 μCi of [$^3$H]palmitate (American Radiolabeled Chemicals, ART0129A). For pulse labelling with L[3-$^3$H]serine, 50 ml of cultures were grown in SC medium and were labelled with 100 μCi of L[3-$^3$H]serine (American Radiolabeled Chemicals, ART0246) at 18 °C for 1 h. Lipids were isolated, for all lipid analysis, using the following method. Lipids from samples were isolated from the $CHCl_3$ phase using $CH_3OH$ (Sigma, 646377):$CHCl_3$ (Sigma, 650498):$H_2O$ at 2:1:0.8 ratio v/v and two salt washes using 0.15 N NaCl and 2 M KCl. A lipid film was created by evaporating the $CHCl_3$ under compressed $N_2$ gas (Airgas). The lipid film was resuspended in 50 μl of $CHCl_3$ separated by thin-layer chromatography, and quantified using a radioactive thin-layer chromatography (TLC) plate reader (Raytest Rita Star) using previously established methods[16,50,51].

### Chemical structure elucidation

The chemical structures were drawn using ChemDraw confirmed by PubChem. The following settings were used: fixed length: 0.381 cm; bond spacing: 18% of length; line width: 0.021 cm; bold width: 0.055 cm; margin width: 0.042 cm; hash spacing: 0.06 cm; chain angle: 120° bond spacing.

### Lipid measurement using HPLC and an ELSD

PE and PC were separated with an Agilent 1260 Infinity II LC system and Agilent evaporative light scattering detector (ELSD). Lipids were extracted from yeast or mammalian cells as previously described[16]. The extracted lipids were dried under $N_2$, resuspended in 50 μl chloroform and 10 μl were injected on a Zorbax CN 4.6 × 150 mm column at 50 °C. The samples were run for 15 min each at a column pressure of 50 psi. The mobile phase was composed of solvent A and B at a 7:3 ratio. Solvent A was Hexane and solvent B was a w/w mixture of 60% toluene, 40% methanol, 0.2% acetic acid and 0.1% triethylamine. The ELSD detector used $N_2$ at a rate of 1.6 ml min$^{-1}$. The evaporator and nebulizer temperatures for the ELSD were set at 80 °C and 50 °C, respectively. Integrated peak values for PE and PC were obtained from Agilent OpenLAB, and the amounts of the lipids were calculated from standard curves using pure lipids from Avanti Polar Lipids.

For PM cholesterol measurements, WT and BLTP2-KO HeLa cells were grown in DMEM with cFBS in poly L-lysine coated plates, washed two times with Dulbecco's phosphate-buffered saline (DPBS; Gibco, 14190-144) and switched to DMEM without serum before PM isolation to avoid cholesterol contamination from the serum[39].

Lipids were extracted and separated using an Agilent 1260 Infinity II high-performance liquid chromatography (HPLC) system with an Agilent Zorbax CN 5 μm 4.6 × 150 mm column. The mobile phase was initially 97% hexane and 3% tertiary butyl methyl ether (TBME). Samples were run at 1 ml min⁻¹ for 3 min and then TBME was increased from 3% to 20% over 7 min and then run for 5 min at 20% TBME. The column was washed for 5 min with 97% hexane and 3% TBME in between samples. Cholesterol was detected using an Agilent ELSD; the evaporator was set at 90 °C, the nebulizer at 50 °C and $N_2$ was 2 ml min⁻¹. The amount of cholesterol was calculated from a standard curve generated using cholesterol from Avanti Polar lipids (SKU: 700000 P). The amount of cholesterol was normalized to the total PM protein.

### Rapid immunopurification of ER membranes from yeast

Strains expressing endogenously tagged Sec63-3xHA were grown to $OD_{600nm}$ of 1 and 25 $OD_{600nm}$ units cells were mixed with 10 mM sodium fluoride and 10 mM sodium azide. Cells were pelleted by spinning in a swinging bucket rotor at 1,900g for 5 min at 4 °C. Cells were washed in 0.5 ml KPBS buffer (136 mM KCl and 10 mM KPO4, pH 7.25) by resuspension in this buffer after being pelleted at 3,381g for 3 min at 4 °C. They were resuspended in 300 μl lysis buffer (KPBS containing protease inhibitor cocktail from Roche, 11836170001, and 1 mM PMSF), 300 μl acid-washed glass beads were added (Sigma-Aldrich, G8772) and lysed with a Precellys 24 homogenizer (Bertin Technologies) with 30 s homogenization cycles for 3 min and 1 min gaps to prevent overheating. Cell debris was removed by spinning at 400g for 5 min at 4 °C. The supernatant was separated and kept in a fresh tube. Then, 600 μl of lysis buffer was added to the glass beads, vortexed, spun down and 550 μl supernatant was obtained and mixed with the previous supernatant, and 100 μl of this supernatant was kept for a cellular lipid measurement. The remaining supernatant was added to 100 μl of anti-HA antibody magnetic beads (Pierce, 88837) that were prewashed in lysis buffer. After rocking at 4 °C for 15 min, the beads were washed three times in 1 ml cold lysis buffer for 5 min each at 4 °C. For western blotting, the beads were resuspended in 4× LDS sample buffer (Invitrogen, NP0007) containing 2% of 2-mercaptoethanol (Thermo Scientific, 125470100) and heated at 70 °C for 10 min before freezing them at −80 °C. For lipid extraction, beads were resuspended in 1.6 ml de-ionized water, 4 ml methanol and 2 ml CHCl₃. Lipid extraction and radiolabelled lipid measurements were conducted using a previously established method[50].

### Isolation and lipid analysis of PM

For yeast, an established method[26] was used. Briefly, 25 OD units of yeast cells in the mid-logarithmic growth phase were mixed with 10 mM sodium fluoride and 10 mM sodium azide. Cells were pelleted by spinning in a swinging bucket rotor at 1,900g for 5 min at 4 °C and washed in 0.5 ml breaking buffer (50 mM Tris pH 7.5, 1 mM EDTA, 1 mM PMSF and protease inhibitor cocktail). They were lysed using 300 μl glass beads in 300 μl breaking buffer. Cell lysis was conducted by hand vortexing at full speed using a Daiger Vortex Genie2 (3030A) for 7 times and 45 s each. Cell debris was removed by spinning at 400g for 5 min at 4 °C. The supernatant was collected. Then, 300 μl breaking buffer was added, vortexed briefly and spun down, and the supernatant was collected and pooled. Next, 500 μl of supernatant was mixed well with 500 μl 76% renografin and 800 μl of this mix was overlayed with 34%, 30%, 26% and 22% of renografin in breaking buffer and centrifuged at 193,911.2g using a Beckman SW55Ti swinging bucket rotor in a Beckman ultracentrifuge. After centrifugation, 14 fractions were collected from the top (Fig. 2b). For western blot to examine organelle membrane fractions, samples were prepared in LDS sample buffer (Invitrogen, LP007) containing 2% β-mercaptoethanol (Thermo Scientific cat# 125470100), and heated at 70 °C for 10 mins. Samples from fractions 1–7 and 8–10 were pooled, the volumes adjusted so they were equal, and 5 μl of the pools were compared by Western blotting. For lipid extraction, membrane fractions were resuspended in 1.6 ml de-ionized Milli-Q grade water (Millipore

Sigma, Milli-Q EQ 7000), 4 ml methanol (Sigma-Aldrich, 646377) and 2 ml chloroform (Sigma-Aldrich 650498). Lipid extractions were conducted as mentioned previously. PE and PC lipids were separated and measured using HPLC–ELSD as previously described.

PM from HeLa cells was isolated following a 'rip off' method as previously described[39]. Briefly, cells were grown in poly L-lysine coated plates and switched to DMEM with dFBS once they were attached. After cells grew to 80–90% confluency, they were washed with DPBS and switched to DMEM without serum before PM isolation to avoid lipid contamination from the serum. Cells were washed three times with DPBS to remove debris and unbound cells. Then, 10 ml of ice-cold tissue culture grade distilled water (Gibco, 15230-147) was used for cell lysis. The attached membranes were washed twice with DPBS to remove debris. Lysis and washes were conducted two more times. The purified PM fraction was resuspended in 500 μl Tris buffer (50 mM Tris pH 7.4 and 150 mM Nacl) with protease inhibitor tablets (Roche) and 1 mM PMSF for protein measurement. For lipid extraction, this resuspended PM fraction was collected in 1.1 ml de-ionized Milli-Q grade water (Millipore Sigma, Milli-Q EQ 7000), 4 ml methanol (Sigma-Aldrich, 646377) and 2 ml chloroform (Sigma-Aldrich, 650498). To conduct a western blot on proteins from the PM fraction, PM fractions were collected in ice-cold radioimmunoprecipitation assay (RIPA) buffer with protease inhibitors (Roche) and 1 mM PMSF and rocked at 4 °C for 20 min. The solubilized PM was centrifuged at 15,871g at 4 °C and samples were suspended in LDS sample buffer (Invitrogen, LP007) with 2% β-mercaptoethanol (Thermo Scientific, 125470100).

### Mitochondrial PE quantification

First, 50 ml of cultures yeast strains expressing endogenously tagged OM45-3xHA were grown into the mid-logarithmic growth phase overnight with 250 μCi [³H]palmitate (American Radioactive Chemicals, ART0129). After 16 h of growth, 10 mM NaF and 10 mM NaN₃ were added to the culture flask to stop growth, and cultures were cooled to 4 °C. Next, 50 $OD_{600nm}$ units of cells were withdrawn and washed with ice-cold KPBS lysis buffer. Cells were transferred to 2 ml screw tubes with 300 μl of glass beads and 300 μl KPBS and lyzed with a Precellys 24 homogenizer (Bertin) at the setting three cycles of 30 s, 6,000 rpm with a 60 s pause in between at 4 °C. Debris was removed by spinning at 400g for 5 min at 4 °C and the supernatant was transferred to a new tube. The beads were washed with 650 μl of KPBS, centrifuged and the supernatant added to the previous one, forming about 800 μl of homogenate. To obtain total lipids, they were extracted from 50 μl of homogenate by mixing with 6 ml of chloroform:methanol mix (1:2) and 1.55 ml of water. To purify OM45-3xHA-containing membranes, 700 μl of homogenate was mixed with 150 μl of magnetic beads anti-HA (Pierce) washed with KPBS and rotated for 15 min at 4 °C. The beads were washed two times for 5 min with 900 μl of cold KPBS and once with 900 μl of cold 0.5 M NaCl. The beads were resuspended in 1 ml of water and transferred to a glass tube containing 6 ml chloroform:methanol (1:2) and 600 μl water. Phase separation was induced by 2 ml 0.9% NaCl and the lower phase was washed twice with 2 ml 2 M KCl. It was dried under N₂, resuspended with 30 μl chloroform:methanol (2:1) and spotted onto a TLC plate (presoaked with 1.8% boric acid and dried). The plates were developed twice with CHCl₃:EtOH:H₂O:triethylamine (30:35:7:35). The plates were then scanned with a RITA Star TLC scanner (Raytest). The location of PE was determined using a cold standard and visualized with copper sulfate staining. The amount of radiolabelled PE was normalized to the total amount of radioactive lipid on the TLC plate.

### Western blotting

Protein samples were loaded on NuPAGE precast polyacrylamide gels (Thermo Fisher Scientific) and were used for SDS–PAGE. Pageruler (and Pageruler plus) prestained protein ladders (Thermo Fisher Scientific, 26616 and 26619) were used as protein molecular weight standards. Separated proteins were transferred to 0.2 μm and 0.45 μm

nitrocellulose membranes (Bio-Rad, 1620115 and 1620112) for transferring proteins below and above 180 kDa, respectively. The membranes containing the transferred proteins were blocked in a blocking buffer (5% non-fat milk in 0.1%TBST). Membranes were probed with the appropriate primary antibodies following the dilutions in the manufacturer's protocol (refer to Supplementary Table 3) in blocking buffer containing either 5% non-fat milk or 5% bovine serum albumin in 0.1% TBST overnight at 4 °C. On the following day, the membranes were washed in 0.1%TBST, three times for 5 min each. Membranes were incubated with 1:8,000 dilutions of LI-COR secondary antibodies (Supplementary Table 3) in a blocking buffer for 1 h at room temperature. Membranes were again washed in 0.1% TBST, three times for 5 min each. The membranes were imaged on an Odyssey LICOR imager.

### Measurement of PM fluidity

Yeast cells were grown to mid-logarithimic growth phase ($OD_{600}$ of 0.6–1) and incubated live with 0.5 μM TMA-DPH (Thermo Fisher Scientific, T204) in TE buffer (10 mM Tris and 1 mM EDTA, pH 7.0) at 25 °C as described[27]. The steady-state fluorescence anisotropy of TMA-DPH was measured using a Horiba spectrofluorometer at the Biophysics core of the National Heart, Lung, and Blood Institute (NHLBI/NIH). The steady-state anisotropy ($r_s$) was calculated using a previously established method[27] ($r_s = (I_{VV} - GI_{VH})/(I_{VV} + 2GI_{VH})$); VV, vertical excitation, vertical emission; VH, vertical excitation, horizontal emission; $I$, intensity; G, instrument grating factor. Briefly, cells were incubated with TMA-DPH at 25 °C for 10 min in the dark. Cells were then washed twice with ice-cold TE buffer and resuspended at 0.25 $OD_{600}$ in TE buffer and placed on ice. Before anisotropy measurement, cells were rocked at the experimental temperatures for 5 min.

The PM fluidity of mammalian cells was measured using 3-((3-((9-(diethylamino)-5-oxo-5H-benzo[a]phenoxazin-2-yl)oxy)propyl)(dodecyl)(methyl)ammonium)propane-1-sulfonate (NR12S; Tocris Bioscience, 7509) using the method described by Smith et al.[52]. The NR12S-stained cells were excited at 520 nm and fluorescence emissions were measured at 560 nm and 630 nm using a BMG Omega plate reader. The ratio of emission at 560 nm and 630 nm was calculated.

### Production of lentivirus and infection

Lentiviruses were produced and transduced using a previously established method[53]. Briefly, the lentiviral plasmid (pLKO.1) expressing shRNA was co-transfected with packaging vectors psPAX2 and pMD2G into HEK293T cells. Media containing the virus was collected after 48 h and filtered through a 0.45 μm polyethersulfone low-binding filter (Millipore Sigma, SLHPM33RS). This was then concentrated using a Lenti-X concentrator (Clontech, 631231). Finally, the reconstituted lentiviruses were stored at −80 °C for future use. shRNAs depleting BLTP2 (TRCN0000128930, TRCN0000129099, TRCN0000130217, TRCN0000128410 ad TRCN0000129809) were purchased from Millipore Sigma as bacterial glycerol stocks. shUT containing an untargeted shRNA was used as previously published in our work[16]. Cells were infected with lentiviruses and 8 μg ml$^{-1}$ of polybrene (Millipore Sigma, TR-1003-G). The growth medium was changed to a fresh medium containing puromycin for selection. Experiments were conducted with puromycin-selected cells within 5–7 days post-infection.

### qRT–PCR

We used quantitative PCR with reverse transcription (qRT–PCR) to verify the knockdown of BLTP2 using a previously established method[16]. For reverse transcription PCR, a qScript cDNA supermix (Quantabio, 95048-025) was used and the manufacturer-recommended protocol was followed. For qPCR, PowerTrack SYBR green master mix (Thermo Fisher Scientific, A46012) and a manufacturer-recommended protocol for a 20 μl reaction were used. qPCR was conducted using a Bio-Rad C1000 Touch Thermal Cycler (CFX96 Real-Time system). The oligonucleotides used for qRT–PCR of BLTP2 mRNA were HuBLTP2qpcr_F

and HuBLTP2qpcr_R. GAPDH mRNA was used as internal control and qPCR was conducted using the oligonucleotides GAPDHqpcr_F and GAPDHqpcr_R. The thermocycler parameter that was used for qRT–PCR was one cycle of 'enzyme activation' (95 °C for 2 min) followed by 40 cycles of 'denature' (95 °C for 15 s) and 'anneal/extend' (60 °C for 60 s).

### Generation of CRISPR KO cell lines

BLTP2 was knocked out in HeLa cells using the pSpCas9(BB)-2A-Puro plasmid[54] and sgRNA targeting Exon2. sgRNA was ordered as oligonucleotides (sgBLTP2-F and sgBLTP2-R) and cloned using a BbsI restriction enzyme digested vector according to the method described by the Feng Zhang lab at the MIT in Addgene (Addgene, 62988)[54]. We followed a previously established method[55] to obtain a single clone of the BLTP2 knockout cell line (BLTP2-KO) and verified BLTP2-KO clones by PCR using the oligonucleotides BLTP2_Ex2_seq_F and BLTP2_Ex2_seq_R followed by Sanger sequencing of the purified PCR product. Precisely, cells were transfected with the pSpCas9(BB)-2A-Puro plasmid containing the sgRNA with Fugene HD transfection reagent (Promega, E2311) following the manufacturer's protocol. Cells were selected with puromycin and plated in a 96-well plate to obtain single clones. gDNA from clones was extracted using a gDNA extraction kit (Zymo Research, D3024) and PCR was conducted using KAPA HiFi DNA Polymerase (Roch, KK2601). The verified BLTP2-KO clone was grown in 10 cm tissue culture plates and stored as liquid nitrogen stocks for subsequent experiments. The TLCD1-KO and isogenic control HeLa cell lines were a kind gift by Dr. Kasparas Petkevicius, MRC Mitochondrial Biology Unit at Cambridge University[43].

### Cell proliferation assays

Human-derived cell lines were plated at equal densities for each experiment. For an automated cell count assay[56], experiments were conducted in a 6-well plate or a 24-well plate and $10^5$ cells in a 6-well plate were plated for each genotype and grown in the indicated medium. For experiments conducted in 24-well plates, $0.25 \times 10^5$ cells were plated for each genotype and condition. On days 4–5, cells were trypsinized with 0.5–1 ml 0.25% trypsin/EDTA and resuspended in 1–3 ml of growth medium. Adding 10 μl trypan blue to 30 μl of cell suspension, 10 μl of the mix was injected in a Countess chamber slide (Invitrogen), and the cell number per milimetre was counted in a Countess 3 (Invitrogen by Thermo Fisher Scientific). Cell numbers were corrected for trypan blue addition and the volume of resuspension and presented as histograms.

For the colony formation assay, crystal violet staining was used. A serial tenfold dilution of cells was plated. After days 4–5 of incubation, plates were placed on ice and washed two times with ice-cold 1× DPBS (Gibco, 14190-144). Cells were fixed with ice-cold methanol for 10 min. After aspirating methanol from the plates, plates were removed from ice and enough 0.5% crystal violet solution was added to cover the bottom of the plate. After incubating at room temperature for 10 min, the crystal violet solution was discarded. The wells were carefully rinsed in de-ionized water until the blue colour was no longer coming off in the rinse. Plates were then dried at room temperature overnight. Images were taken either in UVP GelSolo Imager by Analytik Jena or using a transilluminator and iPhone 11 camera.

WST-1 assays to measure cell proliferation rate (Roche, 5015944001) were conducted using the manufacturer's protocol.

### Annexin-V apoptosis assay

Flow cytometry-based detection of early and late apoptosis of cancer cells was conducted using the Annexin-V apoptosis assay kit (Thermo Fisher Scientific, V13241). Staining of cells with annexin-V and propidium iodide was conducted according to the manufacturer's protocol. The stained cells were analysed by flow cytometry using the UT Southwestern flow cytometry core and plotted as histograms using GraphPad Prism.

## LC–MS/MS-based lipidomics

Total fatty-acid quantification was performed using gas chromatography with flame ionization detector as described[57]. For LC–MS/MS analyses, the lipid extracts corresponding to 25 nmol of total fatty acids were dissolved in 100 µl of chloroform:methanol (2:1 (v/v)) containing 125 pmol of each internal standard. Internal standards used were PE 18:0-18:0 and DAG 18:0-22:6 from Avanti Polar Lipid and sulfoquinovosyldiacylglycerol 16:0-18:0 extracted from spinach thylakoid[58] and hydrogenated as described in ref. 59. Lipids were then separated by HPLC and quantified by MS/MS.

The HPLC separation method was adapted from a previously published method[27]. Lipid classes were separated using an Agilent 1200 HPLC system using a 150 mm × 3 mm (length × internal diameter) 5 µm diol column (Macherey-Nagel), at 40 °C. The mobile phases consisted of hexane/isopropanol/water/ammonium acetate 1 M, pH 5.3 (625/350/24/1, (v/v/v/v)) (A) and isopropanol/water/ammonium acetate 1 M, pH 5.3 (850/149/1, (v/v/v)) (B). The injection volume was 20 µl. After 5 min, the percentage of B was increased linearly from 0% to 100% in 30 min and stayed at 100% for 15 min. This elution sequence was followed by a return to 100% A in 5 min and an equilibration step for 20 min with 100% A before the next injection, leading to a total run time of 70 min. The flow rate of the mobile phase was 200 µl min⁻¹. The distinct glycerophospholipid classes were eluted successively as a function of the polar head group.

Mass spectrometric analysis was done on a 6470 triple quadrupole mass spectrometer (Agilent) equipped with a Jet stream electrospray ion source under the following settings: drying gas heater: 260 °C; drying gas flow 13 l min⁻¹; sheath gas heater: 300 °C; sheath gas flow: 11 l min⁻¹; nebulizer pressure: 25 psi; capillary voltage: ±5,000 V; nozzle voltage ±1,000. Nitrogen was used as collision gas. The quadrupoles Q1 and Q3 were operated at widest and unit resolution, respectively. PC and Lyso-PC analysis was carried out in positive ion mode by scanning for precursors of $m/z$ 184 at a collision energy (CE) of 34 eV. Sulfoquinovosyldiacylglycerol analysis was carried out in negative ion mode by scanning for precursors of $m/z$ −225 at a CE of −56 eV. PE, phosphatidylinositol (PI), PS, PG, phosphatidic acid (PA) and Lyso-PE measurements were performed in positive ion mode by scanning for neutral losses of 141 Da, 277 Da, 185 Da, 189 Da, 115 Da and 141 Da at CEs of 20 eV, 12 eV, 20 eV, 16 eV, 16 eV and 20 eV, respectively. Plasmanyl-ethanolamine (PE-A) and plasmanelyl-ethanolamine (PE-P) measurements were performed in positive ion mode by scanning for $sn-1$ ether + $C_2H_8NO_3P$ product at CE of 20 eV (ref. 60). Quantification was done by multiple reaction monitoring (MRM) with 30 ms dwell time. DAG and TAG species were identified and quantified by MRM as singly charged ions $[M + NH_4]^+$ at a CE of 16 and 22 eV, respectively, with 30 ms dwell time. CL species were quantified by MRM as singly charged ions $[M−H]^-$ at a CE of −45 eV with 50 ms dwell time. Mass spectra were processed by MassHunter Workstation software (Agilent) for identification and quantification of lipids. Lipid amounts (pmol) were corrected for response differences between internal standards and endogenous lipids and by comparison with quality control. Quality control extract corresponds to a known lipid extract from yeast qualified and quantified by TLC and gas chromatography with flame ionization detector[61]. Analysed transitions are based on a previously published method[62] for yeast and on published methods[60,63,64] for mammalian samples.

## Data presentation of lipid species

The mol percentage of measured glycerophospholipid species were measured from triplicate cultures of indicated yeast genotype. The logarithmic (log to the base 2) FC of lipid species from yeast grown at the indicated temperature with or without EtN supplementation was measured using MS Excel. An unpaired $t$-test was conducted to obtain $P$ values between conditions and converted to a log value using MS Excel. Volcano plots were plotted using GraphPad Prism 10.

An unsaturation Index to measure the number of double bonds per glycerophospholipid was used to calculate changes in lipid unsaturation for yeast of indicated genotype, grown at the indicated temperature, with or without EtN supplementation in the medium. Unsaturation index was calculated as (% mono-unsaturated + 2× % di-unsaturated)/100 (ref. 65).

For mammalian glycerophospholipid species analysis, the mol percentage of indicated glycerophospholipid species were measured from triplicate cultures of cells and analysed using the heat map feature of GraphPad Prism 10.

## Zebrafish husbandry and xenograft sample preparation

Zebrafish husbandry and experiments described here have been approved and conducted under the oversight of UT Southwestern's Institutional Animal Care and Use Committee under protocol number 101805 to Gaudenz Danuser. Zebrafish adults and embryos were kept at 28.5 °C and were handled according to established protocols[45]. To visualize cancer cells in a near-physiological environment in situ, a zebrafish (*Danio rerio*) line expressing the vascular marker *Tg(kdrl:Hsa. HRAS-mCherry)*[66] in a Casper background[67] was used.

At 2.25 days post fertilization, zebrafish larvae were xenografted with MDA-MB-231 breast cancer cells. The MDA-MB-231 cells expressed *Ftractin-EGFP* stably to label the cells and reveal their actin organization[68]. The *Ftractin-EGFP* construct was a gift from Dr. Dyche Mullins at UCSF (Addgene plasmid 58473) and MDA-MB-231 cells were a gift from Dr. Rolf Brekken (UT Southwestern Medical Center). To study the effect of BLTP2 on migration and survival in vivo, we knocked down BLTP2 using shBLTP2 in MDA-MB-231 *Ftractin-EGFP* cells 5 days before xenografting as described above. An equal dose of the control shUT virus was transduced into an equal number of MDA-MB-231 *F-tractin-EGFP* cells. We compared the outcome of the xenografts of shBLTP2 transduced cells with a xenograft of shUT-transduced cells.

To xenograft the cells, they were grown up to 70–90% confluency and trypsinized for 3 min (Gibco, 15400-054). For injection, $4 × 10^6$ cells in 40 µl of cell culture media were prepared and stored on ice until xenografting into the yolk near the common cardinal vein of zebrafish larva. The injection was performed with glass capillary needles (World Precision Instruments, 1B100-4), pulled on a micropipette puller (Sutter Instrument, P-1000). Thereby, 50–500 cells were injected per fish. During the injection, zebrafish were anesthetized with Tricaine (Sigma-Aldrich, E10521).

## In vivo cell migration and survival assay

To analyse the dissemination and survival of cancer cells in situ, we performed imaging of xenografted zebrafish. At 5, 24 and 48 h after injection, zebrafish xenografts were imaged on a Leica M205 FA fluorescence stereo microscope with a Planapo 1.0× objective and an X-Cite XYLIS illumination source. In addition, we obtained high-resolution images of cancer cells inside the vasculature in selected xenografts with axially swept light-sheet microscopy[69] on a custom microscope with an NA1.0 detection objective and 55× magnification. To immobilize the zebrafish for imaging, zebrafish embryos were anesthetized with 200 mg l⁻¹ Tricaine (Sigma-Aldrich, E10521) during imaging[70]. Between imaging sessions, zebrafish larvae were maintained at 34 °C.

For each condition, two experimental repeats were performed with the shBTLP2 knockdown experiment containing $n_1 = 74$ and $n_2 = 63$ larvae, and shUT experiments containing $n_1 = 75$ and $n_2 = 80$ larvae.

To quantify cancer dissemination, we counted all zebrafish with micro-metastases to the tail part and with cancer cells remaining at the injection site in the yolk. We further counted all zebrafish with no cancer cells left and larvae that died during the experiment. To determine the spread of the cancer cells, we manually annotated the cancer cell locations of each zebrafish larva in a template fish using Fiji[71]. We obtained quantitative cancer dissemination maps by summing all manual annotations per timepoint and condition. To visualize the

cancer dissemination maps, we overlaid them onto the template larva, with a rainbow look-up table in Adobe Photoshop.

### Design of *C. elegans* strains, culture and imaging

*C. elegans* strains used in this study: N2 Bristol (WT), AG674 *fmp-27Δ(av263) was* generated by CRISPR–Cas9 editing to delete the full length of *bltp-2*. *C. elegans* strains were maintained with standard protocols in the Golden lab. The Bristol N2 strain was used as the WT for CRISPR–Cas9 genome editing. A 20 bp nucleotide sequence of the *fmp-27*-specific crRNAs were selected with the help of a crRNA design tool from Integrated DNA Technologies. All crRNAs and tracrRNA were synthesized by Horizon Discovery (https://horizondiscovery.com). The single-stranded donor oligonucleotides of the *bltp-2* repair template were synthesized by Integrated DNA Technologies, and the detailed sequence information of the CRISPR reagents is listed in Supplementary Table 2. DIC images were taken by a spinning disk confocal system that includes a Photometrics Prime 95B EMCCD camera, and a Yokogawa CSU-X1 confocal scanner unit. Images were acquired by Nikon's NIS imaging software using a Nikon 10× objective with 2–4 µm z-step size; 10–20 planes were captured. The quantification of body length was analysed by ImageJ/FIJI Bio-format plugin (National Institutes of Health)[71,72]. The animals were immobilized on 7% agar pads with an anaesthetic (0.01% levamisole in M9 buffer). Statistical significance was determined by the *P* value from an unpaired two-tailed *t*-test. Both the Shapiro–Wilk and Kolmogorov–Smirnov normality tests indicated that all data follow normal distributions. We thank the Caenorhabditis Genetics Center, which is funded by the National Institutes of Health Office of Research Infrastructure Programs (P40OD010440), for providing strains for this study.

### Statistics and reproducibility

Values were calculated and plotted in GraphPad Prism to generate histograms, violin plots, volcano plots and heat maps. Experiments were conducted in at least triplicates, except for experiments in Extended Data Figs. 1a and 6c, which were repeated twice. All attempts at replication were successful. For statistical comparison, when two mean values were compared against each other, an unpaired *t*-test was conducted in GraphPad Prism to calculate and plot *P* values in the graph. The *P* values are indicated in the figures. To compare the mean values of multiple entities against each other a one-way analysis of variance (ANOVA) was used, and a two-way ANOVA was used when two independent variables were involved in GraphPad Prism. The *P* values are indicated in the figures. Data used in this study are available in the article. No statistical method was used to predetermine sample size. No data were excluded from the analyses and no randomization was performed in this study. Data distribution was assumed to be normal but this was not formally tested. Data collection and analysis were not performed blind to the conditions of the experiments.

### Reporting summary

Further information on research design is available in the Nature Portfolio Reporting Summary linked to this article.

## Data availability

Data used in this study are available in the main figures, extended data figures, source data and Supplementary Tables 1–3 of this Article. Blots for Figs. 1e, 2e and 4d were cut into strips. The publicly available CRISPR Chronos dataset was taken from DepMap. Lipidomics data used in this study are publicly available via the Open Science Framework at https://doi.org/10.17605/OSF.IO/RP7HE. Source data are provided with this paper.

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

## Acknowledgements

This work was supported by NIH no. R35GM153315 (to W.A.P.) and the Intramural Research Program of the National Institute of Diabetes and Digestive and Kidney Diseases (no. 1 ZIA DK060105 10 to W.A.P.). The zebrafish work was supported by NIH grant nos. R35GM133522 (to R.F.) and U54CA268072 (to R.F.) and computational resources provided by the BioHPC supercomputing facility located in the Lyda Hill Department of Bioinformatics, UT Southwestern Medical Center (to R.F.) LC–MS/MS was conducted on the LIPANG (Lipid analysis in Grenoble) platform hosted by the LPCV, Université Grenoble Alpes, and supported by the Rhône-Alpes Region, the fonds FEDER, and GRAL, financed within the University Grenoble Alpes graduate school (Ecoles Universitaires de Recherche) no. CBH-EUR-GS (ANR-17-EURE-0003) to M.M. and J.J The funders had no role in study design, data collection and analysis, decision to publish or preparation of the manuscript. We thank K. Petkevicius (Cambridge University), G. Piszczek (NIH), A. Dutta (NIH), H. R. Shin (UTSW), B. Porter (UTSW), M. Mettlen (Quantitative Light Microscopy Core, UTSW), D. Masison (NIH), J. Hanover (NIH) and O. Cohen-Fix (NIH), J. Nunnari (Altos Labs) and I. Chakraborty (UTSW) for reagents, technical assistance, support, or suggestions on experimental methods. We thank the Animal Resource Center (ARC) and G. Danuser (UTSW) for the use of the zebrafish facility, and the UTSW Flow Cytometry Core for the use of their facility. Schematic diagrams were created with BioRender.com.

## Author contributions

S.B. conceptualized the project, set up collaborations, prepared figures and wrote the manuscript. S.B., S.D., X.B., M.M., J.J., D.B., S.M., E.J., C.-W.W. and A.T. conducted the investigation. W.A.P. and A.T. supervised the project. R.F. supervised the investigation in zebrafish. W.A.P. co-wrote the manuscript.

## Competing interests

The authors declare no competing interests.

## Additional information

**Extended data** is available for this paper at https://doi.org/10.1038/s41556-025-01672-3.

**Correspondence and requests for materials** should be addressed to Alexandre Toulmay or William A. Prinz.

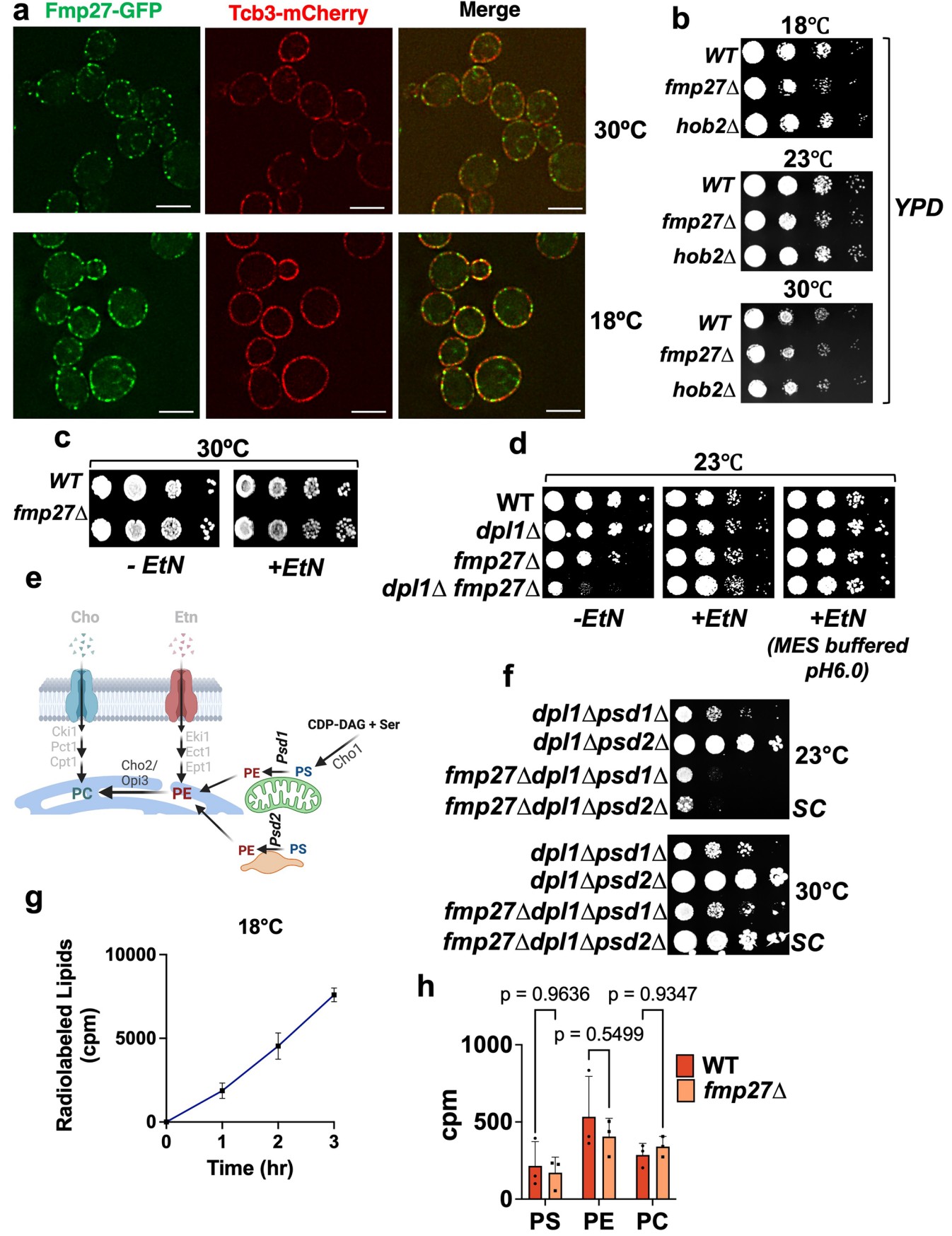

**Extended Data Fig. 1 | See next page for caption.**

**Extended Data Fig. 1 | Fmp27 localizes to ER-PM contact sites and cells lacking Fmp27 do not have defects producing PLs by the CDP-DAG pathway.**
**a**. Images of yeast cells expressing endogenously tagged Fmp27-GFP and Tcb3-mCherry grown at 30 °C (top) or 18 °C (bottom) in SC medium; scale bars = 5 µm. Micrographs are representative images from two experiments. **b**. Serial dilution of yeast strains spotted on YPD plates and incubated for 30 °C (3 days), 23 °C (4 days) or 18 °C (5 days). **c**. Serial dilution of yeast strains spotted on SC plates with or without 4 mM EtN and incubated at 30 °C for 3 days. **d**. Serial dilution of yeast strains spotted on SC plates with or without 4 mM EtN or on plates buffered with 50 mM MES, pH 6 and 4 mM EtN incubated at 23 °C for 3 days. **e**. Scheme showing the synthesis of PS, PE, and PC lipids in the CDP-DAG pathway; the enzymes involved in biosynthesis and their localizations are indicated. **f**. Serial dilution of yeast strains spotted on SC plates and incubated at 23 °C (top) and 30 °C (bottom) for 3 days. **g**. Incorporation of [³H]serine (counts per min, cpm) in lipids (CHCl₃ phase) over time in WT cells grown at 18 °C. Points represent mean ± 95% confidence interval (n = 3); [³H]serine added at time = 0. **h**. Values used to calculate results shown in Fig. 1j; cpm in each PL; mean ± SEM (n = 3); p-values from two-way ANOVA. Source numerical data are available in source data.

**a**

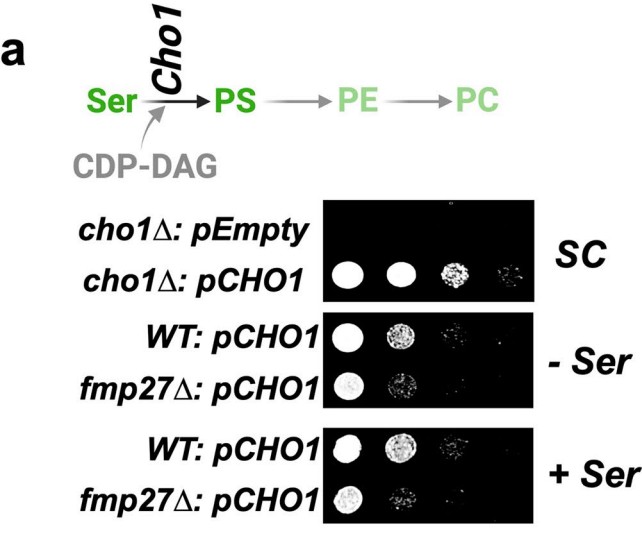

**b**

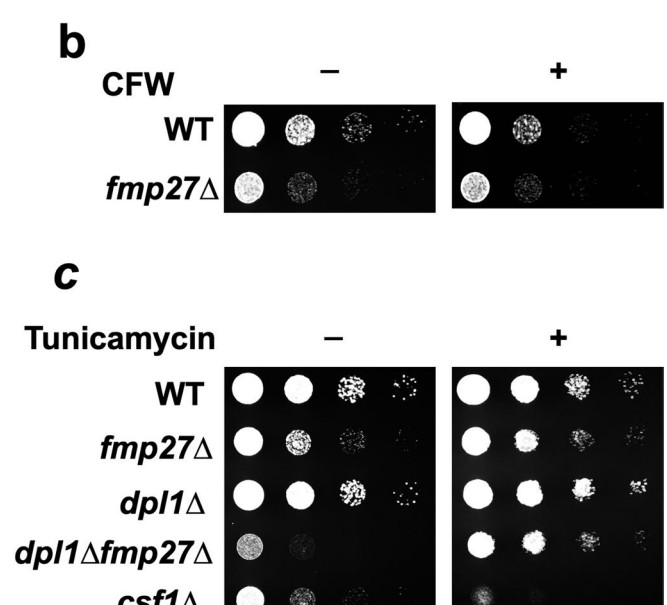

**Extended Data Fig. 2 | Lack of Fmp27 does not compromise PL- and GPI-anchor biosynthesis and does not elicit ER stress. a** Scheme showing how PS synthase (Cho1) uses serine (Ser) and CDP-DAG to form PS, which can be converted to PE and then PC (top). The bottom panel shows serial dilution of yeast strains spotted on SC media plates and grown at 18 °C for 3 days. The strains contained an empty plasmid (pEmpty) or a plasmid expressing the *CHO1* gene under a *GPD1* promoter (pCHO1). Where indicated, the SC media was supplemented with 4 mM serine. **b** Serial dilution of yeast strains spotted on SC medium without (left) or with (right) 10 μg/ml calcofluor white (CFW). Plates were grown at 18 °C for 3 days. **c** Serial dilution of yeast strains spotted on SC medium without (left) or with (right) 0.5 μg/ml tunicamycin. Plates were grown at 23 °C for 3 days.

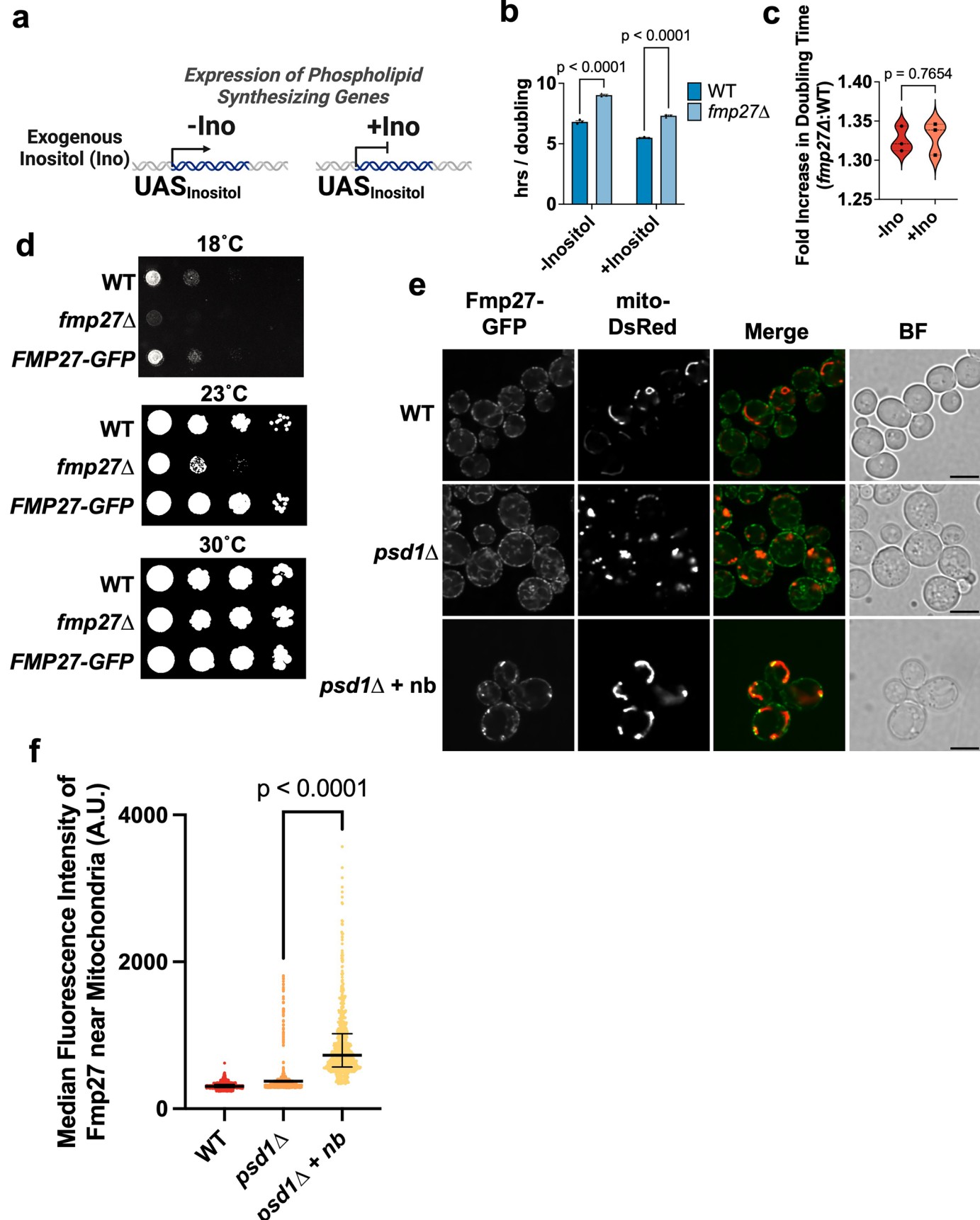

**Extended Data Fig. 3 | See next page for caption.**

**Extended Data Fig. 3 | Fmp27 does not play a role in inositol-mediated regulation of lipid metabolism and controls for Fig. 3(e-i). a**. Scheme describing the role of inositol (Ino) in regulating transcription of PL biosynthetic genes in yeast. **b** Doubling time of yeast strains grown at 18 °C in SC lacking Ino or containing 100 mM Ino. Histograms show mean ± SEM (n = 3). p-values from two-way ANOVA. **c** Box-violin plots of relative difference between the strains from b(n = 3); thick dotted line denoting median and fine dotted lines signifying first and third quartiles. p-values from unpaired two-tailed t-tests. **d** Serial dilution of yeast strains spotted on SC plates at indicated temperatures for 3 days. **e** Images of yeast strains growing at logarithmic growth phase in SC at 30 °C. Panels show Fmp27-GFP, mitoDsRed (mitochondrial marker), merged channels, and brightfield images of the cells; scale bars = 5 μm. Micrographs are representative images from three experiments. **f** Median fluorescence intensity (solid line) with interquartile range (error bars) of Fmp27-GFP (n = 3) within 200 nm of mitochondria in indicated yeast stains. p-value from unpaired two-tailed t-test. Source numerical data are available in source data.

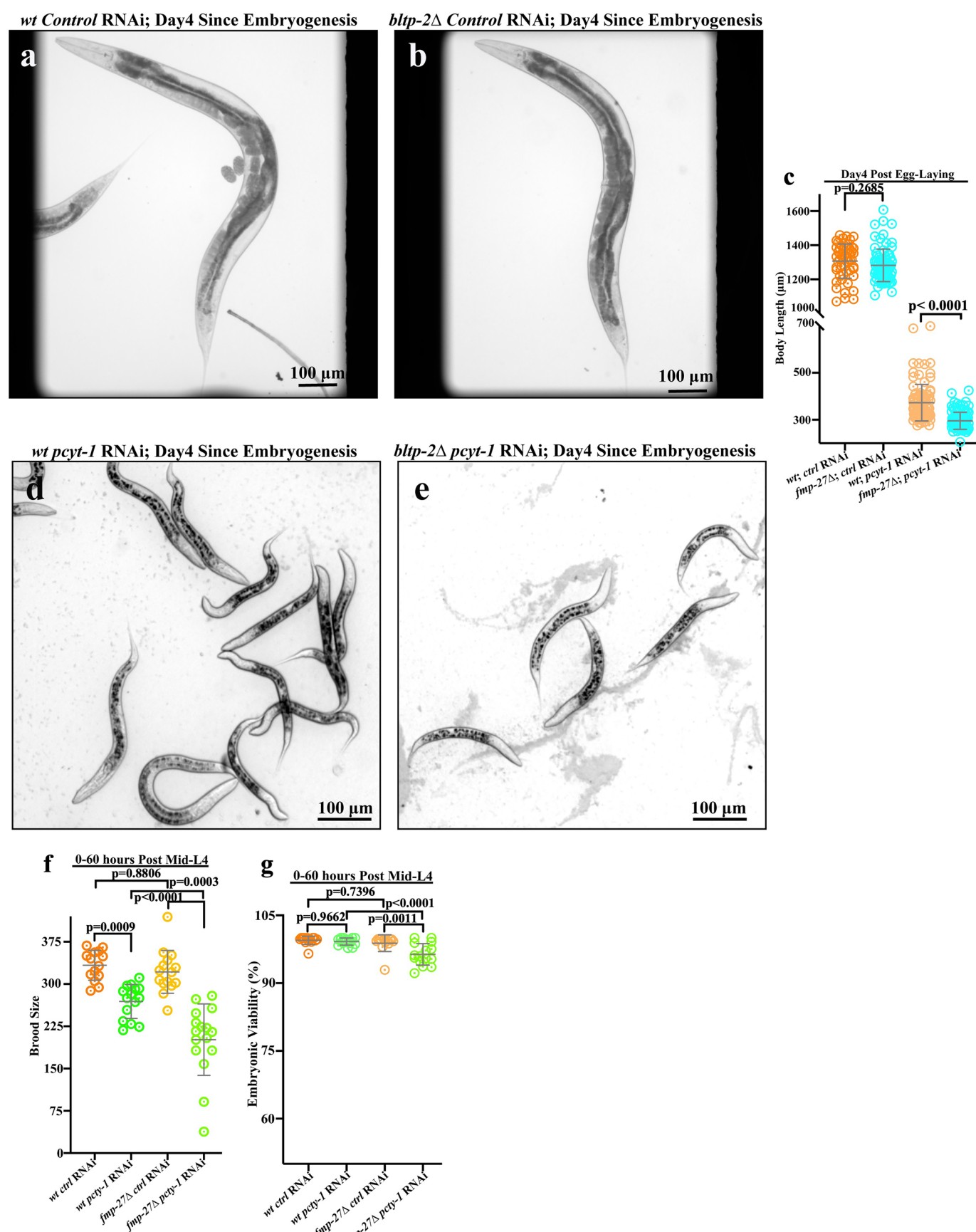

**a** *wt Control* RNAi; Day4 Since Embryogenesis

**b** *bltp-2Δ Control* RNAi; Day4 Since Embryogenesis

**c** Day4 Post Egg-Laying

**d** *wt pcyt-1* RNAi; Day4 Since Embryogenesis

**e** *bltp-2Δ pcyt-1* RNAi; Day4 Since Embryogenesis

**f** 0-60 hours Post Mid-L4

**g** 0-60 hours Post Mid-L4

**Extended Data Fig. 4 | See next page for caption.**

**Extended Data Fig. 4 | The *C. elegans* gene *bltp-2* coordinates with the Kennedy pathway to maintain growth and physiology. a-b**. Representative images of wildtype and *bltp-2Δ* animals treated with control RNAi (a) or *bltp-2* RNAi on day 4 post egg-laying. **c.** Quantification (mean ± SD) of the results shown in a. and b; n = 98, 103, 54 and 81 worms for each genotype from left to right. p-values: ****, p < 0.0001 (unpaired two-tailed t-tests). **d-e.** Depletion of *pcyt-1* by RNAi causes larval arrest in wildtype and *bltp-2Δ* mutant animals. The body length is synergistically smaller in the *bltp-2Δ* mutant animals compared to the wild-type control. **f.** A smaller brood size was observed in both wildtype and *bltp-2Δ* animals when depleted of *pcyt-1* by RNAi. The *pcyt-1* RNAi synergistically reduced the brood size in *bltp-2Δ* animals compared to wildtype. Mean ± SD (n = 14, 15, 15 and 16 broods for each genotype from left to right). p-values from one-way ANOVA. **g.** Embryonic viability was reduced in *bltp-2Δ* animals after depletion of *pcyt-1* by RNAi. Mean ± SD (n = 14, 13, 15 and 15 batches of embryos for each genotype from left to right). p-values from one-way ANOVA. P-values: **: p = 0.0011, ***: p = 0.0009, ****, p < 0.0001 (unpaired t-tests). Source numerical data are available in source data.

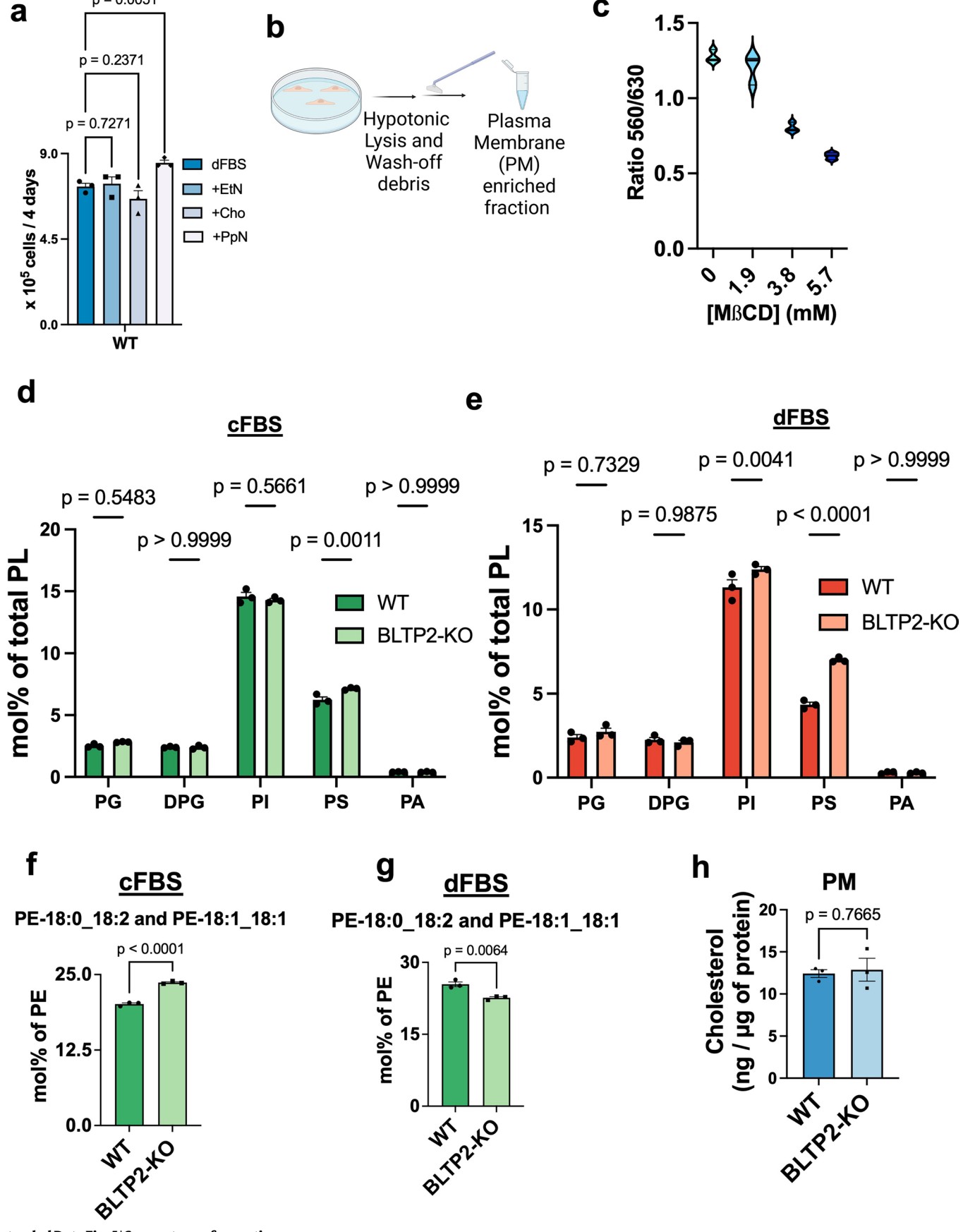

**Extended Data Fig. 5 | See next page for caption.**

**Extended Data Fig. 5 | Controls for Fig. 4 and BLTP2 deletion does not alter PM cholesterol levels. a**. Automated cell count assay for WT HeLa cells grown with DMEM and dFBS with or without 10 µM EtN, Cho, or PpN; mean ± SEM (n = 3) values; p-values are from unpaired two-tailed t-tests. **b** Scheme demonstrating the method of isolating a PM-enriched fraction from HeLa cells, drawn using biorender.com. **c** Violin plot of the ratiometric fluorescence intensity of NR12S (Y-axis) and concentration of [MßCD] (X-axis). Values from 4 independent experiments are shown. thick line denoting median and fine lines signifying first and third quartiles. **d** Relative abundance of PL species (except PE and PC), determined by LC-MS/MS, from HeLa cells grown in DMEM with cFBS; mean ± SEM of 3 independent replicates. p-values from two-way ANOVA.

**e** Relative abundance of PL species (except PE and PC), determined by LC-MS/MS, from HeLa cells grown in DMEM with dFBS; mean ± SEM of 3 independent replicates. p-values from two-way ANOVA. **f** Relative abundance of the indicated PE species determined by LC-MS/MS, from HeLa cells grown in DMEM with cFBS; mean ± SEM of 3 independent replicates. p-values from unpaired two-tailed t-test. **g** Relative abundance of PE species determined by LC-MS/MS, from HeLa cells grown in DMEM with dFBS; mean ± SEM of 3 independent replicates. p-values from unpaired two-tailed t-test. **h** PM cholesterol level normalized to µg of total cellular protein from indicated genotypes of HeLa cells; mean ± SEM of 3 independent replicates. p-values from unpaired two-tailed t-test. Source numerical data are available in source data.

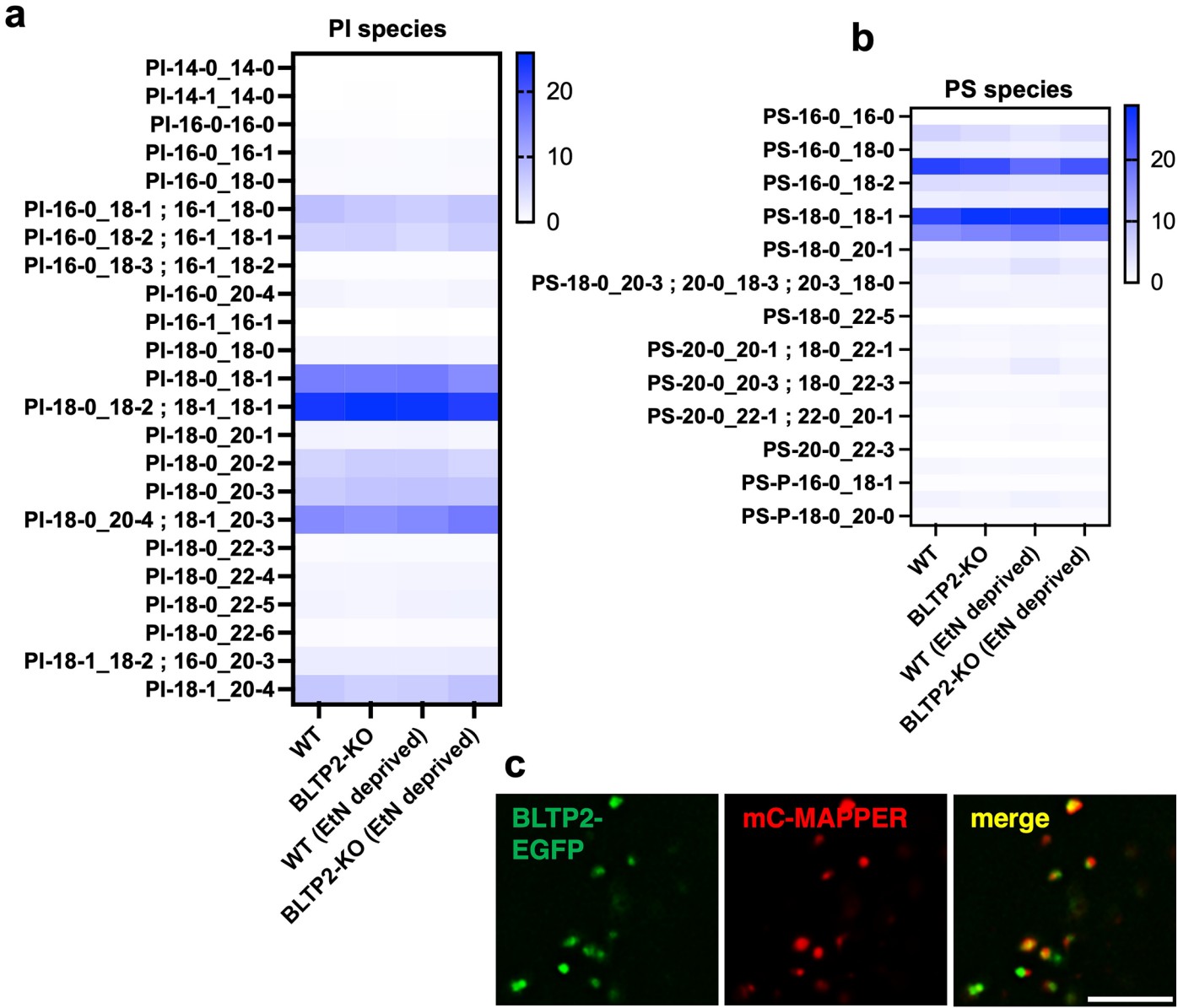

**Extended Data Fig. 6 | PI and PS lipid species and localization of BLTP2-EGFP. a-b.** Results from the experiment shown in Fig. 4g for PI (**a**) and PS (**b**) species. **c.** Images of U2OS cells transiently expressing the ER-PM marker mCherry-MAPPER[49] and BLTP2-EGFP. Scale bar = 5 µm. Micrographs are representative images from two experiments. Source numerical data are available in source data.

## a

*Co-dependency analysis with BLTP2*

| Gene | Entrez Id | Dataset |
|------|-----------|---------|
| WDR44 | 54521 | CRISPR (DepMap Public 23Q2+Score, Chronos) |
| **TLCD1** | 116238 | CRISPR (DepMap Public 23Q2+Score, Chronos) |
| HMGB1 | 3146 | CRISPR (DepMap Public 23Q2+Score, Chronos) |
| XYLT2 | 64132 | CRISPR (DepMap Public 23Q2+Score, Chronos) |
| PIGS | 94005 | CRISPR (DepMap Public 23Q2+Score, Chronos) |

## b

**Extended Data Fig. 7 | BLTP2 and TLCD1. a**. Table of top 5 co-dependencies of the BLTP2 gene from the CRISPR dataset of DepMap[13,14]. **b** Crystal violet-stained colony forming units of HeLa cells with the indicated genotypes plated at the indicated cell number.

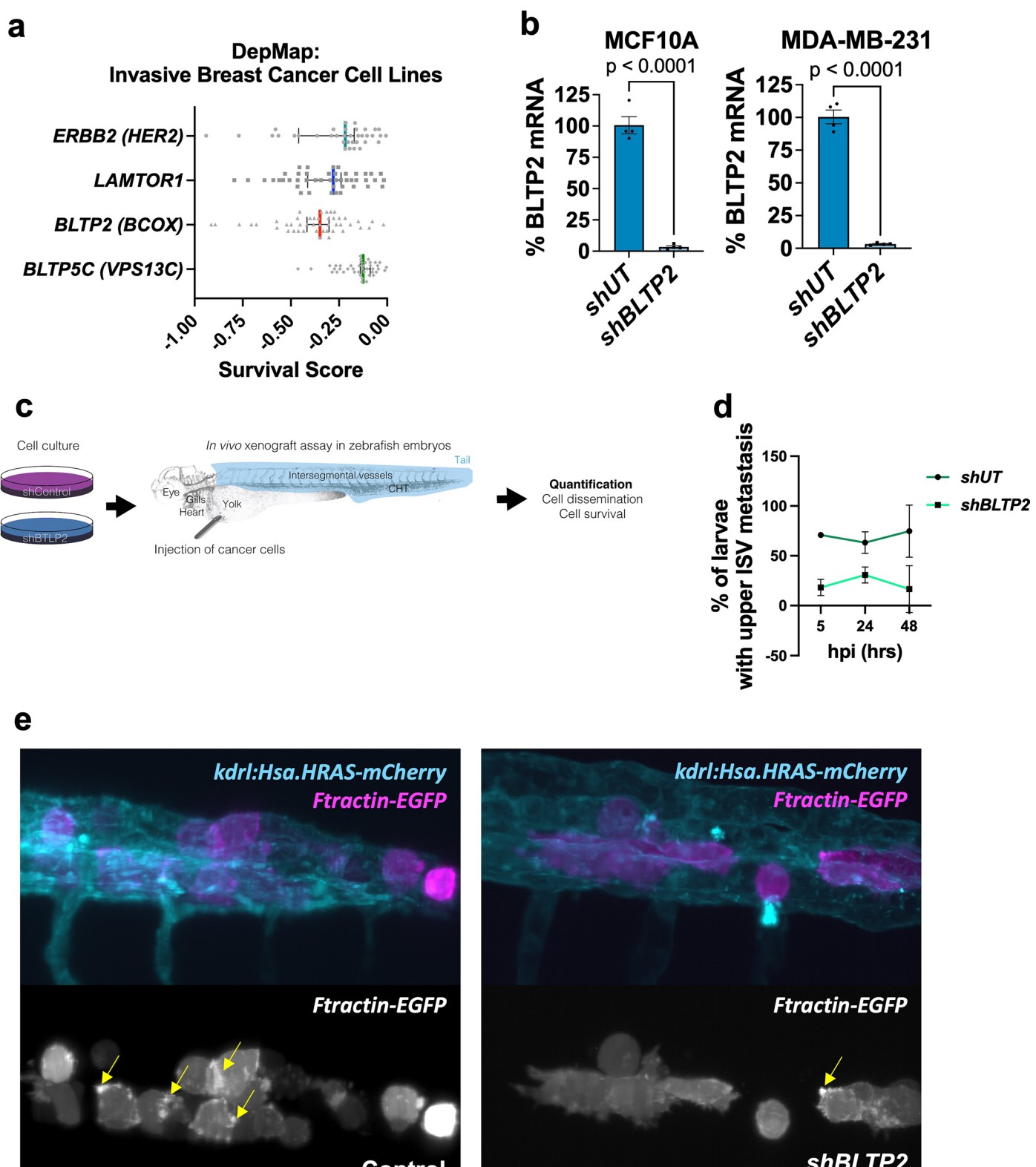

**Extended Data Fig. 8 | See next page for caption.**

**Extended Data Fig. 8 | BLTP2 controls actin ruffling in MDA-MB-231 cells expressing *F-tractin*. a** DepMap survival score of 47 invasive breast cancer cell lines with CRISPR-mediated knockouts of the genes ERBB2, LAMTOR1, BLTP2, or BLTP5C. Colored lines indicate the median survival score, error bars show 95% confidence intervals. The survival score on the x-axis is from 0 (no effect on survival) to −1 (gene is essential for survival). **b** qRT-PCR verifying the knockdown of BLTP2 mRNA when silenced with a shBLTP2 cocktail in MCF10A (left) and MDA-MB-231 (right) cells. Histograms show mean ± SEM (n = 4); p-values from unpaired two-tailed t-tests. **c** Schematic of *in vivo* xenograft assay with MDA-MB-231 *F-tr#ctin* cells in zebrafish larvae. **d** Percentage of zebrafish larvae (mean ± SD) that have MDA-MB-231 cells metastasized to their intersegmental

veins (ISV) following xenografts with cells depleted of BLTP2 (shBLTP2) (n = 137 fishes) or not depleted (shUT) (n = 155 fishes). **e** Top panel shows localization of MDA-MB-231 cells stably expressing the cytoplasmic actin filament reporter *Ftractin-EGFP* (magenta) in zebrafish blood vessels (*Tg(kdrl:Hsa.HRAS-mCherry)*, cyan); The bottom panel shows actin ruffling (indicated by yellow arrows) at PM protrusion sites in control (left) and shBLTP2 cocktail (right) treated MDA-MB-231 cells. Maximum intensity projections of volumetric data acquired with axially swept light-sheet microscopy are shown. Scale bars = 25 μm. Micrographs are representative images from three experiments. Source numerical data are available in source data.

## a

### Gating strategy for Fig. 5f

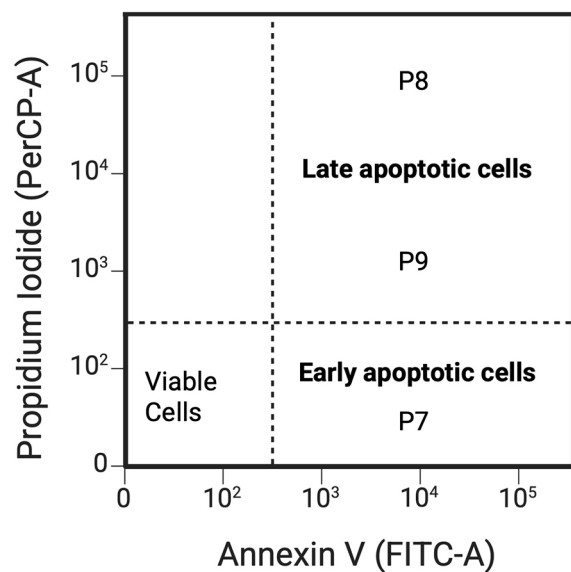

## b

BD FACSDiva 8.0.2

| Tube: SB12MAY23_Sh_UT-1 | | | |
|---|---|---|---|
| Population | #Events | %Parent | %Total |
| ☐ All Events | 12,861 | #### | 100.0 |
| 🟥 P1 | 11,571 | 90.0 | 90.0 |
| 🟦 P2 | 11,066 | 95.6 | 86.0 |
| 🟩 P3 | 10,772 | 97.3 | 83.8 |
| 🟦 P4 | 94 | 0.9 | 0.7 |
| 🟪 P5 | 84 | 0.8 | 0.7 |
| 🟨 P6 | 30 | 0.3 | 0.2 |
| 🟦 P7 | 291 | 2.3 | 2.3 |
| 🟪 P8 | 159 | 1.2 | 1.2 |
| 🟨 P9 | 153 | 1.2 | 1.2 |

**Extended Data Fig. 9 | Flow cytometer gating strategy and representative flow cytometry figure for Fig. 5f. a.** A schematic showing the flow cytometer gating strategy to analyse early- and late-apoptosis. The X-axis and Y-axis indicate fluorescence intensity in a log scale. Dotted lines separate quartiles. Quartiles indicating viable, early apoptotic, and late apoptotic cells are shown. **b** A representative flow cytometry figure showing analysis of MDA-MB-231 cells transduced with shUT stained with Annexin-V and propidium iodide.

# Reporting Summary

## Statistics

For all statistical analyses, confirm that the following items are present in the figure legend, table legend, main text, or Methods section.

| n/a | Confirmed | |
|---|---|---|
| ☐ | ☒ | The exact sample size (*n*) for each experimental group/condition, given as a discrete number and unit of measurement |
| ☐ | ☒ | A statement on whether measurements were taken from distinct samples or whether the same sample was measured repeatedly |
| ☐ | ☒ | The statistical test(s) used AND whether they are one- or two-sided *Only common tests should be described solely by name; describe more complex techniques in the Methods section.* |
| ☒ | ☐ | A description of all covariates tested |
| ☐ | ☒ | A description of any assumptions or corrections, such as tests of normality and adjustment for multiple comparisons |
| ☐ | ☒ | A full description of the statistical parameters including central tendency (e.g. means) or other basic estimates (e.g. regression coefficient) AND variation (e.g. standard deviation) or associated estimates of uncertainty (e.g. confidence intervals) |
| ☐ | ☒ | For null hypothesis testing, the test statistic (e.g. *F*, *t*, *r*) with confidence intervals, effect sizes, degrees of freedom and *P* value noted *Give P values as exact values whenever suitable.* |
| ☒ | ☐ | For Bayesian analysis, information on the choice of priors and Markov chain Monte Carlo settings |
| ☒ | ☐ | For hierarchical and complex designs, identification of the appropriate level for tests and full reporting of outcomes |
| ☒ | ☐ | Estimates of effect sizes (e.g. Cohen's *d*, Pearson's *r*), indicating how they were calculated |

*Our web collection on statistics for biologists contains articles on many of the points above.*

## Software and code

Policy information about availability of computer code

| Data collection | Commercially available software was used to collect data. UVP Gels solo Visionworks 9.1.20063.7760, Raytest Rita TLC control RS232, Raytest Gina Star TLC v.5.01, Agilent OpenLAB CDS ChemStation v2.19.20, Agilent MassHunter WorkStation v10.1, GE DeltaVision Cytiva Softworx, NIS elements AR5.42.0, Licor Odyssey Image Studio5.2.2, Licor acquisition v1.2.0.72. |
|---|---|
| Data analysis | Either open sourced or commercially available softwares were used to collect data; Microsoft Excel for Mac Version 16.94, Graphpad Prism 10, FiJi (ImageJ2 v2.9.0/1.53t), Imaris (CF package, v10.2.0, Oxford Instruments), GE DeltaVision Cytiva Softworx, Nikon (NIS) elements AR Analysis5.42.06 64-bit, BD FACSDiva 8.0.2 |

For manuscripts utilizing custom algorithms or software that are central to the research but not yet described in published literature, software must be made available to editors and reviewers. We strongly encourage code deposition in a community repository (e.g. GitHub). See the Nature Portfolio guidelines for submitting code & software for further information.

## Data

Policy information about availability of data

All manuscripts must include a data availability statement. This statement should provide the following information, where applicable:
- Accession codes, unique identifiers, or web links for publicly available datasets
- A description of any restrictions on data availability
- For clinical datasets or third party data, please ensure that the statement adheres to our policy

Data used in this study are available in the main figures, extended data figures, source data and supplementary tables of this article. Publicly available CRISPR Chronos data set was taken from DepMap.

## Research involving human participants, their data, or biological material

Policy information about studies with human participants or human data. See also policy information about sex, gender (identity/presentation), and sexual orientation and race, ethnicity and racism.

| | |
|---|---|
| Reporting on sex and gender | n/a |
| Reporting on race, ethnicity, or other socially relevant groupings | n/a |
| Population characteristics | n/a |
| Recruitment | n/a |
| Ethics oversight | n/a |

Note that full information on the approval of the study protocol must also be provided in the manuscript.

# Field-specific reporting

Please select the one below that is the best fit for your research. If you are not sure, read the appropriate sections before making your selection.

☒ Life sciences    ☐ Behavioural & social sciences    ☐ Ecological, evolutionary & environmental sciences

For a reference copy of the document with all sections, see nature.com/documents/nr-reporting-summary-flat.pdf

# Life sciences study design

All studies must disclose on these points even when the disclosure is negative.

| | |
|---|---|
| Sample size | No statistical method was used to predetermine sample size. |
| Data exclusions | No data were excluded. |
| Replication | Graphical data and representative immunoblots are from at least three independent replicates. Micrographs are from two to three independent repeats. All attempts at replication were successful. |
| Randomization | No randomization was used as this study does not involve human participants. |
| Blinding | Investigators were not blinded to group allocation. Blinding was not required because the study does not have analysis of data from human participants. |

# Reporting for specific materials, systems and methods

We require information from authors about some types of materials, experimental systems and methods used in many studies. Here, indicate whether each material, system or method listed is relevant to your study. If you are not sure if a list item applies to your research, read the appropriate section before selecting a response.

## Materials & experimental systems

| n/a | Involved in the study |
|---|---|
| ☐ | ☒ Antibodies |
| ☐ | ☒ Eukaryotic cell lines |
| ☒ | ☐ Palaeontology and archaeology |
| ☐ | ☒ Animals and other organisms |
| ☒ | ☐ Clinical data |
| ☒ | ☐ Dual use research of concern |
| ☒ | ☐ Plants |

## Methods

| n/a | Involved in the study |
|---|---|
| ☒ | ☐ ChIP-seq |
| ☐ | ☒ Flow cytometry |
| ☒ | ☐ MRI-based neuroimaging |

## Antibodies

| | |
|---|---|
| Antibodies used | Antibodies used in this study, their description, and dilutions used are listed in supplementary table 2 in the article. |
| Validation | All commercially available antibodies are prevalidated by the manufacturing company. Anti Pma1 antibody was validated in PMID: 29254995 (for Pma1) , and anti Ypt7 antibody was validated by the Dr. William Wickner lab at Dartmouth in PMID: 12177043.

1. CST anti-HA
"Specificity / Sensitivity
HA-Tag (C29F4) Rabbit mAb detects exogenously expressed proteins containing the HA epitope tag. The antibody may cross-react with a protein of unknown origin ~100kDa.

Species Reactivity:

All Species Expected"

2. Thermofisher sientific anti-Porin (459500)

"The antibody was verified by Cell treatment to ensure that the antibody binds to the antigen stated."

3. Thermofisher scientific anti-Dpm1 (A6429)
The manufacturer confirms this antibody is a mouse monoclonal IgG1 (clone number (5C5A7) which reacts to yeast Dpm1p.

4. Abcam anti-Vph1 (10D7A7B2)
The manufacturer confirms that the mouse monoclonal IgG2a antibody (clone number (10D7A7B2) antibody reacts with yeast Vph1p.

5. CST Anti-Caveolin-1 (D46G3)

"Specificity / Sensitivity
Caveolin-1 (D46G3) XP® Rabbit mAb detects endogenous levels of total caveolin-1 protein.

Species Reactivity:

Human, Mouse, Rat, Hamster, Monkey, Bovine, Dog"

6. CST anti-GAPDH antibody (14C10)

"Specificity / Sensitivity
GAPDH (14C10) Rabbit mAb detects endogenous levels of total GAPDH protein.

Species Reactivity:

Human, Mouse, Rat, Monkey, Bovine, Pig"

7. CST Anti-VDAC antibody (D73D12)

"Specificity / Sensitivity
VDAC (D73D12) Rabbit mAb detects endogenous levels of total VDAC protein.

Species Reactivity:

Human, Mouse, Rat, Monkey"

8. CST Anti-PDI antibody (C81H6)

"Specificity / Sensitivity
PDI (C81H6) Rabbit mAb detects endogenous levels of total PDI protein. |

Species Reactivity:

Human, Mouse, Rat, Monkey"

9. IRDye 800CW anti-Rabbit IgG (LI-COR 926-32211)

"Highly cross-adsorbed goat (polyclonal) anti-rabbit IgG (H+L) antibody conjugated to IRDye 800CW.

Immunogen
Rabbit IgG

Purity and Specificity
Isolation of specific antibodies was accomplished by affinity chromatography using pooled rabbit IgG covalently linked to agarose. Based on ELISA and flow cytometry, this antibody reacts with the heavy and light chains of rabbit IgG, and with the light chains of rabbit IgM and IgA. This antibody was tested by dot blot and and/or solid-phase adsorbed for minimal cross-reactivity with human, mouse, rat, sheep, and chicken serum proteins, but may cross-react with immunoglobulins from other species. The conjugate has been specifically tested and qualified for Western blot and In-Cell Western™ Assay applications."

10. IRDye 680RD anti-mouse IgG (LI-COR 926-68070)

"Highly cross-adsorbed goat (polyclonal) anti-mouse IgG (H+L) antibody conjugated to IRDye 680RD.

Immunogen
Mouse IgG paraproteins

Purity and Specificity
Isolation of specific antibodies was accomplished by affinity chromatography using pooled mouse IgG covalently linked to agarose. Based on ELISA and flow cytometry, this antibody reacts with the heavy and light chains of mouse IgG1, IgG2a, IgG2b, and IgG3, and with the light chains of mouse IgM and IgA. This antibody was tested by dot blot and and/or solid-phase adsorbed for minimal cross-reactivity with human, rabbit, goat, rat, and horse serum proteins, but may cross-react with immunoglobulins from other species. The conjugate has been specifically tested and qualified for Western blot and In-Cell Western™ Assay applications."

# Eukaryotic cell lines

Policy information about cell lines and Sex and Gender in Research

| Cell line source(s) | S. cerevisiae cell lines were either collected from the non-essential gene deletion library or constructed in the Prinz lab; Mammalian Cell lines (HeLa, MDA-MB-231, and MCF10A) were purchased from ATCC. HeLa cells with TLCD1 deletion and isogenic untargeted guide RNA transfected containing control HeLa cell lines were a gift from the Kasparas Petkevicius lab in Cambridge University, UK. |
|---|---|
| Authentication | S. cerevisiae cell lines were authenticated by PCR using primers against S. cerevisiae genome, selecting against selection markers, immunoblotting, and crossing . Mammalian cell lines were authenticated by ATCC. Cell lines gifted by the Petkevicius lab were authenticated by their lab by PCR using human genome specific primers. |
| Mycoplasma contamination | Not contaminated. Assessed by MycoAlert Mycoplasma Detection Kit (Lonza Cat# LT07-118). |
| Commonly misidentified lines (See ICLAC register) | Commonly misidentified cell lines were not used in this study |

# Animals and other research organisms

Policy information about studies involving animals; ARRIVE guidelines recommended for reporting animal research, and Sex and Gender in Research

| Laboratory animals | Only animals used in this study is the zebrafish Danio rerio. Details about Zebrafish husbandry is mentioned in the materials and methods section. |
|---|---|

| Wild animals | n/a |
|---|---|
| Reporting on sex | Sex of zebrafish was not considered in this study. |
| Field-collected samples | n/a |
| Ethics oversight | n/a |

Note that full information on the approval of the study protocol must also be provided in the manuscript.

# Plants

| Seed stocks | n/a |
|---|---|
| Novel plant genotypes | n/a |
| Authentication | n/a |

# Flow Cytometry

## Plots

Confirm that:

☒ The axis labels state the marker and fluorochrome used (e.g. CD4-FITC).

☒ The axis scales are clearly visible. Include numbers along axes only for bottom left plot of group (a 'group' is an analysis of identical markers).

☒ All plots are contour plots with outliers or pseudocolor plots.

☒ A numerical value for number of cells or percentage (with statistics) is provided.

## Methodology

| Sample preparation | Mentioned in the materials and methods section. Samples were prepared following the manufacturer's protocol of the Annexin-V apoptosis assay kit (Thermofisher Scientific Cat. No. V13241). |
|---|---|
| Instrument | BD FACS Flow Cytometer |
| Software | BD FACSDiva 8.0.2 |
| Cell population abundance | Cells were from homogenous cultures, unlike mixed cell types from a tissue. Therefore, this field is not applicable. |
| Gating strategy | A schematic figure explaining the gating strategy for the experiment is provided as Extended data Fig. 9. A representative flow cytometry figure is in Extended data fig. 10. |

☒ Tick this box to confirm that a figure exemplifying the gating strategy is provided in the Supplementary Information.

