## [Peer Review File · Nature Cell Biology]

The Vps13-like protein BLTP2 regulates phosphatidylethanolamine levels to maintain plasma membrane fluidity and breast cancer aggressiveness.

Corresponding Author: Dr William Prinz

Version 0:

Decision Letter:

Dear Will,

Thank you for your interest in submitting your work to Nature Cell Biology and for submitting the presubmission inquiry online, it's much appreciated. I also apologize for the delay as I was out of the office this past Friday, and also yesterday for the US holiday.

I have discussed the information you provided with my colleagues, and we think that the study sounds interesting and could be appropriate for this journal. However, given the limited information provided, we would need to evaluate the complete dataset and full manuscript to fully understand the depth of studies and degree of advance, before we can make a decision whether to formally peer review it. I am sorry we can't provide a more definitive answer yet.

For a full evaluation, please use this link to submit the complete manuscript:

Link Redacted

Please feel free to contact me if you have any questions and thank you again for considering NCB for your work,

Kind regards,

Version 1:

Reviewer comments:

Reviewer #1

(Remarks to the Author)

This manuscript addresses the biological function of a bridge-like lipid transport protein, BLTP2. Inspection of the DepMap database reveals three metastatic breast cancer cell lines with high BLTP2 levels and a requirement for this gene, and its knockout slows HeLa cell proliferation. The yeast homolog, Fmp27, leads to cold sensitivity in the absence of ethanolamine, and this is restored by ethanolamine or an orthoganol head group alcohol, propanolamine. Curiously, whereas Fmp27 yeast have reduced PE levels, PE production through the minor CDP-DAG pathway is unaffected. Fmp27 loss prevents cold-induced increases in PE-PC ratios and decreases in fluidity of the PM (but only in the absence of ethanolamine or propanolamine). Similar findings are made in HeLa cells. Curiously, other aspects of homeoviscous adaptation, such as acyl chain remodeling, are unaffected in these cells. Finally, the manuscript demonstrates that in HeLa cells, BLTP2 synergizes with another PE and fluidity regulating pathway, and that loss of BLTP2 decreases a metastatic breast cancer cell lines' proliferation, survival and metastatic potential in a xenograft model.

Overall, the results provide a link between a bridge-like lipid transport protein and PM enrichment of PE in the PM, and suggest a requirement for cold adaptation in yeast and for metastatic cancer progression in humans. This could represent a novel insight into both the function of these still enigmatic protein bridges, the function of PE at the PM, and the physiological requirement for both. However, the reviewer sees several areas where the manuscript would need to be strengthened to fully support these insights.

(1) Requirements for bridge like function at ER:PM contact sites. The data on PE/PC ratios clearly implicate transport of PE via BLTP2/Fmp27. However, currently there is no demonstration that PE transfer activity of BLTP2 is the mechanism. Can a mutant be identified in the protein that maintains protein architecture and ER:PM contact site localization, but is deficient in lipid transport and that fails to rescue Fmp27 cells or BLTP2 KO? For example, inclusion of anionic residues in the hydrophobic lipid binding cavity? Similarly, can a variant be generated that fails to target ER:PM contact sites? This would conclusively demonstrate a requirement for PE transport from ER to PM.

(2) Relationship of BLTP2 to the Kennedy pathway: As the manuscript states on p.9 : "BLTP2 in humans, like its yeast homolog, regulates PM fluidity by increasing the amount of PE in the PM when PE synthesis by the Kennedy pathway is compromised by a lack of exogenous EtN." This confused the reviewer. The reviewer's interpretation of the EtN (and PpN) rescue data is that BLTP2 is required to

increase PM PE:PC ratios when the Kennedy pathway is inactive, i.e. when cells are deficient in PE synthesis. The only way I can square this in my head is if BLTP2 is required to reallocate limiting amounts of PE from ER to PM... This isn't required when PE is abundant and the cell can simply synthesize more. However, since the Kennedy pathway synthesizes PE and PC in the ER membrane, if BLTP2 is primarily functioning as a conduit to traffic PE to the PM, defects should persist even when the Kennedy pathway is active. So why would cells not need BLTP2 in the face of continued PE synthesis? The manuscript does not address this question.

(3) The specific requirements for BLTP2 in metastatic breast cancers: the data do not really support a general role for BLTP2 in this disease. Firstly, in fig. 1, DepMap data identifies three highly invasive breast cancer cell lines with high BLTP2 expression and dependency. However, when the reviewer reproduced this plot on the DepMap site, it was clear that invasive breast carcinoma cell lines (41 in the DepMap database) do not all have high dependency on BLTP2, nor high expression - and there is no obvious correlation between the two. Notably, the MDA-MB-231 cell line employed in figure 5, although having relatively high expression, has a modest survival score (-0.42). Although comparisons are made to the breast epithelial line MCF10A, the generality of a BLTP2 requirement across multiple invasive breast carcinomas cannot really be established from these limited experiments.

There are also some minor, technical amendments that would improve the manuscript:

(4) Figure 1J and associated text: Although the methods clearly states the source of the tritiated serine, and it is L-[3-3H]-Serine, the position of the tritium label should be indicated on the figure, so the reader does not have to search beyond the manuscript for reassurance that the label is not lost during PS metabolism.

(5) Figure 2: the figure legend describes "cells". Although it is clear that these are yeast because Fmp27 is described, the reader might miss this point. The legend would benefit by more explicitly stating that these are experiments on yeast.

(6) In figure 1C, the decrease in PE levels in BLTP2 yeast are clear, but neither the figure nor associated text makes it explicit whether these are EtN replete growth conditions or not.

(7) the axis of figure 5A is missing the superscript on the y-axis label "x 10⁵ cells".

Reviewer #2

(Remarks to the Author)

This study investigates the role of the conserved, Vps13-like lipid transfer protein BLTP2 (Fmp27 in *S. cerevisiae*) in phosphatidylethanolamine (PE)-homeostasis. The authors build a convincing case that Fmp27 is crucial for cell growth at low temperatures and that the growth defects of a Fmp27 deletion are rescued by ethanol supplementation of the medium. The authors show convincingly that the defects in PE regulation of a fmp27D mutant at low temperatures are not due to defects in the de novo biosynthesis of PE. Instead, the subcellular distribution of PE is affected with most significant changes in the plasma membrane. These observations are consistent with a role of Fmp27 as a lipid transfer protein between the ER and the plasma membrane.

Given that the plasma membrane fluidity is affected by FMP27 deletion and rescued by ethanolamine, the authors conclude that Fmp27 may be crucial to provide membrane fluidity in the cold by delivering PE lipids from the ER to the plasma membrane.

Some of the key observations on the important role of BLTP2/Fmp27 are recapitulated in HeLa cells, thereby highlighting that BLTP2 may work in parallel to a second, previously described pathway (TLCD1) that also regulates plasma membrane fluidity. Strikingly, the interference with BLTP2 expression dramatically reduces growth and aggressiveness of the triple-negative breast cancer cell line MDA-MB-231, but not that of a non-cancerous breast epithelial cell line MCF10A.

This study has several strengths making it a very good candidate for the broad, general audience of Nature Cell Biology. It is timely, by establishing the functional and biomedical relevance of a putative lipid transfer protein. It is original and novel. Furthermore, it demonstrates the evolutionary conservation of a machinery that regulates the PE level in the plasma membrane thereby affecting plasma membrane fluidity. Thirdly, the study elegantly connects lipid metabolism, genetic perturbation in yeast and mammalian cell culture, membrane biophysics and xenograft models to approach a challenging problem in cell biology in a holistic fashion. The conclusions are reliable and the manuscript is clearly written. Nevertheless, there are several crucial points that should be addressed before I can recommend this manuscript for publication.

Major point 1 – related to Figure 1: The experiments where choline, ethanolamine or propanolamine are supplemented to the medium look very convincing. It is very important, however, that the effect of the pH is not overlooked in such experiments as ethanolamine and propanolamine are basic compounds, while choline is not. In fact, the pH is an important regulator of cellular stress resistance and condensation. Unfortunately, the manuscript does not contain sufficient information as to whether the medium was buffered and how the supplementation was performed. It is important to repeat some of the key observations (e.g. as demonstrated in Figure 1D, 1G and/or 1I) with a more carefully controlled pH. If the pH of the medium indeed plays an important role on the observed phenotypes, the authors would have to consider more experiments to dissect the contribution of the pH to the observed phenotypes.

Major point 2 – related to the previous point and Materials and Methods: Given the important role of lipids in this study and the important impact of cell cultivation and media composition on the PC-to-PE ratio, the authors should clearly indicate and media composition and their (complete!) cultivation protocol for each experiment.

Major point 3 – related to Figures 2D and 4F: Membrane fluidity in the plasma membrane is measured by two different methods in two different systems, which is fine. However, the authors do not sufficiently discuss the fact that both reporters only incorporate in the outer leaflet of the plasma membrane, while PE is highly enriched in the inner leaflet. Hence, the mechanism how PE regulates membrane fluidity in the outer membrane leaflet remains entirely unclear. If membrane fluidity or other plasma membrane characteristics (e.g. lipid packing defects in the inner leaflet, which may help recruiting certain signaling-active proteins) are the relevant parameters for the cellular phenotypes remains unexplored. Especially in the context of the breast cancer cell line, it may be very interesting to learn, if the recruitment and signaling at the plasma membrane (e.g. by AKT) is affected upon BLTP2 depletion and/or ethanolamine supplementation.

Major point 4 – related to Figure 4A: I am missing a WT control for this experiment.

Major point 5 – related to Figure 4g: The authors conclude that 'we found that the relative abundance of the major phospholipid species

does not change in BLTP2-LO cells'. This is not correct for PE lipids (lipid species involving e.g. 18-1;18-1 are substantially more abundant). Given that PE lipids are at the center-stage of this study, this is a relevant observation, which should be discussed. The authors should provide a bar diagram (showing also the 3 replicates of this measurement). In fact, some repercussions of a perturbed species compositions are also seen in PC lipids. Is there any general difference in the acyl chain composition of PE/PC lipids when comparing yeast and mammalian cells?

Major point 6 – related to selectivity of transport: The authors explain that the basis transport directionality is not understood for any Vps13-like lipid transfer protein. However, the authors do not comment on the specificity/selectivity of transport. How comes that PE lipids are particularly well transported by this machinery? The authors should at least comment on this point.

Minor point 1 – p4: The authors refer to PE lipids as lipids with 'high negative monolayer spontaneous curvature' that 'forms hexagonal phase structures rather than membrane bilayers'. This, however, is only true for PE lipids with two unsaturated lipid acyl chains such as DOPC. The authors should either add more data regarding the lipid species composition of PE or adjust their discussion on the role of PE lipids.

Minor point 2 – related to Figure 1E: The experiment convincingly shows that Fmp27 is much more abundant in cells cultivated at 18°C. However, it also seems that Por1, which was the loading control, is lower. Could the authors comment on this? Does that imply that the mitochondrial abundance is reduced at low temperatures (thereby affecting mitochondrial lipids)?

Minor point 3 – related to Figure 2B: I wonder if there is a problem with the Western blot related to the plasma membrane isolation by density centrifugation. 14 fractions were taken from the gradient and all 14 fractions are shown. However, all four marker proteins show an abrupt change between fraction 7 and 8. Related to this, the vacuole marker Vph1 has two populations. Is this real, were the Western blots accidentally mirrored or are some fractions missing?

Minor point 4 – related to Figure 1G, Figure 3E, Figure 3F and the text on p6 (top): The text in the manuscript is misleading. The authors should clearly indicate that inositol seem limiting for the growth of both WT and fmp27D cells. The authors should also indicate that inositol speeds up growth of the fmp27D quite substantially. The authors should also comment on the different doubling times observed between Figure 1G and 3F.

Given that the UAS-INO system is controlled by PA in the ER membrane (Loewen et al. 2003 and 2004) and given that PA is much more abundant in fmp27D cells (Figure 1H), how comes that there is no 'specific' effect on fmp27D upon inositol supplementation. Can the authors comment?

Minor point 5: The Material and Methods section should be carefully adopted to allow the reader for repeat the experiment. Referring to a previous paper (e.g. as for the plasma membrane 'rip-off' from HeLa cells or for the TMH-DPH experiment) is not recommended.

Despite these criticisms, I very much enjoyed reading the manuscript and strongly believe that it is both highly interesting and highly relevant to the broad readership of NCB. Once the concerns are addressed (which can be done in reasonable amount of time), this manuscript is a very strong candidate for publication in NCB.

Reviewer #3

(Remarks to the Author)

This is an interesting paper that describes the role of BLTP2/Fmp27, a member of the Vps13 family of bridge-like lipid transfer proteins, in regulating plasma membrane fluidity through phosphatidylethanolamine (PE) homeostasis. The experiments reveal unexpected outcomes of disrupting this protein in yeast and cancer cells.

The paper is hard to read as the authors seem to have thrown everything they have at the problem without much selection. The figures are not well organized, and they are aesthetically poor, making reading even more difficult. Also, the paper is primarily descriptive - it identifies an interesting role for BLTP2/Fmp27 but leaves open the question of what makes this pathway work. This question is highlighted in the final paragraph where the authors admit that they do not know what drives PE to the PM and what is the underlying role of this pathway in cancer.

Notes on the figures:

Fig 1a - conventional x-y axes please, with y-axis on the left.

Fig 1b - scale this better so that bar chart is clearly associated with 1b and not 1c (reduce the size of the image to make room for the bar chart!); the bar chart needs a time unit on the y-axis as it represents a rate of proliferation.

Fig 1d- indicate SC medium (the corresponding figure is S1b where YPD is indicated)

Figs 1d, S1b and S1c - these data should be shown together in Fig 1 to make things easier for the reader.

Fig 1e- blot replicates are not necessary to show; single lanes would be fine so that the bar chart can be moved alongside.

Figs 1i and 1k - switch these around to match the order in which they are presented in the text.

Fig 2b - the fractionation quality as seen by the blot seems rather poor. The bar separating fractions 1-7 and 8-14 provides a sharp cut-off of the signals corresponding to all the markers used. This should be done using a 15-well gel or run fractions 5-11 on one gel is the number of wells is limiting.

Figs 3a and 3b- these are black-boxes - they are not explained properly and consequently offer little information. They should be eliminated or moved to a supplementary figure.

Figs 3c and 3d - these could be combined with Fig 2 as the information they provide is consistent with other data shown in Fig 2

Figs 3e and 3f - these could be moved to a supplementary figure.

Fig 4a- this is supposed to indicate a growth rate but there is no time unit.

Other notes:

Fmp stands for 'found in mitochondrial proteome' and a previous paper by the authors (Toulmay et al) described localization of Fmp27 to ER-PM and ER-mito contact sites. The authors should elaborate on this, given the role of mitochondria in PE homeostasis and the uniquely PM-adjacent staining of Fmp27 shown in Fig S1a.

Page 3 - what is 'rate of flux'? Flux represents a time-based measurement, so the word rate is unnecessary.

Page 3- radiolabeling for 60 min does not provide a measure of synthesis rate (or flux??) unless the authors show that this represents a time point within a linear range of continuous incorporation of label, in which case they could represent the data as per min or per hour.

Page 4- it would be interesting to know if the ergosterol content of the PM fraction is altered in the fmp27 cells (ergosterol is a mediator of PM fluidity as the authors demonstrate in Fig 2d).

Page 4- it would also be interesting to know if the fmp27 cells are differently sensitive to duramycin compared with wild-type cells under the various conditions tested.

Page 5, middle para- the comment about the ELSD detector is unnecessary; presumably the authors simply want to make a distinction between a mass measurement of lipids, reflecting actual pool sizes, and a short-term radiolabeling experiment which is not taken to steady state.

Page 6- the section of *C. elegans* seems gratuitous and should be dropped.

Page 7, second para, last line- should be S3g not S3e.

Decision Letter:

*Please delete the link to your author homepage if you wish to forward this email to co-authors.

Dear Will,

Thank you again for submitting your manuscript, "The Vps13-like protein BLTP2 is pro-survival and regulates phosphatidylethanolamine levels in the plasma membrane to maintain its fluidity and function.", to Nature Cell Biology, and I am very sorry for the delay in sharing our decision with you. Your manuscript has now been seen by 3 referees, who are experts in lipid transport (Referee #1); molecular membrane biology (Referee #2); and membrane and lipid biology, yeast (Referee #3). As you will see from their comments (attached below), they find the work of potential interest but have raised substantial concerns that in our view would need to be addressed with considerable revisions before we can consider publication in Nature Cell Biology.

As you may know, as per our standard editorial process, Nature Cell Biology editors discuss the referee reports in detail within the editorial team, including the chief editor, to identify key referee points that should be addressed with priority, and requests that are overruled as being beyond the scope of the current study. To guide the scope of the revisions, I have listed these points below. Our standard revision period is six months and we are committed to providing a fair and constructive peer-review process, so please feel free to contact me if you would like to discuss any of the referee comments further or if you anticipate any delays or issues addressing the reviews.

In our view, the referees' concerns regarding the strength of the mechanism linking BLTP2 to intracellular PE distribution and PM fluidity via a function in lipid transport and their other concerns about the analyses supporting the model are significant and would need to be addressed experimentally to bolster the conclusions. Reconsideration of the study for this journal and re-engagement of referees will depend on the strength of these revisions.

We therefore recommend dedicating efforts in revision to address the points listed below (please see our numbered list below). On the other hand, please note that the analyses linking BLTP2 to cancer biology are not a priority in our view in revision.

1- The reviewers highlighted that the importance of the lipid transfer function of BLTP2 for its impact on PM fluidity needs to be tested conclusively and directly, and we agree:

Rev#1 major point #1

Please also see Rev#2's major points 6 and 3 about the methods used to study membrane fluidity and Rev#3's point in other notes "Page 4- it would be interesting to know.." paragraph

We appreciate that these experiments may be challenging – while we feel that this level of evidence is necessary for reconsideration at NCB, please do feel free to contact us to discuss these experiments. In the event that it would become relevant or of interest (or at any point at your request), we could also contact our sister journals (e.g., Nature Communications, Nature Structural & Molecular Biology, The EMBO Journal) to discuss a potential transfer of the study.

2- The reviewers felt that further clarity was needed about the role of BLTP2 with regards to the activity of the Kennedy pathway and about its impact on the lipid landscape in cells and across organelles/membranes, including because of questions regarding the experimental protocols:

Rev#1 point #2

Rev#2 major point 1-2, 5; minor point 1

Rev#3 other note "Fmp stands for.." paragraph

3- All other referee concerns pertaining to strengthening existing data, providing controls, methodological details, clarifications and textual changes should also be addressed.

4- Finally, please pay close attention to our guidelines on statistical and methodological reporting (listed below) as failure to do so may delay the reconsideration of the revised manuscript. In particular, please provide:

We would be happy to consider a revised manuscript that would satisfactorily address these points, unless a similar paper is published elsewhere or is accepted for publication in Nature Cell Biology in the meantime.

- ensure that it conforms to our format instructions and publication policies (see below and <https://www.nature.com/nature/for-authors>).
- provide a point-by-point rebuttal to the full referee reports verbatim, as provided at the end of this letter.
- provide the completed Reporting Summary (found here <https://www.nature.com/documents/nr-reporting-summary.pdf>). This is essential for reconsideration of the manuscript will be available to editors and referees in the event of peer review. For more information see <http://www.nature.com/authors/policies/availability.html> or contact me.

Nature Cell Biology is committed to improving transparency in authorship. As part of our efforts in this direction, we are now requesting that all authors identified as 'corresponding author' on published papers create and link their Open Researcher and Contributor Identifier (ORCID) with their account on the Manuscript Tracking System (MTS), prior to acceptance. ORCID helps the scientific community achieve unambiguous attribution of all scholarly contributions. You can create and link your ORCID from the home page of the MTS by clicking on 'Modify my Springer Nature account'. For more information please visit www.springernature.com/orcid.

This journal strongly supports public availability of data. Please place the data used in your paper into a public data repository, or alternatively, present the data as Supplementary Information. If data can only be shared on request, please explain why in your Data Availability Statement, and also in the correspondence with your editor. Please note that for some data types, deposition in a public repository is mandatory - more information on our data deposition policies and available repositories appears below.

Link Redacted

We hope that you find our referees' comments and editorial guidance helpful. Please do not hesitate to contact me if there is anything you would like to discuss. Thank you again for considering NCB for your work,

Best wishes,

Reviewers' Comments:

Reviewer #1:

Remarks to the Author:

This manuscript addresses the biological function of a bridge-like lipid transport protein, BLTP2. Inspection of the DepMap database reveals three metastatic breast cancer cell lines with high BLTP2 levels and a requirement for this gene, and its knockout slows HeLa cell proliferation. The yeast homolog, Fmp27, leads to cold sensitivity in the absence of ethanolamine, and this is restored by ethanolamine or an orthoganol head group alcohol, propanolamine. Curiously, whereas Fmp27 yeast have reduced PE levels, PE production through the minor CDP-DAG pathway is unaffected. Fmp27 loss prevents cold-induced increases in PE-PC ratios and decreases in fluidity of the PM (but only in the absence of ethanolamine or propanolamine). Similar findings are made in HeLa cells. Curiously, other aspects of homeoviscous adaptation, such as acyl chain remodeling, are unaffected in these cells. Finally, the manuscript demonstrates that in HeLa cells, BLTP2 synergizes with another PE and fluidity regulating pathway, and that loss of BLTP2 decreases a metastatic breast cancer cell lines' proliferation, survival and metastatic potential in a xenograft model.

Overall, the results provide a link between a bridge-like lipid transport protein and PM enrichment of PE in the PM, and suggest a requirement for cold adaptation in yeast and for metastatic cancer progression in humans. This could represent a novel insight into both the function of these still enigmatic protein bridges, the function of PE at the PM, and the physiological requirement for both. However, the reviewer sees several areas where the manuscript would need to be strengthened to fully support these insights.

(1) Requirements for bridge like function at ER:PM contact sites. The data on PE/PC ratios clearly implicate transport of PE via BLTP2/Fmp27. However, currently there is no demonstration that PE transfer activity of BLTP2 is the mechanism. Can a mutant be identified in the protein that maintains protein architecture and ER:PM contact site localization, but is deficient in lipid transport and that fails to rescue Fmp27 cells or BLTP2 KO? For example, inclusion of anionic residues in the hydrophobic lipid binding cavity? Similarly, can a variant be generated that fails to target ER:PM contact sites? This would conclusively demonstrate a requirement for PE transport from ER to PM.

(2) Relationship of BLTP2 to the Kennedy pathway: As the manuscript states on p.9: "BLTP2 in humans, like its yeast homolog, regulates PM fluidity by increasing the amount of PE in the PM when PE synthesis by the Kennedy pathway is compromised by a lack of exogenous EtN." This confused the reviewer. The reviewer's interpretation of the EtN (and PpN) rescue data is that BLTP2 is required to increase PM PE:PC ratios when the Kennedy pathway is inactive, i.e. when cells are deficient in PE synthesis. The only way I can square this in my head is if BLTP2 is required to reallocate limiting amounts of PE from ER to PM... This isn't required when PE is abundant and the cell can simply synthesize more. However, since the Kennedy pathway synthesizes PE and PC in the ER membrane, if BLTP2 is primarily functioning as a conduit to traffic PE to the PM, defects should persist even when the Kennedy pathway is active. So why would cells not need BLTP2 in the face of continued PE synthesis? The manuscript does not address this question.

(3) The specific requirements for BLTP2 in metastatic breast cancers: the data do not really support a general role for BLTP2 in this disease. Firstly, in fig. 1, DepMap data identifies three highly invasive breast cancer cell lines with high BLTP2 expression and dependency. However, when the reviewer reproduced this plot on the DepMap site, it was clear that invasive breast carcinoma cell lines (41 in the DepMap database) do not all have high dependency on BLTP2, nor high expression - and there is no obvious correlation between the two. Notably, the MDA-MB-231 cell line employed in figure 5, although having relatively high expression, has a modest survival score (-0.42). Although comparisons are made to the breast epithelial line MCF10A, the generality of a BLTP2 requirement across multiple invasive breast carcinomas cannot really be established from these limited experiments.

There are also some minor, technical amendments that would improve the manuscript:

(4) Figure 1J and associate text: Although the methods clearly states the source of the tritiated serine, and it is L-[3-3H]-Serine, the position of the tritium label should be indicated on the figure, so the reader does not have to search beyond the manuscript for reassurance that the label is not lost during PS metabolism.

(5) Figure 2: the figure legend describes "cells". Although it is clear that these are yeast because Fmp27 is described, the reader might miss this point. The legend would benefit by more explicitly stating that these are experiments on yeast.

(6) In figure 1C, the decrease in PE levels in BLTP2 yeast are clear, but neither the figure nor associated text makes it explicit whether these are EtN replete growth conditions or not.

(7) the axis of figure 5A is missing the superscript on the y-axis label "x 10⁵ cells".

Reviewer #2:

Remarks to the Author:

This study investigates the role of the conserved, Vps13-like lipid transfer protein BLTP2 (Fmp27 in *S. cerevisiae*) in phosphatidylethanolamine (PE)-homeostasis. The authors build a convincing case that Fmp27 is crucial for cell growth at low temperatures and that the growth defects of a Fmp27 deletion are rescued by ethanol supplementation of the medium. The authors show convincingly that the defects in PE regulation of a fmp27D mutant at low temperatures are not due to defects in the de novo biosynthesis of PE. Instead, the subcellular distribution of PE is affected with most significant changes in the plasma membrane. These observations are consistent with a role of Fmp27 as a lipid transfer protein between the ER and the plasma membrane.

Given that the plasma membrane fluidity is affected by FMP27 deletion and rescued by ethanolamine, the authors conclude that Fmp27 may be crucial to provide membrane fluidity in the cold by delivering PE lipids from the ER to the plasma membrane.

Some of the key observations on the important role of BLTP2/Fmp27 are recapitulated in HeLa cells, thereby highlighting that BLTP2 may work in parallel to a second, previously described pathway (TLCD1) that also regulates plasma membrane fluidity. Strikingly, the interference with BLTP2 expression dramatically reduces growth and aggressiveness of the triple-negative breast cancer cell line MDA-MB-231, but not that of a non-cancerous breast epithelial cell line MCF10A.

This study has several strengths making it a very good candidate for the broad, general audience of Nature Cell Biology. It is timely, by establishing the functional and biomedical relevance of a putative lipid transfer protein. It is original and novel. Furthermore, it demonstrates the evolutionary conservation of a machinery that regulates the PE level in the plasma membrane thereby affecting plasma membrane fluidity. Thirdly, the study elegantly connects lipid metabolism, genetic perturbation in yeast and mammalian cell culture, membrane biophysics and xenograft models to approach a challenging problem in cell biology in a holistic fashion. The conclusions are reliable and the manuscript is clearly written. Nevertheless, there are several crucial points that should be addressed before I can recommend this manuscript for publication.

Major point 1 – related to Figure 1: The experiments where choline, ethanolamine or propanolamine are supplemented to the medium look very convincing. It is very important, however, that the effect of the pH is not overlooked in such experiments as ethanolamine and propanolamine are basic compounds, while choline is not. In fact, the pH is an important regulator of cellular stress resistance and condensation. Unfortunately, the manuscript does not contain sufficient information as to whether the medium was buffered and how the supplementation was performed. It is important to repeat some of the key observations (e.g. as demonstrated in Figure 1D, 1G and/or 1I) with a more carefully controlled pH. If the pH of the medium indeed plays an important role on the observed phenotypes, the authors would have to consider more experiments to dissect the contribution of the pH to the observed phenotypes.

Major point 2 – related to the previous point and Materials and Methods: Given the important role of lipids in this study and the important impact of cell cultivation and media composition on the PC-to-PE ratio, the authors should clearly indicate and media composition and their (complete!) cultivation protocol for each experiment.

Major point 3 – related to Figures 2D and 4F: Membrane fluidity in the plasma membrane is measured by two different methods in two different systems, which is fine. However, the authors do not sufficiently discuss the fact that both reporters only incorporate in the outer leaflet of the plasma membrane, while PE is highly enriched in the inner leaflet. Hence, the mechanism how PE regulates membrane fluidity in the outer membrane leaflet remains entirely unclear. If membrane fluidity or other plasma membrane characteristics (e.g. lipid packing defects in the inner leaflet, which may help recruiting certain signaling-active proteins) are the relevant parameters for the cellular phenotypes remains unexplored. Especially in the context of the breast cancer cell line, it may be very interesting to learn, if the recruitment and signaling at the plasma membrane (e.g. by AKT) is affected upon BLTP2 depletion and/or ethanolamine supplementation.

Major point 4 – related to Figure 4A: I am missing a WT control for this experiment.

Major point 5 – related to Figure 4 g: The authors conclude that 'we found that the relative abundance of the major phospholipid species does not change in BLTP2-LO cells'. This is not correct for PE lipids (lipid species involving e.g. 18-1;18-1 are substantially more abundant). Given that PE lipids are at the center-stage of this study, this is a relevant observation, which should be discussed. The authors should provide a bar diagram (showing also the 3 replicates of this measurement). In fact, some repercussions of a perturbed species compositions are also seen in PC lipids. Is there any general difference in the acyl chain composition of PE/PC lipids when comparing yeast and mammalian cells?

Major point 6 – related to selectivity of transport: The authors explain that the basis transport directionality is not understood for any Vps13-like lipid transfer protein. However, the authors do not comment on the specificity/selectivity of transport. How comes that PE lipids are particularly well transported by this machinery? The authors should at least comment on this point.

Minor point 1 – p4: The authors refer to PE lipids as lipids with 'high negative monolayer spontaneous curvature' that 'forms hexagonal phase structures rather than membrane bilayers'. This, however, is only true for PE lipids with two unsaturated lipid acyl chains such as DOPC. The authors should either add more data regarding the lipid species composition of PE or adjust their discussion on the role of PE lipids.

Minor point 2 – related to Figure 1E: The experiment convincingly shows that Fmp27 is much more abundant in cells cultivated at 18°C. However, it also seems that Por1, which was the loading control, is lower. Could the authors comment on this? Does that imply that the mitochondrial abundance is reduced at low temperatures (thereby affecting mitochondrial lipids)?

Minor point 3 – related to Figure 2B: I wonder if there is a problem with the Western blot related to the plasma membrane isolation by density centrifugation. 14 fractions were taken from the gradient and all 14 fractions are shown. However, all four marker proteins show an abrupt change between fraction 7 and 8. Related to this, the vacuole marker Vph1 has two populations. Is this real, were the Western blots accidentally mirrored or are some fractions missing?

Minor point 4 – related to Figure 1G, Figure 3E, Figure 3F and the text on p6 (top): The text in the manuscript is misleading. The authors should clearly indicate that inositol seem limiting for the growth of both WT and fmp27D cells. The authors should also indicate that inositol speeds up growth of the fmp27D quite substantially. The authors should also comment on the different doubling times observed between Figure 1G and 3F.

Given that the UAS-INO system is controlled by PA in the ER membrane (Loewen et al. 2003 and 2004) and given that PA is much more abundant in fmp27D cells (Figure 1H), how comes that there is no 'specific' effect on fmp27D upon inositol supplementation. Can the authors comment?

Minor point 5: The Material and Methods section should be carefully adopted to allow the reader for repeat the experiment. Referring to a previous paper (e.g. as for the plasma membrane 'rip-off' from HeLa cells or for the TMH-DPH experiment) is not recommended.

Despite these criticisms, I very much enjoyed reading the manuscript and strongly believe that it is both highly interesting and highly relevant to the broad readership of NCB. Once the concerns are addressed (which can be done in reasonable amount of time), this manuscript is a very strong candidate for publication in NCB.

Reviewer #3:

Remarks to the Author:

This is an interesting paper that describes the role of BLTP2/Fmp27, a member of the Vps13 family of bridge-like lipid transfer proteins, in regulating plasma membrane fluidity through phosphatidylethanolamine (PE) homeostasis. The experiments reveal unexpected outcomes of disrupting this protein in yeast and cancer cells.

The paper is hard to read as the authors seem to have thrown everything they have at the problem without much selection. The figures are not well organized, and they are aesthetically poor, making reading even more difficult. Also, the paper is primarily descriptive - it identifies an interesting role for BLTP2/Fmp27 but leaves open the question of what makes this pathway work. This question is highlighted in the final paragraph where the authors admit that they do not know what drives PE to the PM and what is the underlying role of this pathway in cancer.

Notes on the figures:

Fig 1a - conventional x-y axes please, with y-axis on the left.

Fig 1b - scale this better so that bar chart is clearly associated with 1b and not 1c (reduce the size of the image to make room for the bar chart!); the bar chart needs a time unit on the y-axis as it represents a rate of proliferation.

Fig 1d- indicate SC medium (the corresponding figure is S1b where YPD is indicated)

Figs 1d, S1b and S1c - these data should be shown together in Fig 1 to make things easier for the reader.

Fig 1e- blot replicates are not necessary to show; single lanes would be fine so that the bar chart can be moved alongside.

Figs 1i and 1k - switch these around to match the order in which they are presented in the text.

Fig 2b - the fractionation quality as seen by the blot seems rather poor. The bar separating fractions 1-7 and 8-14 provides a sharp cut-off of the signals corresponding to all the markers used. This should be done using a 15-well gel or run fractions 5-11 on one gel if the number of wells is limiting.

Figs 3a and 3b- these are black-boxes - they are not explained properly and consequently offer little information. They should be eliminated or moved to a supplementary figure.

Figs 3c and 3d - these could be combined with Fig 2 as the information they provide is consistent with other data shown in Fig 2

Figs 3e and 3f - these could be moved to a supplementary figure.

Fig 4a- this is supposed to indicate a growth rate but there is no time unit.

Other notes:

Fmp stands for 'found in mitochondrial proteome' and a previous paper by the authors (Toulmay et al) described localization of Fmp27 to ER-PM and ER-mito contact sites. The authors should elaborate on this, given the role of mitochondria in PE homeostasis and the uniquely PM-adjacent staining of Fmp27 shown in Fig S1a.

Page 3 - what is 'rate of flux'? Flux represents a time-based measurement, so the word rate is unnecessary.

Page 3- radiolabeling for 60 min does not provide a measure of synthesis rate (or flux??) unless the authors show that this represents a time point within a linear range of continuous incorporation of label, in which case they could represent the data as per min or per hour.

Page 4- it would be interesting to know if the ergosterol content of the PM fraction is altered in the fmp27 cells (ergosterol is a mediator of PM fluidity as the authors demonstrate in Fig 2d).

Page 4- it would also be interesting to know if the fmp27 cells are differently sensitive to duramycin compared with wild-type cells under the various conditions tested.

Page 5, middle para- the comment about the ELSD detector is unnecessary; presumably the authors simply want to make a distinction between a mass measurement of lipids, reflecting actual pool sizes, and a short-term radiolabeling experiment which is not taken to steady state.

Page 6- the section of *C. elegans* seems gratuitous and should be dropped.

Page 7, second para, last line- should be S3g not S3e.

FINANCIAL AND NON-FINANCIAL COMPETING INTERESTS – the authors must include one of three declarations: (1) that they have no financial and non-financial competing interests; (2) that they have financial and non-financial competing interests; or (3) that they decline to respond, after the Author Contributions section. This statement will be published with the article, and in cases where financial

and non-financial competing interests are declared, these will be itemized in a web supplement to the article. For further details please see <https://www.nature.com/licenceforms/nrg/competing-interests.pdf>.

Methods should be written concisely, but should contain all elements necessary to allow interpretation and replication of the results. As a guideline, Methods sections typically do not exceed 3,000 words. The Methods should be divided into subsections listing reagents and techniques. When citing previous methods, accurate references should be provided and any alterations should be noted. Information must be provided about: antibody dilutions, company names, catalogue numbers and clone numbers for monoclonal antibodies; sequences of RNAi and cDNA probes/primers or company names and catalogue numbers if reagents are commercial; cell line names, sources and information on cell line identity and authentication. Animal studies and experiments involving human subjects must be reported in detail, identifying the committees approving the protocols. For studies involving human subjects/samples, a statement must be included confirming that informed consent was obtained. Statistical analyses and information on the reproducibility of experimental results should be provided in a section titled "Statistics and Reproducibility".

All Nature Cell Biology manuscripts submitted on or after March 21 2016 must include a Data availability statement as a separate section after Methods but before references, under the heading "Data Availability". For Springer Nature policies on data availability see <http://www.nature.com/authors/policies/availability.html>; for more information on this particular policy see <http://www.nature.com/authors/policies/data/data-availability-statements-data-citations.pdf>. The Data availability statement should include:

- Accession codes for primary datasets (generated during the study under consideration and designated as "primary accessions") and secondary datasets (published datasets reanalysed during the study under consideration, designated as "referenced accessions"). For primary accessions data should be made public to coincide with publication of the manuscript. A list of data types for which submission to community-endorsed public repositories is mandated (including sequence, structure, microarray, deep sequencing data) can be found here <http://www.nature.com/authors/policies/availability.html#data>.
- Unique identifiers (accession codes, DOIs or other unique persistent identifier) and hyperlinks for datasets deposited in an approved repository, but for which data deposition is not mandated (see here for details <http://www.nature.com/sdata/data-policies/repositories>).
- At a minimum, please include a statement confirming that all relevant data are available from the authors, and/or are included with the manuscript (e.g. as source data or supplementary information), listing which data are included (e.g. by figure panels and data types) and mentioning any restrictions on availability.
- If a dataset has a Digital Object Identifier (DOI) as its unique identifier, we strongly encourage including this in the Reference list and citing the dataset in the Methods.

We recommend that you upload the step-by-step protocols used in this manuscript to the Protocol Exchange. More details can be found at www.nature.com/protocolexchange/about.

All imaging data should be accompanied by scale bars, which should be defined in the legend.

Cropped images of gels/blots are acceptable, but need to be accompanied by size markers, and to retain visible background signal within the linear range (i.e. should not be saturated). The boundaries of panels with low background have to be demarked with black lines. Splicing of panels should only be considered if unavoidable, and must be clearly marked on the figure, and noted in the legend with a statement on whether the samples were obtained and processed simultaneously. Quantitative comparisons between samples on different gels/blots are discouraged; if this is unavoidable, it should only be performed for samples derived from the same experiment with gels/blots were processed in parallel, which needs to be stated in the legend.

The total number of Supplementary Figures (not including the "unprocessed scans" Supplementary Figure) should not exceed the number of main display items (figures and/or tables (see our Guide to Authors and March 2012 editorial <http://www.nature.com/ncb/authors/submit/index.html#suppinfo>; <http://www.nature.com/ncb/journal/v14/n3/index.html#ed>). No restrictions apply to Supplementary Tables or Videos, but we advise authors to be selective in including supplemental data.

GUIDELINES FOR EXPERIMENTAL AND STATISTICAL REPORTING

REPORTING REQUIREMENTS – We are trying to improve the quality of methods and statistics reporting in our papers. To that end, we are now asking authors to complete a reporting summary that collects information on experimental design and reagents. The Reporting Summary can be found here <https://www.nature.com/documents/nr-reporting-summary.pdf> <https://www.nature.com/documents/nr-reporting-summary.pdf> If you would like to reference the guidance text as you complete the template, please access these flattened versions at <http://www.nature.com/authors/policies/availability.html>

STATISTICS – Wherever statistics have been derived the legend needs to provide the n number (i.e. the sample size used to derive statistics) as a precise value (not a range), and define what this value represents. Error bars need to be defined in the legends (e.g. SD, SEM) together with a measure of centre (e.g. mean, median). Box plots need to be defined in terms of minima, maxima, centre, and percentiles. Ranges are more appropriate than standard errors for small data sets. Wherever statistical significance has been derived, precise p values need to be provided and the statistical test used needs to be stated in the legend. Statistics such as error bars must not

be derived from $n < 3$. For sample sizes of $n < 5$ please plot the individual data points rather than providing bar graphs. Deriving statistics from technical replicate samples, rather than biological replicates is strongly discouraged. Wherever statistical significance has been derived, precise p values need to be provided and the statistical test stated in the legend.

Version 2:

Reviewer comments:

Reviewer #1

(Remarks to the Author)

The reviewer appreciates the thorough response to both my own and the other reviewer's comments. I believe the major concerns with the original manuscript are now sufficiently addressed to warrant publication in the current form, pending minor revisions. In response to my major concerns, I only have the following minor comments:

(1) The reviewer hugely appreciates the elegant experiments added in Fig. 3e, h, and I to address the PE transport function of Fmp27. This indeed addresses my major concern that PE transport is truly required for the phenotypes reported elsewhere, and I believe strengthens the manuscript greatly.

(2) I also agree with the author's argument that Fmp27 becomes critical after inhibition of the Kennedy pathway because when it is active, other pathways can probably deliver sufficient PE to the PM, albeit inefficiently. My only minor quibble is can the authors suggest the nature of this pathway? I.e. other lipid transporters (E-Syts, etc.) and, of course, vesicular transport which likely shuffles these abundant phospholipids around.

(3) To my concern about the generality of BLTP2 requirements for breast cancer cell line growth, I can only say: Touché! The reviewer is convinced by the data included in Rebuttal figure 1, as should anyone who has ever prescribed trastuzumab to a cancer patient. I would suggest including these data and pointing them out in the manuscript, perhaps in the supplement.

Reviewer #2

(Remarks to the Author)

The authors have done a good job in addressing the points raised by the reviewers. Especially the experiment of redirecting Fmp27 is elegant. However, there are two points that I feel should be addressed:

With respect to major point 3, the authors state in the text on page 5, para 1 'While PE is primarily localized to the inner leaflet of the PM26, there is evidence that the composition of the inner leaflet of the PM affects the fluidity of the outer leaflet27 28, 29.' (missing comma between 27 and 28). Importantly, Reference 28 describes interleaflet coupling for non-fluid bilayers and states: 'That is, the structure of fluid membranes is dominated by layer-specific membrane properties and is not influenced by that of the opposing leaflet.' Hence, they provide evidence against interleaflet coupling in living (fluid) membranes. The authors should consider rephrasing this section.

With respect to minor point 3 (Western Blots), I am still puzzled by the abrupt change in marker intensity, which is also not described in the references referred to by the authors. I also do not find evidence in the referred papers for two populations of a vacuolar marker. I trust the authors that their protocol enriches the plasma membrane sufficiently, but I remain confused about whether the source data and the data in Fig 2b fit together. I wonder if there were some peculiarities in the fraction experiment, which are not being described.

Specifically, I still have a hard time seeing how the data in Fig. 2B and the source data for Fig. 2b are from the same experiment indeed. I do not exclude the possibility that things are okay, but if were the case, there must have been numerous image processing steps involved and some of them rather extreme and potentially unevenly applied to the different blots. I would recommend using either less processed images for Fig 2B, or indicating the individual processing steps (step-by-step from source data to presented data).

Confusing observations are for example: The Pma1 blot appears as a double band in the source data, only one band is shown in 2B. For Vph1, the band intensity for fractions 1-7 is much higher in Fig 2b compared to the source data, but the band intensity for fractions 10-14 barely changes. The background for the bands 1-7 and 8-14 is grayish, but white in Fig 2b. The relative band intensity for Dpm1 compared to the background signal appears higher than observable in the source data.

Given that reviewer 3 was also irritated by the Western Blot data, I recommend the authors provide a more transparent view on how the data were processed or present less processed data. Also, they should comment on the distribution of band intensities in the manuscript in order to prevent confusing/irritating the reader.

Other than that: An excellent manuscript!

Reviewer #3

(Remarks to the Author)

The paper is much clearer, and the authors have responded for the most part to my comments and those of the other reviewers. I have some further notes:

1. I am still not happy with the western blot in Fig 2b, and these data were also flagged by reviewer 2. It may be that these are standard experiments, but the data quality is pretty poor. This is an important technical step as it defines the PE increase, reflected in the PE/PC ratio shown in Fig. 2c. Perhaps the authors could re-assess their PM fraction in comparison with homogenate, in the same way that they assess the immuno-isolated ER fraction in Fig 2e and mito fraction in Fig 3h. They could also measure ergosterol/phospholipid in the PM fraction versus fractions 2-7, to see if it is high as expected.

2. The authors comment briefly that specific methodologies account for the different PE:PC ratio of total cell lipid extracts shown in Fig 2a versus Fig 2f, where the ratio (at 18C for WT cells) is >1 (Fig 2a) versus approx 0.5 (Fig 2f). These are very different ratios with significant implications for membrane structure, and for the overall thesis of this paper, so what ratio value should one be thinking about? I guess that a PE/PC of approx 0.5 would be typically of most cell membranes.

3. Figure 3a and 3b remain baffling. My comment that these were 'black boxes' was to imply that they did not have any useful or readable content – the authors assumed that I meant the black outlines that circumscribed the graphs and simply removed these outlines! Quite humorous, actually. The figure legend states: Volcano plots showing log fold changes (Log₂ FC) of PL species against statistical significance (-Log p-value). What PLs are being looked at? What do the symbols mean?

4. Figs 3e-i are a useful addition to the paper!

Decision Letter:

*Please delete the link to your author homepage if you wish to forward this email to co-authors.

Dear Will,

Your manuscript, "The Vps13-like protein BLTP2 is pro-survival and regulates phosphatidylethanolamine levels in the plasma membrane to maintain its fluidity and function.", has now been seen by the original reviewers, and I am sorry for the delay in sharing our decision given the end-of-the-year holidays.

As you will see from their comments (attached below), the reviewers continue to find the work of interest and appreciated the revisions but have raised some important persisting points. Although we are also very interested in this study, we believe that their concerns should be addressed before we can consider publication in Nature Cell Biology.

As per our standard process, we have discussed their comments within the editorial team. We are committed to providing a fair and constructive peer-review process, so please feel free to contact me if you would like to discuss any of the referee comments further. We strive to limit our manuscripts to a single round of major experimental revision, to limit the overall time spent in peer review, and in this case, given the overall interest and support, and as the remaining points are relatively minor, we are open to a final round of minor revision.

You will see that an important persisting concern shared by Revs#2 and #3 relates to the data in 2b - neither reviewer finds the data sufficiently clear and compelling. At this stage, we feel it will be important to follow Rev#3's suggestion to include stronger data (either from existing experimental replicates or new experiments) and additional data as needed to provide convincing evidence. We will seek reviewer input on the revision and will need the reviewers to find the data of high quality and convincing to move forward with publication. In addition, please clearly detail the image processing steps in the manuscript text and rebuttal.

Please address the remaining points from all reviewers as best you can.

Additionally, to save future revision efforts, we ask that you please already make changes to the formatting to meet what we would require at acceptance:

1- You have written the manuscript as a Letter. NCB stopped publishing Letters in 2024; with apologies for this, please convert the manuscript to an Article. This means that the manuscript should have an un-referenced abstract, an Introduction section, followed by a Results section divided in subsections (with headers < 60 characters including spaces) and lastly a Discussion section.

2- The supplementary Tables should be provided as Excel files, not in the word document as they are currently. Please note NCB Articles do not have main tables - they can only have Supplementary Tables. All supplementary tables should be provided within 1 excel workbook, as one supplementary table per tab.

3- Please submit each figure file separately (not in the Word file) and in the legends and text, please name and refer to ED figs as "Extended Data Figure X", not "Extended Figure X".

4- "Source Data" figures should be provided in separate source data files, not in the main text file. It is important that you please provide the most uncropped versions of gels that you have. Source data should not be processed or cropped images.

5- Methods-only references should be included in a separate reference list that follows the Methods section. It should be numbered continuously from the main reference list (e.g., if the references in the article end at 50, the Methods-only references should start at 51).

6- The gating strategy panel should be part of an Extended Data figure (Articles can have up to 8 main and 10 Ed figures so there is space). This is in the interest of accessibility for readers. We only allow Supplementary figures when the main and ED figures are all full.

Finally, as before, please pay close attention to our guidelines on statistical and methodological reporting (listed below) as failure to do so may delay the reconsideration of the revised manuscript. In particular, please provide with your revision:

We therefore invite you to take these points into account when revising the manuscript. In addition, when preparing the revision please:

- ensure that it conforms to our format instructions and publication policies (see below and www.nature.com/nature/authors/).

- provide a point-by-point rebuttal to the full referee reports verbatim, as provided at the end of this letter.

- provide the completed Editorial Policy Checklist (found here <https://www.nature.com/authors/policies/Policy.pdf>), and Reporting Summary (found here <https://www.nature.com/authors/policies/ReportingSummary.pdf>). This is essential for reconsideration of the manuscript and these documents will be available to editors and referees in the event of peer review. For more information see <http://www.nature.com/authors/policies/availability.html> or contact me.

Nature Cell Biology is committed to improving transparency in authorship. As part of our efforts in this direction, we are now requesting that all authors identified as 'corresponding author' on published papers create and link their Open Researcher and Contributor Identifier (ORCID) with their account on the Manuscript Tracking System (MTS), prior to acceptance. ORCID helps the scientific community achieve unambiguous attribution of all scholarly contributions. You can create and link your ORCID from the home page of the MTS by clicking on 'Modify my Springer Nature account'. For more information please visit <http://www.springernature.com/orcid>.

Link Redacted

We would like to receive the revision within four weeks. If submitted within this time period, reconsideration of the revised manuscript will not be affected by related studies published elsewhere, or accepted for publication in Nature Cell Biology in the meantime. We would be happy to consider a revision even after this timeframe, but in that case we will consider the published literature at the time of resubmission when assessing the file.

We hope that you will find our referees' comments and editorial guidance helpful. Please do not hesitate to contact me if there is anything you would like to discuss. Thank you again for your efforts in revision and for considering the journal for your work.

Best wishes,

Reviewers' Comments:

Reviewer #1 (Remarks to the Author):

The reviewer appreciates the thorough response to both my own and the other reviewer's comments. I believe the major concerns with the original manuscript are now sufficiently addressed to warrant publication in the current form, pending minor revisions. In response to my major concerns, I only have the following minor comments:

(1) The reviewer hugely appreciates the elegant experiments added in Fig. 3e, h, and I to address the PE transport function of Fmp27. This indeed addresses my major concern that PE transport is truly required for the phenotypes reported elsewhere, and I believe strengthens the manuscript greatly.

(2) I also agree with the author's argument that Fmp27 becomes critical after inhibition of the Kennedy pathway because when it is active, other pathways can probably deliver sufficient PE to the PM, albeit inefficiently. My only minor quibble is can the authors suggest the nature of this pathway? I.e. other lipid transporters (E-Syts, etc.) and, of course, vesicular transport which likely shuffles these abundant phospholipids around.

(3) To my concern about the generality of BLTP2 requirements for breast cancer cell line growth, I can only say: Touché! The reviewer is convinced by the data included in Rebuttal figure 1, as should anyone who has ever prescribed trastuzumab to a cancer patient. I would suggest including these data and pointing them out in the manuscript, perhaps in the supplement.

Reviewer #2 (Remarks to the Author):

The authors have done a good job in addressing the points raised by the reviewers. Especially the experiment of redirecting Fmp27 is elegant. However, there are two points that I feel should be addressed:

With respect to major point 3, the authors state in the text on page 5, para 1 'While PE is primarily localized to the inner leaflet of the PM26, there is evidence that the composition of the inner leaflet of the PM affects the fluidity of the outer leaflet^{27, 28, 29.}' (missing comma between 27 and 28). Importantly, Reference 28 describes interleaflet coupling for non-fluid bilayers and states: 'That is, the structure of fluid membranes is dominated by layer-specific membrane properties and is not influenced by that of the opposing leaflet.' Hence, they provide evidence against interleaflet coupling in living (fluid) membranes. The authors should consider rephrasing this section.

With respect to minor point 3 (Western Blots), I am still puzzled by the abrupt change in marker intensity, which is also not described in the references referred to by the authors. I also do not find evidence in the referred papers for two populations of a vacuolar marker. I trust the authors that their protocol enriches the plasma membrane sufficiently, but I remain confused about whether the source data and the data in Fig 2b fit together. I wonder if there were some peculiarities in the fraction experiment, which are not being described.

Specifically, I still have a hard time seeing how the data in Fig. 2B and the source data for Fig. 2b are from the same experiment indeed. I do not exclude the possibility that things are okay, but if were the case, there must have been numerous image processing steps involved and some of them rather extreme and potentially unevenly applied to the different blots. I would recommend using either less processed images for Fig 2B, or indicating the individual processing steps (step-by-step from source data to presented data).

Confusing observations are for example: The Pma1 blot appears as a double band in the source data, only one band is shown in 2B. For Vph1, the band intensity for fractions 1-7 is much higher in Fig 2b compared to the source data, but the band intensity for fractions 10-14 barely changes. The background for the bands 1-7 and 8-14 is grayish, but white in Fig 2b. The relative band intensity for Dpm1 compared to the background signal appears higher than observable in the source data.

Given that reviewer 3 was also irritated by the Western Blot data, I recommend the authors provide a more transparent view on how the data were processed or present less processed data. Also, they should comment on the distribution of band intensities in the manuscript in order to prevent confusing/irritating the reader.

Other than that: An excellent manuscript!

Reviewer #3 (Remarks to the Author):

The paper is much clearer, and the authors have responded for the most part to my comments and those of the other reviewers. I have some further notes:

1. I am still not happy with the western blot in Fig 2b, and these data were also flagged by reviewer 2. It may be that these are standard experiments, but the data quality is pretty poor. This is an important technical step as it defines the PE increase, reflected in the PE/PC ratio shown in Fig. 2c. Perhaps the authors could re-assess their PM fraction in comparison with homogenate, in the same way that they assess the immuno-isolated ER fraction in Fig 2e and mito fraction in Fig 3h. They could also measure ergosterol/phospholipid in the PM fraction versus fractions 2-7, to see if it is high as expected.

2. The authors comment briefly that specific methodologies account for the different PE:PC ratio of total cell lipid extracts shown in Fig 2a versus Fig 2f, where the ratio (at 18C for WT cells) is >1 (Fig 2a) versus approx 0.5 (Fig 2f). These are very different ratios with significant implications for membrane structure, and for the overall thesis of this paper, so what ratio value should one be thinking about? I guess that a PE/PC of approx 0.5 would be typically of most cell membranes.

3. Figure 3a and 3b remain baffling. My comment that these were 'black boxes' was to imply that they did not have any useful or readable content – the authors assumed that I meant the black outlines that circumscribed the graphs and simply removed these outlines! Quite humorous, actually. The figure legend states: Volcano plots showing log fold changes (Log₂ FC) of PL species against statistical significance (-Log p-value). What PLs are being looked at? What do the symbols mean?

4. Figs 3e-i are a useful addition to the paper!

GUIDELINES FOR SUBMISSION OF NATURE CELL BIOLOGY ARTICLES

ARTICLE FORMAT

ABSTRACT – should not exceed 150 words and should be unreferenced. This paragraph is the most visible part of the paper and should briefly outline the background and rationale for the work, and accurately summarize the main results and conclusions. Key genes, proteins and organisms should be specified to ensure discoverability of the paper in online searches.

TEXT – the main text consists of the Introduction, Results, and Discussion sections and must not exceed 3500 words including the abstract. The Introduction should expand on the background relating to the work. The Results should be divided in subsections with subheadings, and should provide a concise and accurate description of the experimental findings. The Discussion should expand on the findings and their implications. All relevant primary literature should be cited, in particular when discussing the background and specific findings.

REFERENCES – are limited to a total of 70 in the main text and Methods combined,. They must be numbered sequentially as they appear in the main text, tables and figure legends and Methods and must follow the precise style of Nature Cell Biology references. References only cited in the Methods should be numbered consecutively following the last reference cited in the main text. References only associated with Supplementary Information (e.g. in supplementary legends) do not count toward the total reference limit and do not need to be cited in numerical continuity with references in the main text. Only published papers can be cited, and each publication cited should be included in the numbered reference list, which should include the manuscript titles. Footnotes are not permitted.

Methods should be written concisely, but should contain all elements necessary to allow interpretation and replication of the results. As a guideline, Methods sections typically do not exceed 3,000 words. The Methods should be divided into subsections listing reagents and techniques. When citing previous methods, accurate references should be provided and any alterations should be noted. Information must be provided about: antibody dilutions, company names, catalogue numbers and clone numbers for monoclonal antibodies; sequences of RNAi and cDNA probes/primers or company names and catalogue numbers if reagents are commercial; cell line names, sources and information on cell line identity and authentication. Animal studies and experiments involving human subjects must be reported in detail, identifying the committees approving the protocols. For studies involving human subjects/samples, a statement must be included confirming that informed consent was obtained. Statistical analyses and information on the reproducibility of experimental results should be provided in a section titled "Statistics and Reproducibility".

All Nature Cell Biology manuscripts submitted on or after March 21 2016, must include a Data availability statement as a separate section after Methods but before references, under the heading "Data Availability". For Springer Nature policies on data availability see <http://www.nature.com/authors/policies/availability.html>; for more information on this particular policy see <http://www.nature.com/authors/policies/data/data-availability-statements-data-citations.pdf>. The Data availability statement should include:

- Accession codes for primary datasets (generated during the study under consideration and designated as "primary accessions") and secondary datasets (published datasets reanalysed during the study under consideration, designated as "referenced accessions"). For primary accessions data should be made public to coincide with publication of the manuscript. A list of data types for which submission to community-endorsed public repositories is mandated (including sequence, structure, microarray, deep sequencing data) can be found here <http://www.nature.com/authors/policies/availability.html#data>.
- Unique identifiers (accession codes, DOIs or other unique persistent identifier) and hyperlinks for datasets deposited in an approved repository, but for which data deposition is not mandated (see here for details <http://www.nature.com/sdata/data-policies/repositories>).
- At a minimum, please include a statement confirming that all relevant data are available from the authors, and/or are included with the manuscript (e.g. as source data or supplementary information), listing which data are included (e.g. by figure panels and data types) and mentioning any restrictions on availability.
- If a dataset has a Digital Object Identifier (DOI) as its unique identifier, we strongly encourage including this in the Reference list and citing the dataset in the Methods.

We recommend that you upload the step-by-step protocols used in this manuscript to [protocols.io](https://www.protocols.io). More details can be found at <https://www.protocols.io/help/publish-articles>.

DISPLAY ITEMS – main display items are limited to 6-8 main figures and/or main tables. For Supplementary Information see below.

FIGURES – Colour figure publication costs \$395 per colour figure. All panels of a multi-panel figure must be logically connected and arranged as they would appear in the final version. Unnecessary figures and figure panels should be avoided (e.g. data presented in small tables could be stated briefly in the text instead).

All imaging data should be accompanied by scale bars, which should be defined in the legend.

Cropped images of gels/blots are acceptable, but need to be accompanied by size markers, and to retain visible background signal within the linear range (i.e. should not be saturated). The boundaries of panels with low background have to be demarked with black lines. Splicing of panels should only be considered if unavoidable, and must be clearly marked on the figure, and noted in the legend with a statement on whether the samples were obtained and processed simultaneously. Quantitative comparisons between samples on different gels/blots are discouraged; if this is unavoidable, it has to be performed for samples derived from the same experiment with gels/blots were processed in parallel, which needs to be stated in the legend.

Regardless of format, all figures must be vector graphic compatible files, not supplied in a flattened raster/bitmap graphics format, but should be fully editable, allowing us to highlight/copy/paste all text and move individual parts of the figures (i.e. arrows, lines, x and y axes, graphs, tick marks, scale bars etc). The only parts of the figure that should be in pixel raster/bitmap format are photographic images or 3D rendered graphics/complex technical illustrations.

Unprocessed scans of all key data generated through electrophoretic separation techniques need to be presented in a supplementary figure that should be labeled and numbered as the final supplementary figure, and should be mentioned in every relevant figure legend. This figure does not count towards the total number of figures and is the only figure that can be displayed over multiple pages, but should be provided as a single file, in PDF or TIFF format. Data in this figure can be displayed in a relatively informal style, but size markers and the figures panels corresponding to the presented data must be indicated.

The total number of Supplementary Figures (not including the "unprocessed scans" Supplementary Figure) should not exceed the number of main display items (figures and/or tables (see our Guide to Authors and March 2012 editorial

<http://www.nature.com/ncb/authors/submit/index.html#suppinfo>; <http://www.nature.com/ncb/journal/v14/n3/index.html#ed>). No restrictions apply to Supplementary Tables or Videos, but we advise authors to be selective in including supplemental data.

GUIDELINES FOR EXPERIMENTAL AND STATISTICAL REPORTING

REPORTING REQUIREMENTS – To improve the quality of methods and statistics reporting in our papers we have recently revised the reporting checklist we introduced in 2013. We are now asking all life sciences authors to complete two items: an Editorial Policy Checklist (found here <https://www.nature.com/authors/policies/Policy.pdf>) that verifies compliance with all required editorial policies and a Reporting Summary (found here <https://www.nature.com/authors/policies/ReportingSummary.pdf>) that collects information on experimental design and reagents. These documents are available to referees to aid the evaluation of the manuscript. Please note that these forms are dynamic 'smart pdfs' and must therefore be downloaded and completed in Adobe Reader. We will then flatten them for ease of use by the reviewers. If you would like to reference the guidance text as you complete the template, please access these flattened versions at <http://www.nature.com/authors/policies/availability.html>.

Version 3:

Reviewer comments:

Reviewer #2

(Remarks to the Author)

The authors have addressed all remaining concerns. Congratulations to an excellent paper!

Decision Letter:

Dear Dr. Prinz,

Thank you for submitting your revised manuscript, "The Vps13-like protein BLTP2 is pro-survival and regulates phosphatidylethanolamine levels in the plasma membrane to maintain its fluidity and function." (NCB-A52106C). It has now been seen by the original referees and their comments are below. The reviewers find that the paper has improved in revision, and therefore we'll be happy in principle to publish it in Nature Cell Biology, pending minor revisions to satisfy the referees' final requests and to comply with our editorial and formatting guidelines.

Thank you again for your interest in Nature Cell Biology Please do not hesitate to contact me if you have any questions.

Best regards,

George Inglis

George Inglis, PhD
Senior Editor
Research Cross-Journal Editorial Team
Nature Cell Biology

Reviewer #2 (Remarks to the Author):

The authors have addressed all remaining concerns. Congratulations to an excellent paper!

Version 4:

Decision Letter:

Dear Dr. Prinz,

I am pleased to inform you that your manuscript, "The Vps13-like protein BLTP2 regulates phosphatidylethanolamine levels to maintain plasma membrane fluidity and breast cancer aggressiveness", has now been accepted for publication in *Nature Cell Biology*.

Over the next few weeks, your paper will be copyedited to ensure that it conforms to *Nature Cell Biology* style. Once your paper is typeset, you will receive an email with a link to choose the appropriate publishing options for your paper and our Author Services team will be in touch regarding any additional information that may be required.

Once your paper has been scheduled for online publication, the Nature press office will be in touch to confirm the details. An online order form for reprints of your paper is available at https://www.nature.com/reprints/author-reprints.html. All co-authors, authors' institutions and authors' funding agencies can order reprints using the form appropriate to their geographical region.

Publication is conditional on the manuscript not being published elsewhere and on there being no announcement of this work to any media outlet until the online publication date in *Nature Cell Biology*.

Please note that *Nature Cell Biology* is a Transformative Journal (TJ). Authors may publish their research with us through the traditional subscription access route or make their paper immediately open access through payment of an article-processing charge (APC). Authors will not be required to make a final decision about access to their article until it has been accepted. Find out more about Transformative Journals

Authors may need to take specific actions to achieve compliance with funder and institutional open access mandates. If your research is supported by a funder that requires immediate open access (e.g. according to Plan S principles) then you should select the gold OA route, and we will direct you to the compliant route where possible. For authors selecting the subscription publication route, the journal's standard licensing terms will need to be accepted, including self-archiving policies. Those licensing terms will supersede any other terms that the author or any third party may assert apply to any version of the manuscript.

If you have not already done so, we strongly recommend that you upload the step-by-step protocols used in this manuscript to protocols.io (<https://protocols.io>), an open online resource that allows researchers to share their detailed experimental know-how. All uploaded protocols are made freely available and are assigned DOIs for ease of citation. Protocols and Nature Portfolio journal papers in which they are used can be linked to one another, and this link is clearly and prominently visible in the online versions of both. Authors who performed the specific experiments can act as primary authors for the Protocol as they will be best placed to share the methodology details, but the Corresponding Author of the present research paper should be included as one of the authors. By uploading your Protocols onto protocols.io, you are enabling researchers to more readily reproduce or adapt the methodology you use, as well as increasing the visibility of your protocols and papers. You can also establish a dedicated workspace to collect your lab Protocols. Further information can be found at <https://www.protocols.io/help/publish-articles>.

Nature Cell Biology encourages authors presenting evidence for cell, biological, molecular, and genetic interactions to consider communicating these findings using Biofactoid (<https://biofactoid.org/>). This tool helps users share a searchable representation of interactions (e.g. binding, gene expression, post-translational modification) between genes, gene products, or chemicals. Information added to Biofactoid, with author attribution, is shared on social media and public databases, such as Pathway Commons, where it can be discovered and analyzed in the context of a large and growing corpus of knowledge.

Best regards,

George

George Inglis, PhD

Senior Editor

[Research Cross-Journal Editorial Team](https://www.nature.com/ncb/research-cross-journal-editorial-team)
Nature Cell Biology

** Visit the Springer Nature Editorial and Publishing website at http://editorial-jobs.springernature.com?utm_source=ejp_NCB_email&utm_medium=ejp_NCB_email&utm_campaign=ejp_NCB for more information about our career opportunities. If you have any questions please click [here](mailto:editorial.publishing.jobs@springernature.com).**

We thank all the reviewers for their thoughtful, constructive comments, which helped us significantly improve our study. Here are our point-by-point responses.

Reviewer #1:

Remarks to the Author:

This manuscript addresses the biological function of a bridge-like lipid transport protein, BLTP2. Inspection of the DepMap database reveals three metastatic breast cancer cell lines with high BLTP2 levels and a requirement for this gene, and its knockout slows HeLa cell proliferation. The yeast homolog, Fmp27, leads to cold sensitivity in the absence of ethanolamine, and this is restored by ethanolamine or an orthoganol head group alcohol, propanolamine. Curiously, whereas Fmp27 yeast have reduced PE levels, PE production through the minor CDP-DAG pathway is unaffected. Fmp27 loss prevents cold-induced increases in PE-PC ratios and decreases in fluidity of the PM (but only in the absence of ethanolamine or propanolamine). Similar findings are made in HeLa cells. Curiously, other aspects of homeoviscous adaptation, such as acyl chain remodeling, are unaffected in these cells. Finally, the manuscript demonstrates that in HeLa cells, BLTP2 synergizes with another PE and fluidity regulating pathway, and that loss of BLTP2 decreases a metastatic breast cancer cell lines' proliferation, survival and metastatic potential in a xenograft model.

Overall, the results provide a link between a bridge-like lipid transport protein and PM enrichment of PE in the PM, and suggest a requirement for cold adaptation in yeast and for metastatic cancer progression in humans. This could represent a novel insight into both the function of these still enigmatic protein bridges, the function of PE at the PM, and the physiological requirement for both. However, the reviewer sees several areas where the manuscript would need to be strengthened to fully support these insights.

(1) Requirements for bridge like function at ER:PM contact sites. The data on PE/PC ratios clearly implicate transport of PE via BLTP2/Fmp27. However, currently there is no demonstration that PE transfer activity of BLTP2 is the mechanism. Can a mutant be identified in the protein that maintains protein architecture and ER:PM contact site localization, but is deficient in lipid transport and that fails to rescue Fmp27 cells or BLTP2 KO? For example, inclusion of anionic residues in the hydrophobic lipid binding cavity? Similarly, can a variant be generated that fails to target ER:PM contact sites? This would conclusively demonstrate a requirement for PE transport from ER to PM.

As suggested, we attempted to introduce mutations into Fmp27 that retain protein structure and localization but block lipid transport. We introduced a ring of charged amino acids into the putative lipid transport groove. We chose the sites for the mutations based on similar mutations in another BLTP, ATG2A (PMID: 30952800, Fig. 2C). However, all the combinations of mutations we tried made the protein unstable and affected its localization.

Instead, we followed the second suggestion of the reviewer and relocalized Fmp27 from ER-PM to contacts to ER-mitochondria contacts. We reasoned that if this increases

PE levels in mitochondria, it will indicate that Fmp27 transports PE. We redirected Fmp27 to ER-mitochondria contact sites in cells that lack the enzyme Psd1, which produces PE in mitochondria; cells lacking this enzyme have a 5-fold reduction in mitochondrial PE and a defect in mitochondrial shape. Strikingly, redirected Fmp27 significantly rectifies mitochondrial morphology defect of cells lacking Psd1 (Fig 3e-g, and Page 6, para 2, lines 1-15). We also showed that enriching Fmp27 at ER-mitochondria contact sites increases PE levels in mitochondria (Fig 3e, h-i, and Page 6, para 2, lines 15-23). Together, these findings support the conclusion that Fmp27 is indeed a lipid transporter that moves PE.

(2) Relationship of BLTP2 to the Kennedy pathway: As the manuscript states on p.9 :”BLTP2 in humans, like its yeast homolog, regulates PM fluidity by increasing the amount of PE in the PM when PE synthesis by the Kennedy pathway is compromised by a lack of exogenous EtN.” This confused the reviewer. The reviewer’s interpretation of the EtN (and PpN) rescue data is that BLTP2 is required to increase PM PE:PC ratios when the Kennedy pathway is inactive, i.e. when cells are deficient in PE synthesis. The only way I can square this in my head is if BLTP2 is required to reallocate limiting amounts of PE from ER to PM... This isn’t required when PE is abundant, and the cell can simply synthesize more. However, since the Kennedy pathway synthesizes PE and PC in the ER membrane, if BLTP2 is primarily functioning as a conduit to traffic PE to the PM, defects should persist even when the Kennedy pathway is active. So why would cells not need BLTP2 in the face of continued PE synthesis? The manuscript does not address this question.

This is an important question. We should have been clearer in our first submission. As the reviewer suggests, we think BLTP2/Fmp27 moves PE from the ER to the PM regardless of whether the Kennedy pathway is active. PE must also be able to reach the PM by other pathways since cells lacking BLTP2/Fmp27 still have PE in the PM, albeit at lower amounts than wt cells. We suspect that these pathway(s) must be sensitive to how much PE is available in the cell. When PE is replete (for example, when ethanolamine is available in the medium) there must be enough PE in cells to maintain sufficient PE in the PM to maintain fluidity. Consistent with this, previous studies have shown that for both yeast and mammalian cells, total PE is higher in cells grown in media with exogenous ethanolamine (PMID: 25571976, PMID: 39312446). We have discussed these points in our revised manuscript (page 6, para 3, lines 1-4).

(3) The specific requirements for BLTP2 in metastatic breast cancers: the data do not really support a general role for BLTP2 in this disease. Firstly, in fig. 1, DepMap data identifies three highly invasive breast cancer cell lines with high BLTP2 expression and dependency. However, when the reviewer reproduced this plot on the DepMap site, it was clear that invasive breast carcinoma cell lines (41 in the DepMap database) do not all have high dependency on BLTP2, nor high expression - and there is no obvious correlation between the two. Notably, the MDA-MB-231 cell line employed in figure 5, although having relatively high expression, has a modest survival score (-0.42). Although comparisons are made to the breast epithelial line MCF10A, the generality of a BLTP2 requirement across multiple invasive breast carcinomas cannot really be established from these limited experiments.

These points are well taken. We are not the first to suggest a connection between BLTP2 and breast cancers; the gene was named Breast Cancer Overexpressed (BCOX) until a few months ago. We also note that the metabolic profile of breast cancer cell lines differs widely between and even within subtypes. Our main point is that BLTP2 is overexpressed and required for optimal growth of many, though not all, breast cancer cell lines.

Rebuttal Fig.1, which shows data from Depmap, supports our suggestion that BLTP2 sustains the proliferation of invasive breast cancer cell lines. The median survival score these these cell lines following deletion of BLTP2 (-0.349) is lower than when ERBB2/HER2 is eliminated (-0.218); ERBB2/HER2 is well known to support the growth of breast cancer cell lines (PMID: 11156524). Similarly, deletion of LAMTOR, a gene critical for mTORC1 activity downstream of ERBB2/HER2 (PMID: 25701120), results in a median survival score of -0.269, similar to that of BLTP2. On the other hand, deletion of VPS13C, which encodes a BLTP2 homolog not implicated in cancer, results in a median survival score of -0.125. Together, findings suggest BLTP2 supports the growth of many invasive breast cancer cell lines.

Rebuttal Figure 1: DepMap survival score of 47 invasive breast cancer cell lines with CRISPR-mediated knockouts of the genes ERBB2, LAMTOR1, BLTP2, or BLTP5C. Colored lines indicate the median survival score, error bars show 95% confidence intervals. The survival score on the x-axis is from 0 (no effect on survival) to -1 (gene is essential for survival).

There are also some minor, technical amendments that would improve the manuscript:

(4) Figure 1J and associate text: Although the method clearly states the source of the tritiated serine, and it is L-[3-3H]-Serine, the position of the tritium label should be indicated on the figure, so the reader does not have to search beyond the manuscript for reassurance that the label is not lost during PS metabolism.

We have incorporated the chemical structure of L-[3-3H]-Serine in Fig. 1j.

(5) Figure 2: the figure legend describes "cells". Although it is clear that these are yeast because Fmp27 is described, the reader might miss this point. The legend would benefit by more explicitly stating that these are experiments on yeast.

We have changed "cells" to "yeast cells" in the figure legend of Fig. 2

(6) In figure 1C, the decrease in PE levels in BLTP2 yeast are clear, but neither the figure nor associated text makes it explicit whether these are EtN replete growth conditions or not.

These cells were grown in DMEM with complete FBS (i.e., EtN replete). We now indicate this in the figure legend and text.

(7) the axis of figure 5A is missing the superscript on the y-axis label "x 10⁵ cells".

The y-axis label is corrected to "x 10⁵" cells in Fig. 5A.

Reviewer #2:

Remarks to the Author:

This study investigates the role of the conserved, Vps13-like lipid transfer protein BLTP2 (Fmp27 in *S. cerevisiae*) in phosphatidylethanolamine (PE)-homeostasis. The authors build a convincing case that Fmp27 is crucial for cell growth at low temperatures and that the growth defects of a Fmp27 deletion are rescued by ethanol supplementation of the medium. The authors show convincingly that the defects in PE regulation of a fmp27D mutant at low temperatures are not due to defects in the de novo biosynthesis of PE. Instead, the subcellular distribution of PE is affected with most significant changes in the plasma membrane. These observations are consistent with a role of Fmp27 as a lipid transfer protein between the ER and the plasma membrane.

Given that the plasma membrane fluidity is affected by FMP27 deletion and rescued by ethanolamine, the authors conclude that Fmp27 may be crucial to provide membrane fluidity in the cold by delivering PE lipids from the ER to the plasma membrane.

Some of the key observations on the important role of BLTP2/Fmp27 are recapitulated in

HeLa cells, thereby highlighting that BLTP2 may work in parallel to a second, previously described pathway (TLCD1) that also regulates plasma membrane fluidity. Strikingly, the interference with BLTP2 expression dramatically reduces growth and aggressiveness of the triple-negative breast cancer cell line MDA-MB-231, but not that of a non-cancerous breast epithelial cell line MCF10A.

This study has several strengths making it a very good candidate for the broad, general audience of Nature Cell Biology. It is timely, by establishing the functional and biomedical relevance of a putative lipid transfer protein. It is original and novel. Furthermore, it demonstrates the evolutionary conservation of a machinery that regulates the PE level in the plasma membrane thereby affecting plasma membrane fluidity. Thirdly, the study elegantly connects lipid metabolism, genetic perturbation in yeast and mammalian cell culture, membrane biophysics and xenograft models to approach a challenging problem in cell biology in a holistic fashion. The conclusions are reliable and the manuscript is clearly written. Nevertheless, there are several crucial points that should be addressed before I can recommend this manuscript for publication.

Major point 1 – related to Figure 1: The experiments where choline, ethanolamine or propanolamine are supplemented to the medium look very convincing. It is very important, however, that the effect of the pH is not overlooked in such experiments as ethanolamine and propanolamine are basic compounds, while choline is not. In fact, the pH is an important regulator of cellular stress resistance and condensation. Unfortunately, the manuscript does not contain sufficient information as to whether the medium was buffered and how the supplementation was performed. It is important to repeat some of the key observations (e.g. as demonstrated in Figure 1D, 1G and/or 1I) with a more carefully controlled pH. If the pH of the medium indeed plays an important role on the observed phenotypes, the authors would have to consider more experiments to dissect the contribution of the pH to the observed phenotypes.

To address this concern, we ruled out that the growth of our yeast strains is caused by the effects of EtN on the pH of unbuffered yeast growth media. We compared the growth of yeast cells in media supplemented with 4 mM EtN that was unbuffered or was buffered with 50mM MES, pH 6 and found no difference (Extended Data Fig. 1d). This indicates that our growth phenotypes are not caused by the effects of EtN on the pH of the media. We have buffered our DMEM media with 25mM HEPES pH 7.4 in experiments where 100 μ M EtN was added. There was no change in the color of phenol red in buffered DMEM after EtN addition, indicating no change in pH.

Major point 2 – related to the previous point and Materials and Methods: Given the important role of lipids in this study and the important impact of cell cultivation and media composition on the PC-to-PE ratio, the authors should clearly indicate and media composition and their (complete!) cultivation protocol for each experiment.

We now clearly indicate the growth media composition in the figure legends and the Materials and Methods section.

Major point 3 – related to Figures 2D and 4F: Membrane fluidity in the plasma membrane is measured by two different methods in two different systems, which is fine. However, the authors do not sufficiently discuss the fact that both reporters only incorporate in the outer leaflet of the plasma membrane, while PE is highly enriched in the inner leaflet. Hence, the mechanism how PE regulates membrane fluidity in the outer membrane leaflet remains entirely unclear. If membrane fluidity or other plasma membrane characteristics (e.g. lipid packing defects in the inner leaflet, which may help recruiting certain signaling-active proteins) are the relevant parameters for the cellular phenotypes remains unexplored. Especially in the context of the breast cancer cell line, it may be very interesting to learn, if the recruitment and signaling at the plasma membrane (e.g. by AKT) is affected upon BLTP2 depletion and/or ethanolamine supplementation.

The point is well taken. We note there are several studies indicating that lipids on the inner leaflet of the PM or an artificial membrane can affect the order of the outer leaflet. For example, PMID: 25910209; PMID: 37533258; PMID: 29320681. The last study suggests lipids with high intrinsic negative curvature, like some PEs, can promote inter-leaflet coupling. We have added a brief discussion of transbilayer coupling (page 5, para 1 and lines 3-4).

Major point 4 – related to Figure 4A: I am missing a WT control for this experiment.

This data is now incorporated in Extended Data Fig. 5a.

Major point 5 – related to Figure 4 g: The authors conclude that 'we found that the relative abundance of the major phospholipid species does not change in BLTP2-LO cells'. This is not correct for PE lipids (lipid species involving e.g. 18-1;18-1 are substantially more abundant). Given that PE lipids are at the center-stage of this study, this is a relevant observation, which should be discussed. The authors should provide a bar diagram (showing also the 3 replicates of this measurement). In fact, some repercussions of a perturbed species compositions are also seen in PC lipids. Is there any general difference in the acyl chain composition of PE/PC lipids when comparing yeast and mammalian cells?

We have added bar graphs showing the lipid species PE-18:0_18:2 and PE-18:1_18:1 for WT and BLTP2-KO since they are the ones with significant differences (Extended Data Fig. 5f,g). These changes likely contribute to the lipid landscape and membrane fluidity and suggest BLTP2 may have different affinities for some PE species. We have changed our sentence in the manuscript to “we found only minor changes in the relative abundance of the major phospholipid species in BLTP2-KO cells” (page 8, para 1 and lines 2-3).

Major point 6 – related to selectivity of transport: The authors explain that the basis transport directionality is not understood for any Vps13-like lipid transfer protein. However, the authors do not comment on the specificity/selectivity of transport. How comes that PE lipids are particularly well transported by this machinery? The authors should at least comment on this point.

These points are well taken, but we only have partial answers. Our finding that relocating Fmp27 to ER-mitochondria contacts increases PE in mitochondria (see response to Rev 1, pt1 and Fig. 3e-i), indicates that Fmp27 moves PE, but we cannot rule out that it moves other lipids as well. If Fmp27 has specificity for PE, it is possible that Fmp27 binds PE or works together with partner proteins that determine which lipids are transported. The finding that PE acyl chain composition does not change much when BLTP2/Fmp27 are depleted could indicate that the proteins have similar affinities for most PE species, though, as the author points out, there may be some differences. We do not know how transport specificity is determined or what drives the movement of lipids through BLTP2/Fmp27. These points are briefly discussed (page 6, para 2, last line, page 9, para 2, lines 3-4) and will be the topic of future work.

Minor point 1 – p4: The authors refer to PE lipids as lipids with 'high negative monolayer spontaneous curvature' that 'forms hexagonal phase structures rather than membrane bilayers'. This, however, is only true for PE lipids with two unsaturated lipid acyl chains such as DOPC. The authors should either add more data regarding the lipid species composition of PE or adjust their discussion on the role of PE lipids.

This is an important point and we should have been more careful in our first submission. We have now adjusted our discussion of PE species.

Minor point 2 – related to Figure 1E: The experiment convincingly shows that Fmp27 is much more abundant in cells cultivated at 18°C. However, it also seems that Por1, which was the loading control, is lower. Could the authors comment on this? Does that imply that the mitochondrial abundance is reduced at low temperatures (thereby affecting mitochondrial lipids)?

As the reviewer points out, it is possible that mitochondrial abundance is reduced at low temperatures, though we only measured one mitochondrial marker (Por1). Even if there is a reduction, it does not change our interpretation of our findings.

Minor point 3 – related to Figure 2B: I wonder if there is a problem with the Western blot related to the plasma membrane isolation by density centrifugation. 14 fractions were taken from the gradient and all 14 fractions are shown. However, all four marker proteins show an abrupt change between fraction 7 and 8. Related to this, the vacuole marker Vph1 has two populations. Is this real, were the Western blots accidentally mirrored or are some fractions missing?

The Western Blots are not mirrored and no fractions are missing in Fig. 2b; the original blots are now included in the supplemental information. We and others have used this method before with similar results (PMID: 15316012; PMID: 21689253).

Minor point 4 – related to Figure 1G, Figure 3E, Figure 3F and the text on p6 (top): The text in the manuscript is misleading. The authors should clearly indicate that inositol seems limiting for the growth of both WT and *fmp27D* cells. The authors should also indicate that inositol speeds up growth of the *fmp27D* quite substantially. The authors should also comment on the different doubling times observed between Figure 1G and 3F.

Given that the UAS-INO system is controlled by PA in the ER membrane (Loewen et al. 2003 and 2004) and given that PA is much more abundant in *fmp27D* cells (Figure 1H), how comes that there is no 'specific' effect on *fmp27D* upon inositol supplementation. Can the authors comment?

We should have indicated that an inositol-free medium was used for these growth experiments, which is now stated in the text and figure legend. Therefore, it is not surprising that the addition of inositol improves the growth of both wt and *fmp27Δ* cells. The main point of these growth experiments is that inositol addition increases the *relative* growth rate of wt and *fmp27Δ* cells similarly (Fig. 3F, right panel), suggesting that there are no major differences in the regulation of phospholipid metabolism between the strains. It should also be noted that an inositol-free medium was not used for the results shown in Fig. 1G, explaining why they differ from those in 3F.

Minor point 5: The Material and Methods section should be carefully adopted to allow the reader to repeat the experiment. Referring to a previous paper (e.g. as for the plasma membrane 'rip-off' from HeLa cells or for the TMH-DPH experiment) is not recommended.

We have improved the Material and Methods section.

Reviewer #3:

Remarks to the Author:

This is an interesting paper that describes the role of BLTP2/Fmp27, a member of the Vps13 family of bridge-like lipid transfer proteins, in regulating plasma membrane fluidity through phosphatidylethanolamine (PE) homeostasis. The experiments reveal unexpected outcomes of disrupting this protein in yeast and cancer cells.

The paper is hard to read as the authors seem to have thrown everything they have at the problem without much selection. The figures are not well organized, and they are aesthetically poor, making reading even more difficult. Also, the paper is primarily descriptive - it identifies an interesting role for BLTP2/Fmp27 but leaves open the question of what makes this pathway work. This question is highlighted in the final paragraph where

the authors admit that they do not know what drives PE to the PM and what is the underlying role of this pathway in cancer.

We now agree that we have improved the writing and presentation of data in the main figures and Extended Data figures.

We added data that strongly suggests BLTP2/Fmp27 directly transports PE in cells (see response to Rev 1, pt 1), providing mechanistic insight into how the transporter regulates PM fluidity. We feel that determining what drives PE to the PM is beyond the scope of this study, which identifies a new mechanism cells use to maintain PM fluidity.

Notes on the figures:

Fig 1a - conventional x-y axes please, with y-axis on the left.

We have made this change.

Fig 1b - scale this better so that bar chart is clearly associated with 1b and not 1c (reduce the size of the image to make room for the bar chart!); the bar chart needs a time unit on the y-axis as it represents a rate of proliferation.

We made the suggested changes.

Fig 1d- indicate SC medium (the corresponding figure is S1b where YPD is indicated).

We have indicated SC medium as suggested.

Figs 1d, S1b and S1c - these data should be shown together in Fig 1 to make things easier for the reader.

While we agree that it would be good to include the supplemental data in Fig 1, we did not make this change because it makes the figure too crowded and difficult to read.

Fig 1e- blot replicates are not necessary to show; single lanes would be fine so that the bar chart can be moved alongside.

We have moved the bar chart so that it is next to the blots.

Figs 1i and 1k - switch these around to match the order in which they are presented in the text.

We have made this change as suggested.

Fig 2b - the fractionation quality as seen by the blot seems rather poor. The bar separating fractions 1-7 and 8-14 provides a sharp cut-off of the signals corresponding to

all the markers used. This should be done using a 15-well gel or run fractions 5-11 on one gel is the number of wells is limiting.

While it would have been preferable to run fractions 5-11 in one gel, as the reviewer suggests, the results would not be different. This type of gradient has been successfully used by our lab and others to isolate fractions highly enriched in PM. The results in these previous studies are similar to those here (PMID: 15316012; PMID: 21689253).

Figs 3a and 3b- these are black-boxes - they are not explained properly and consequently offer little information. They should be eliminated or moved to a supplementary figure.

We have eliminated the black boxes as suggested by the reviewer.

Figs 3c and 3d - these could be combined with Fig 2 as the information they provide is consistent with other data shown in Fig 2

We thank the reviewer for this suggestion, however, the limited space in Fig 2 did not allow us to make this change. We have also revised Fig. 3.

Figs 3e and 3f - these could be moved to a supplementary figure.

We made this change as suggested.

Fig 4a- this is supposed to indicate a growth rate but there is no time unit.

We have added a time unit to the y-axis. Note that cells were seeded at an equal number for each condition on day zero.

Other notes:

Fmp stands for 'found in mitochondrial proteome' and a previous paper by the authors (Toulmay et al) described localization of Fmp27 to ER-PM and ER-mito contact sites. The authors should elaborate on this, given the role of mitochondria in PE homeostasis and the uniquely PM-adjacent staining of Fmp27 shown in Fig S1a.

As the reviewer points out, Fmp27 is partially localized to ER-mitochondrial contacts, and our previous study suggests most of the protein is at ER-PM contacts. The role of Fmp27 in mitochondrial PE homeostasis remains to be determined. We have now added data showing that forcing Fmp27 to ER-mitochondrial contacts increases PE levels in mitochondria (Fig 3e-i), which suggests Fmp27 could normally transport PE to mitochondria. It is possible that some stresses increase Fmp27 localization to ER-mitochondria contacts, resulting in increased PE in mitochondria. This will be the subject of future work.

.

Page 3 - what is 'rate of flux'? Flux represents a time-based measurement, so the word rate is unnecessary.

We agree and have removed the phrase 'rate of flux' from the manuscript.

Page 3- radiolabeling for 60 min does not provide a measure of synthesis rate (or flux??) unless the authors show that this represents a time point within a linear range of continuous incorporation of the label, in which case they could represent the data as per min or per hour.

We previously showed that [³H]serine radiolabels PS, PE, and PC at linear rates in cells grown at 30°C (PMID: 18836080). We now demonstrate this is also true for cells grown at 18°C up to 3 hours after [³H]serine addition (Extended Data Fig. 1g).

Page 4- it would be interesting to know if the ergosterol content of the PM fraction is altered in the fmp27 cells (ergosterol is a mediator of PM fluidity as the authors demonstrate in Fig 2d).

This is an important point. We now show that PM levels of cholesterol (equivalent to yeast ergosterol) do not change in HeLa cells lacking BLTP2 (Extended Data Fig. 5h), suggesting BLTP2 does not regulate cholesterol levels in the PM. This finding suggests BLTP2/Fmp27 primarily regulates PM fluidity by modulating PE levels, not sterol levels.

Page 4- it would also be interesting to know if the fmp27 cells are differently sensitive to duramycin compared with wild-type cells under the various conditions tested.

This is an interesting suggestion. However, in our hands, the sensitivity of wt cells to duramycin is variable and is affected by factors other than PM PE levels. We feel that measuring PE levels in the PM is more direct and useful than assessing duramycin sensitivity.

Page 5, middle para- the comment about the ELSD detector is unnecessary; presumably the authors simply want to make a distinction between a mass measurement of lipids, reflecting actual pool sizes, and a short-term radiolabeling experiment which is not taken to steady state.

We have removed the comment as suggested.

Page 6- the section of *C. elegans* seems gratuitous and should be dropped.

We respectfully disagree. The *C. elegans* data is further evidence that our proposed role for BLTP2/Fmp27 in PE homostasis is conserved and we think it is appropriate to include the data in this study.

Page 7, second para, last line- should be S3g not S3e.

Thanks for noting! This figure is now Extended Data Fig. 6c in our revised manuscript.

Reviewer #1

The reviewer appreciates the thorough response to both my own and the other reviewer's comments. I believe the major concerns with the original manuscript are now sufficiently addressed to warrant publication in the current form, pending minor revisions. In response to my major concerns, I only have the following minor comments:

(1) The reviewer hugely appreciates the elegant experiments added in Fig. 3e, h, and I to address the PE transport function of Fmp27. This indeed addresses my major concern that PE transport is truly required for the phenotypes reported elsewhere, and I believe strengthens the manuscript greatly.

Thank you.

(2) I also agree with the author's argument that Fmp27 becomes critical after inhibition of the Kennedy pathway because when it is active, other pathways can probably deliver sufficient PE to the PM, albeit inefficiently. My only minor quibble is can the authors suggest the nature of this pathway? I.e. other lipid transporters (E-Syts, etc.) and, of course, vesicular transport which likely shuffles these abundant phospholipids around.

I suspect it is largely vesicular trafficking. This is now mentioned in the text at the end of the Results subsection "*Fmp27 facilitates PE transport in cells.*"

(3) To my concern about the generality of BLTP2 requirements for breast cancer cell line growth, I can only say: Touché! The reviewer is convinced by the data included in Rebuttal figure 1, as should anyone who has ever prescribed trastuzumab to a cancer patient. I would suggest including these data and pointing them out in the manuscript, perhaps in the supplement.

Thank you. We have added the data to Extended Data Fig. 8a and added the text in page 10, paragraph 3, lines 1-7

Reviewer #2 (Remarks to the Author):

The authors have done a good job in addressing the points raised by the reviewers. Especially the experiment of redirecting Fmp27 is elegant. However, there are two points that I feel should be addressed:

With respect to major point 3, the authors state in the text on page 5, para 1 'While PE is primarily localized to the inner leaflet of the PM²⁶, there is evidence that the composition of the inner leaflet of the PM affects the fluidity of the outer leaflet^{27 28, 29.}' (missing comma between 27 and 28). Importantly, Reference 28 describes interleaflet coupling for non-fluid bilayers and states: 'That is, the structure of fluid membranes is dominated by layer-specific membrane properties and is not influenced

by that of the opposing leaflet.' Hence, they provide evidence against interleaflet coupling in living (fluid) membranes. The authors should consider rephrasing this section.

Thank you for pointing out our misinterpretation of Ref 28. We have now rephrased this section.

With respect to minor point 3 (Western Blots), I am still puzzled by the abrupt change in marker intensity, which is also not described in the references referred to by the authors. I also do not find evidence in the referred papers for two populations of a vacuolar marker. I trust the authors that their protocol enriches the plasma membrane sufficiently, but I remain confused about whether the source data and the data in Fig 2b fit together. I wonder if there were some peculiarities in the fraction experiment, which are not being described.

Specifically, I still have a hard time seeing how the data in Fig. 2B and the source data for Fig. 2b are from the same experiment indeed. I do not exclude the possibility that things are okay, but if were the case, there must have been numerous image processing steps involved and some of them rather extreme and potentially unevenly applied to the different blots. I would recommend using either less processed images for Fig 2B, or indicating the individual processing steps (step-by-step from source data to presented data).

Confusing observations are for example: The Pma1 blot appears as a double band in the source data, only one band is shown in 2B. For Vph1, the band intensity for fractions 1-7 is much higher in Fig 2b compared to the source data, but the band intensity for fractions 10-14 barely changes. The background for the bands 1-7 and 8-14 is grayish, but white in Fig 2b. The relative band intensity for Dpm1 compared to the background signal appears higher than observable in the source data.

Given that reviewer 3 was also irritated by the Western Blot data, I recommend the authors provide a more transparent view on how the data were processed or present less processed data. Also, they should comment on the distribution of band intensities in the manuscript in order to prevent confusing/irritating the reader.

Other than that: An excellent manuscript!

To address these concerns and similar ones from Rev 3, we first re-made Fig. 2b using the raw images of the blots. The result is below (re-done old Fig. 2b). The only way the source images of the blots were modified is that they were cropped and horizontally flipped, since the fractions run backwards in the original images (i.e., 7-1 and 14-8). The brightness and contrast have not been adjusted, and the images are not altered in any other way. Rev 2 also asked why Pma1 runs as a double band in the original but not in 2b. We think the upper band is probably ubiquitinated Pma1. Both bands are now shown. We feel the re-done Figs 2b makes a better case that fractions 8-10 are PM-enriched and relatively free of other organelle markers. However, we wanted to make this figure stronger and have now entirely redone the blots for Fig. 2b. They are shown below (new Fig 2b). The new Fig. 2b now also includes blots of the homogenate as Rev 3 suggested. Please also note that the Dpm1 signal in the original Fig. 2b was very weak because the antibody was old. After obtaining new antibody, we have much

stronger Dpm1 signal. We feel the new Fig. 2b makes a stronger case that fractions 8-10 of the gradients are enriched in the PM and have only a small amount of material from the vacuole and trace amounts of material from the ER and mitochondria.

Re-done old Fig. 2b- using the raw images of the blots:

New Fig. 2b. H = homogenate, numbers indicate pooled fractions:

Note that proteins from the homogenate run slightly faster than the samples from the gradients because they do not contain Renografin.

Here is the source data for the new Fig. 2b:

Reviewer #3 (Remarks to the Author):

The paper is much clearer, and the authors have responded for the most part to my comments and those of the other reviewers. I have some further notes:

1. I am still not happy with the western blot in Fig 2b, and these data were also flagged by reviewer 2. It may be that these are standard experiments, but the data quality is pretty poor. This is an important technical step as it defines the PE increase, reflected in the PE/PC ratio shown in Fig. 2c. Perhaps the authors could re-assess their PM fraction in comparison with homogenate, in the same way that they assess the immuno-isolated ER fraction in Fig 2e and mito fraction in Fig 3h. They could also measure ergosterol/phospholipid in the PM fraction versus fractions 2-7, to see if it is high as expected.

Please see the response to Rev 2, pt 2.

2. The authors comment briefly that specific methodologies account for the different PE:PC ratio of total cell lipid extracts shown in Fig 2a versus Fig 2f, where the ratio (at 18C for WT cells) is >1 (Fig 2a) versus approx 0.5 (Fig 2f). These are very different ratios with significant implications for membrane structure, and for the overall thesis of this paper, so what ratio value should one be thinking about? I guess that a PE/PC of approx 0.5 would be typically of most cell membranes.

We think the PE:PC ratios in Fig. 2a are closer to values one should be thinking about. To quantify the lipids for Fig. 2a, we used an evaporative light scattering detector. We could not use this detector for the results in Fig. 2f because we had much less material and were below the linear range of the detector. Instead, we labeled cells to steady-state with [³H]palmitate. This will not uniformly label all species of PE and PC, while the evaporative light scattering detector should detect all species with about the same efficiency. Therefore, we feel the results in Fig. 2a are closer to the truth.

3. Figure 3a and 3b remain baffling. My comment that these were 'black boxes' was to imply that they did not have any useful or readable content – the authors assumed that I meant the black outlines that circumscribed the graphs and simply removed these outlines! Quite humorous, actually. The figure legend states: Volcano plots showing log fold changes (Log₂ FC) of PL species against statistical significance (-Log p-value). What PLs are being looked at? What do the symbols mean?

Sorry we misunderstood what you were asking for; at least we provided something humorous. Each point on the volcano plots is a PL species. The graphs show the fold change of each PL species in *fmp27Δ* cells relative to wt. We have now color coded the points in the graphs so that PLs of the same type (PC, PE, etc.) are the same color. We added this sentence to the Results section to help explain the graphs: “Volcano plots of the fold change of phospholipid species in cells lacking Fmp27 compared to wild-type show that all changes are either statistically insignificant changes or less than 2-fold.”

4. Figs 3e-i are a useful addition to the paper!

Thank you.

-There are several files in our system that have not been labeled (including, "Related Manuscript Files" that correspond to Extended Data figures). Please ensure that all submitted items are clearly labeled to avoid any confusion.

Sorry. Everything is labeled now.

-Given that the numerical data for Ext Data Fig 6a is provided in the source data, we would still ask that the same data be provided for Ext Data Fig 6b, for consistency

Done.

-Thank you for adding your lipidomics data to OSF, though please cite the DOI (10.17605/OSF.IO/RP7HE) as an in-text, numerical reference (ex. ref #72) in the Data Availability

Done.

-Thank you for adding the overexposed image of Fmp27-3xHA blot related to Fig 1e, though this does not appear to match the cropped version in the main text.

We have added labels and text to explain what part of the source data blot is shown in the Fig 1e. We also added an image of the uncropped image adjusted as in main figure.

- Please upload main figures in PDF, EPS, or AI format.

The pptx files were uploaded. If they are good enough, we will generate ai files.